# CHARTBENCH: A BENCHMARK FOR COMPLEX VISUAL REASONING IN CHARTS

## ABSTRACT

Multimodal Large Language Models (MLLMs) have shown impressive capabilities in image understanding and generation. However, current benchmarks fail to accurately evaluate the chart comprehension of MLLMs due to limited chart types and inappropriate metrics. To address this, we propose ChartBench, a comprehensive benchmark designed to assess chart comprehension and data reliability through complex visual reasoning. ChartBench includes 42 categories, 66.6k charts, and 600k question-answer pairs. Notably, we do not provide data point annotations on charts explicitly, which requires models to derive values by leveraging inherent chart elements such as color, legends, and coordinate systems. We also design an enhanced evaluation metric named $Acc++$ to evaluate MLLMs without extensive manual or costly LLM-based evaluations. Furthermore, we propose two baselines based on the chain of thought and supervised fine-tuning to improve model performance on unannotated charts. Extensive experimental evaluations of 18 open-sourced and 3 proprietary MLLMs reveal their limitations in chart comprehension and offer valuable insights for further research.

## 1 INTRODUCTION

Given the groundbreaking advancements in Large Language Models (LLMs) Radford et al. (2021); Brown et al. (2020); Chowdhery et al. (2023); Touvron et al. (2023a), Multimodal Large Language Models (MLLMs) Li et al. (2023c); Liu et al. (2023e); Zhu et al. (2023) have become the leading approach in multimodal learning, which exhibit excellent visual semantics understanding performance OpenAI (2023); Wang et al. (2023b). However, existing MLLMs face challenges in effectively reading, comprehending, and summarizing articles that contain embedded charts Masry et al. (2022); Han et al. (2023); Li & Tajbakhsh (2023). Unlike natural images, which are typically interpreted based on discernible objects, relative positions, or interactions, charts convey nuanced semantic meanings through *visual-grounded logic*, such as trend lines or color-coded legends. They present detailed and intricate data narratives in visual formats, making it essential to evaluate MLLMs' chart comprehension ability and data reliability in understanding these visual representations.

Previous works Masry et al. (2022); Methani et al. (2020); Kantharaj et al. (2022a); Xia et al. (2024); Chen et al. (2024a) have attempted to address this issue but have encountered some limitations. 1) They primarily focus on 3 regular chart types (i.e., line, bar, and pie charts), neglecting more intricate formats such as scatter or combination charts, which are equally prevalent in real-world scenarios. Robust MLLMs should be able to adeptly handle a diverse range of chart types. 2) They heavily rely on *datapoint annotation* on charts or *meta table data* as textual prompts Masry et al. (2022); Han et al. (2023); Chen et al. (2024a) to generate content, allowing models to easily obtain candidate answers while ignoring the charts' *visual-grounded logic*. This will cause MLLMs to struggle with unannotated charts in real-world applications. 3) Current evaluation metrics like judgment or multi-choice question cannot avoid lucky guesses and thus result in overestimated baseline performance, which requires refinement to enhance assessment objectivity and precision.

To address these limitations, we introduce ChartBench, which comprehensively evaluates the performance of MLLMs on a wider variety of chart types, including both annotated and unannotated charts. As summarized in Tab. 1, ChartBench includes over 68k charts and more than 600k high-quality instruction data, covering 9 major categories and 42 subcategories of charts. ChartBench has 5 chart question-answering tasks to assess the models' cognitive and perceptual abilities. Each subclass

Table 1: Comparative analysis with the existing benchmarks for chart-related evaluations. *Aggregated* charts are derived from consolidating existing datasets. # refers to the corresponding quantity. * refers to the lack of explicit task-type labeling. *Visually Grounded* indicates that models are required to answer queries via interpreting the visual logic of charts without relying on OCR. Please refer to Appendix A.1 for specific cases.

| Benchmark | Image Source | Type | | Train Set | | Test Set | | Multi-task Evaluation | *Visually Grounded* |
| | | #Chart | #Task | #Chart | #QA | #Chart | #QA | | |
|---|---|---|---|---|---|---|---|---|---|
| ChartQA Masry et al. (2022) | *Original* | 3 | 1* | 21.9K | 32.7K | 1.5K | 2.5K | ✗ | ✗ |
| PlotQA Methani et al. (2020) | *Original* | 3 | 1* | 224K | 28M | 33.7K | 33.7K | ✗ | ✗ |
| Chart-to-text Kantharaj et al. (2022b) | *Original* | 6 | 1* | 44K | 44K | 6.6K | 6.6K | ✗ | ✗ |
| OpenCQA Kantharaj et al. (2022a) | *Original* | 5 | 1* | 6.5K | 6.5K | 1.2K | 1.2K | ✗ | ✗ |
| UniChart Masry et al. (2023) | *Aggregated* | 3 | 3 | 627K | 7M | - | - | ✔ | ✗ |
| ChartLlama Han et al. (2023) | *Original* | 10 | 7 | 11K | 160K | 2.1K | 3.5K | ✔ | ✗ |
| MMC Liu et al. (2023c) | *Aggregated* | 6 | 9 | 600K | 600K | 2K | 2K | ✔ | ✗ |
| ChartX Xia et al. (2024) | *Original* | 18 | 7 | - | - | 6K | 6K | ✔ | ✔ |
| ChartBench (ours) | *Original* | 9 / 42 | 5 | 66.6K | 599.6K | 2.1K | 18.9K | ✔ | ✔ |

Table 2: ChartBench comprises 3 regular charts and expands to include 6 additional types. ChartBench emphasizes charts that lack data point annotations, requiring the MLLMs to infer the correct answers by considering elements such as color, legends, and coordinate systems like humans.

| Data Split | Annotation Distribution | | Chart Type Distribution | | | | | | | | |
| | *w/i* | *w/o* | Line | Bar | Pie | Area | Box | Radar | Scatter | Node | Combin. |
|---|---|---|---|---|---|---|---|---|---|---|---|
| Train Set | 15.04% | 84.96% | 11.75% | 36.89% | 12.72% | 8.42% | 6.11% | 4.59% | 3.07% | 5.97% | 10.47% |
| Test Set | 23.80% | 76.20% | 11.90% | 31.00% | 11.90% | 7.10% | 7.10% | 9.50% | 7.10% | 4.80% | 11.90% |

in the test set contains at least 50 table-chart pairs sourced from the real world. Additionally, we generate more samples with different chart prototypes based on the code rendering to construct the train set. We implement a hierarchical quality control process, with detailed information available in Appendix B. Experimental results show a significant performance gap between charts with and without datapoint annotations (Tab. 6). To enhance model capabilities on unannotated charts, over 80% of the training set in ChartBench are unannotated charts (Tab. 2). The significant performance improvement on the ChartQA and ChartBench test set achieved through supervised fine-tuning demonstrates the effectiveness and applicability of the ChartBench train set.

We further improve the *Acc+* metric introduced by MME Fu et al. (2023a), where MLLMs can only score if they correctly answer a query from both affirmative and negative views. The negative query is typically generated by simply negating the affirmative statement, usually by adding *not* before the verb. However, the semantic differences between these two forms are substantial and do not effectively prevent the model from making lucky guesses. To address this, we propose generating the negative query by randomly replacing the ground truth value from the same meta table, named *Acc++*. This approach generates two views with similar representational and semantic embedding features, thereby reducing instances of lucky guessing. If the model fails to accurately interpret the chart's visual information, it will provide identical responses and fail to get the *Acc++* score.

The evaluation of 18 mainstream open-source and 3 closed-source models shows that current MLLMs cannot effectively understand complex charts, especially those without data annotations, raising concerns about the reliability of their data interpretation. Detailed examinations on ChartBench reveal the reasons behind the suboptimal performance of MLLMs on charts, highlighting ChartBench's meticulous curation to explore the nuances of chart reasoning. We introduce two simple yet effective baselines based on the chain of thought (CoT, Fig. 4) and supervised fine-tuning (SFT) to improve MLLMs' performance on ChartBench, aiming to inspire more innovative proposals in the future.

Our contributions can be summarized as follows:

a) We introduce ChartBench, a large-scale dataset with over 42 types of charts, 66k charts, and 600k instructions. It primarily includes charts without data point annotations, assessing MLLMs' ability to reason through visual elements instead of OCR.

b) We refine the *Acc+* metric and value matching criteria to effectively reduce random guesses and provide more robust evaluation results of 18 open-sourced and 3 closed-sourced MLLMs.

c) We propose two efficient baselines based on the chain of thought and supervised fine-tuning, inspiring more methods to enhance MLLMs' understanding of unannotated charts.

d) Extensive experiments reveal existing MLLMs' inadequacies in chart comprehension, highlighting potential directions for future optimization.

## 2 RELATED WORKS

### 2.1 MULTIMODAL LLMS

Current LLMs (Vaswani et al., 2017; Radford et al., 2018; Brown et al., 2020; Zhang et al., 2022; Chowdhery et al., 2023; Touvron et al., 2023a;b; Cai et al., 2024) successfully bridge the multimodal areas via instruction tuning (Ouyang et al., 2022; Li et al., 2023a; Wang et al., 2022). The connectors are proposed to align visual and text modality to train MLLMs Chen et al. (2022); Alayrac et al. (2022), e.g., Q-Former (Li et al., 2023c) or MLP Bai et al. (2023b). Mini-GPT4 (Zhu et al., 2023; Chen et al., 2023a), mPLUG-Owl (Ye et al., 2023b), and InstructBLIP (Dai et al., 2023) extend language-only instruction tuning to multimodal tasks using Q-Former. LLaVA (Liu et al., 2023e;d) maps visual features into the LLaMA (Touvron et al., 2023a) embedding space by a linear layer, while concurrently fine-tuning with LLaMA. The closed-source Baidu ERNIE BaiDu and GPT-4 (OpenAI, 2023) further show satisfactory image understanding capabilities. Despite the impressive achievements of existing MLLMs (Ding et al., 2021; Du et al., 2022; Zhang et al., 2023; Bai et al., 2023b; Chen et al., 2023b; Lin et al., 2023) in common multimodal tasks like VQA (Antol et al., 2015) and image captioning (Vinyals et al., 2015), their focus tends to be on general image understanding, neglecting the specialized task of comprehending chart data in domain-specific contexts (Masry et al., 2022; Li & Tajbakhsh, 2023; Han et al., 2023; Liu et al., 2023c; Xia et al., 2023). Existing research can be divided into two categories. 1) two-stage methods mainly transform multimodal queries into text QAs by extracting table information as prompt Lee et al. (2023); Liu et al. (2023b;a); Xia et al. (2024). 2) end-to-end approaches adopt chart-question pair data to align and supervised fine-tune the MLLMs Han et al. (2023); Carbune et al. (2024); Meng et al. (2024); Liu et al. (2023c); Ye et al. (2023a); Liu et al. (2024); Wang et al. (2023a); Zhuowan et al. (2024); Yan et al. (2024); Chen et al. (2024a); Zhang et al. (2024). Although these efforts have improved the chart understanding ability of MLLMs, there are still limited benchmarks to properly evaluate their performance on the charts, especially unannotated ones.

### 2.2 MULTIMODAL BENCHMARKS

MLLMs have been fully evaluated on numerous traditional benchmarks (Goyal et al., 2017; Hudson & Manning, 2019; Xu et al., 2023; Ye et al., 2023c; Fu et al., 2023a; Yu et al., 2023; Li et al., 2023b; Liu et al., 2023f), while largely ignoring the requirement for complex visual chart understanding and reasoning. HallusionBench (Guan et al., 2023) exposes the susceptibility of formidable models like GPT-4V (OpenAI, 2023) and LLaVA-1.5 (Liu et al., 2023d) to severe hallucinations when confronted with complex charts. VisText (Tang et al., 2023) introduces a benchmark to incorporate multi-level and fine-grained chart labeling, covering aspects such as chart construction, summary statistics, relations, and complex trends. SciCap (Hsu et al., 2021), Chart2Text (Kantharaj et al., 2022b), AutoChart (Zhu et al., 2021), and ChartSumm (Rahman et al., 2023) address chart-to-text summarization tasks. ChartQA Masry et al. (2022) and PlotQA Methani et al. (2020) are currently mainstream benchmark datasets for evaluating the chart comprehension abilities of MLLMs, which focus on three commonly encountered chart types. Chartllama Han et al. (2023) and ChartX Xia et al. (2024) expand the range of available chart types, while ChartY Chen et al. (2024a) significantly expands the number of regular chart types with LLMs. However, these benchmarks have limited chart types, and their charts are always accompanied by detailed datapoint annotations, which allow MLLMs to obtain candidate answers via simple OCR. Comparatively, the advantages of ChartBench stem from its larger scale, more diverse chart types, richer plot styles, and high proportion of unannotated charts.

## 3 CHARTBENCH

### 3.1 DATA PROCESSING PIPELINE

Fig. 1 illustrates the specific data processing flow of Chartbench. The core idea is *to generate unannotated charts of various types and their corresponding instruction data.*

**Data Collection**. To design charts reflecting real-world scenarios, we gather themes and data suitable for scientific research from Kaggle, anonymizing all real names and identifiable entities to ensure

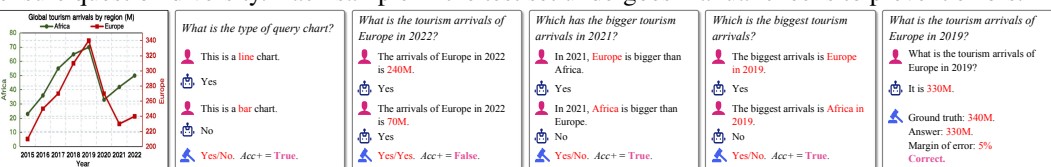

Figure 1: Illustration of the overall data collection and annotation pipeline. We adopt desensitized and GPT-generated data. We employ various charting methods, styles, and color combinations to ensure chart diversity. We provide over 200 question templates and GPT-generated questions to ensure question diversity. Each sample in the test set undergoes manual checks to prevent errors.

Figure 2: Illustration of five proposed tasks. Tasks (a-d) are with $Acc++$ and (e) with GPT-acc metric.

privacy. To ensure the diversity of chart types, we also use LLMs Radford et al. (2019); Bi et al. (2024); Bai et al. (2023a) to generate realistic virtual themes and data for additional chart types.

**Data Filtering**. We establish standard JSON formats for 42 chart types and filter out all table data that does not conform to these standards to ensure proper code rendering. We further remove insufficiently differentiated data (such as data with small differences between maximum and minimum values) to avoid creating confusing charts.

**Chart Generation**. With effective data filtering, we plot various charts using various chart plotting libraries (such as *Matplotlib*, etc.). We randomly applied different plotting styles and color schemes to ensure chart diversity and provide 9 major categories and 42 subcategories of charts (Tab. 2). Refer to Appendix A & H for detailed descriptions and thumbnail visualizations. Specifically, we designate a proportion of charts without data point markers, which is a significant feature of ChartBench.

**Instructions Generation**. We set 5 different tasks for each type of chart and propose $Acc++$ for evaluation. Detailed instruction tasks will be explained in Sec. 3.2. The goal is to evaluate the conception and perception capabilities, especially on the chart with no data-point annotations.

**Dataset Splitting**. We randomly select 50 samples for each chart type to build the benchmark, with the specific distribution shown in Tab. 2. Unlike the training set, which uses code generation, we re-render these charts using *online plotting websites* to ensure there is no domain gap with real-world charts. We also employ both automated and manual reviews to ensure the quality and diversity of the charts. Refer to Appendix B for details.

## 3.2 AUTOMATIC INSTRUCTIONS GENERATION

ChartBench consists of 5 tasks, encompassing *perception* and *conception* Fu et al. (2023a) tasks. *Perception* tasks primarily entail perceiving and processing raw data to extract valuable features and information. Conversely, *conception* tasks involve processing and comprehending abstract concepts and higher-level information.

**Chart Type Recognition** (CR, Fig. 2a) task aims to evaluate the MLLMs' capability to identify chart types accurately. Determining the chart type is the simplest but most basic step in the chain of thought, which determines the steps and logic to analyze the chart elements. The model is required to choose the correct candidate chart categories from both positive and negative views.

**Value Extraction** (VE, Fig. 2b) task aims to assess whether MLLMs can correctly extract the relevant values when confronted with complex visual logic. Without annotated data, MLLMs are required to rely on legends, axes, and corresponding graphical elements to provide answers. If the extracted numbers are not accurate, the analysis or summary of the MLLM will be incredible.

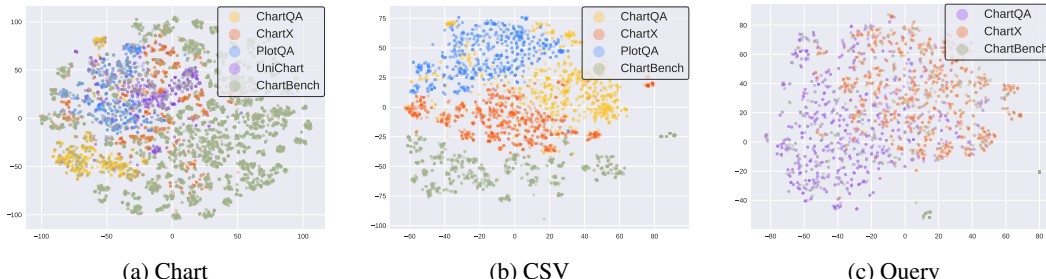

|              (a) Chart              |              (b) CSV              |              (c) Query              |

Figure 3: t-SNE Van der Maaten & Hinton (2008) visualisation of CLIP encoding features Radford et al. (2021). ChartBench (a) covers extensive distribution of charts, particularly with the unannotated chart; (b) stands apart from other datasets in terms of both topic and table data; (c) maintains consistent query manners with other datasets.

**Value Comparison** (VC, Fig. 2c) assesses MLLMs' visual reasoning by relying solely on visual-grounded elements to determine comparison answers instead of meta table data. MLLMs are not required to identify all chart metadata or element layouts. Instead, accurately observing graphic elements and identifying key components is sufficient for drawing correct conclusions.

**Global Conception** (GC, Fig. 2d) task assesses the ability to perceive global indicators, such as maximum values, from a holistic standpoint. This task requires that the model correctly parse all the information expressed in the charts.

**Number QA** (NQA, Fig. 2e). Considering the excessive number of negative samples in the VE task, we employ a tolerance evaluation method similar to ChartQA. Values within a specific error range are deemed correct. This step requires assistance from LLMs to format the responses from MLLMs with weak instruction-following ability.

## 3.3 DATASET ANALYSIS

Considering that ChartBench is primarily based on code rendering and website tool drawing, we conduct in-depth data analysis with other chart datasets, including real-world ones. Fig. 3 illustrates the distribution of chart, meta CSV, and query data, respectively. We randomly sample 10,000 data points respectively and extract corresponding features via CLIP (ViT-B/16) encoder. We adopt t-SNE Van der Maaten & Hinton (2008) for feature dimension reduction for visualizations.

**Chart Distribution**. As shown in Fig. 3a, ChartBench encompasses the primary range of charts from previous benchmarks and exhibits similar distribution trends to ChartX Xia et al. (2024). ChartBench incorporates a wider variety of plot styles (e.g., *classic, solarize, mpl, bmh, seaborn, ggplot*, etc.) to enhance stylistic diversity. ChartQA distinctly sets it apart from other datasets for real-world charts. However, our ChartBench maintains the same data distribution by drawing charts from real websites.

**CSV Distribution**. As shown in Fig. 3b, the CSVs of each dataset exhibit different distributions, indicating significant variations in table information. Considering the text truncation length of the CLIP text encoder, this distribution also reflects the differences between the original data topics, as the leading data usually includes titles or labels for the *x* and *y* axes.

**Query Distribution**. As shown in Fig. 3c, the query style of ChartBench is generally consistent with ChartQA Masry et al. (2022) and ChartX Xia et al. (2024). Note that we only display the QA task features of each dataset. Since the queries in these datasets primarily focus on numerical aspects of chart elements, their feature distributions are relatively consistent. This consistency facilitates the comparison and analysis of model performance across different datasets.

## 3.4 EVALUATION METRICS

**From *Acc+* to *Acc++*.** As shown in Fig. 2, for a base query $Q_i$ on chart $c$, we expand $Q_i$ into correct ($Q_i^r$) and incorrect ($Q_i^w$) assertions using a given query prompt. ChartBench requires the MLLM $\mathcal{M}$ to determine the correctness of the queries, providing boolean outputs $A_i^r := \mathcal{M}(Q_i^r; c)$ and $A_i^w := \mathcal{M}(Q_i^w; c)$. Because of the concise outputs, we can use regular expression matching instead

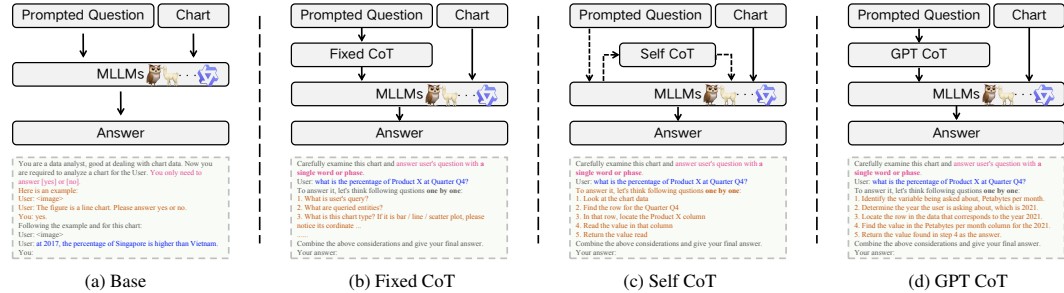

(a) Base      (b) Fixed CoT      (c) Self CoT      (d) GPT CoT

Figure 4: Illustration of different Chain of Thought. (a) No CoT. (b) All charts utilize the same CoT template that we provide. (c) The CoT for each chart is generated by its own LLM, given the prompted question. (d) GPT generates the CoT for each chart based on the prompted question.

of additional LLM judgement Fu et al. (2023b). In previous $Acc+$, $Q_i^w$ is typically formed by adding negation to $Q_i^r$, resulting in a significant semantic distance between them (completely opposite). Hence, a model is likely to produce different responses for $Q_i^w$ and $Q_i^r$. In $Acc++$, 1) $Q_i^r$ and $Q_i^w$ differ only in the ground truth value, resulting in similar token sequences. 2) $A_i^r$ and $A_i^w$ are derived from independent inferences. 3) The incorrect value in $Q_i^w$ is randomly selected from metadata to maintain rationality. We formally define the $Acc++$ metric as follows: Given $N$ base queries in ChartBench, $Acc++ = \frac{1}{N} \sum_{i=1}^{N} \mathbb{1} \left[ \mathcal{M}(Q_i^r; c) \wedge \neg \mathcal{M}(Q_i^w; c) \right]$, where $\wedge$, $\neg$ and $\mathbb{1}[x]$ are *and*, *not* and indicator function, respectively. The MLLM is considered to understand the query chart only if it accurately answers both $Q_i^r$ and $Q_i^w$ simultaneously.

**Confusion Rate (CoR).** During the evaluation, we find that many MLLMs produce the same output for both assertions, likely because they fail to utilize the chart information. To assess this failure, we introduce the *CoR* metric. Formally, $CoR = \frac{1}{N} \sum_{i=1}^{N} \mathbb{1} \left[ \mathcal{M}(Q_i^r; c) \oplus \neg \mathcal{M}(Q_i^w; c) \right]$, where $\oplus$ denotes the XOR operation. If an MLLM fails to use the visual information from charts, it tends to generate identical answers, resulting in *CoR* approaching 100% and $Acc++$ approaching 0%.

**GPT-acc.** While $Acc++$ is an efficient way to evaluate model responses, it falls short for specific numerical questions, as correctly answering a negative sample doesn't fully demonstrate the model's generalization ability and differs from methods used in datasets like ChartQA. To address this, we propose an improved error margin evaluation (5%) from ChartQA Masry et al. (2022). Our improvements include: 1) using LLMs Radford et al. (2019); Bai et al. (2023a); Bi et al. (2024) to filter responses and extract numerical answers, avoiding pattern-matching errors due to extraneous text, and 2) restricting NQA task questions to exclude elements like years and months, which could make the error margin too lenient and the evaluation meaningless.

## 4 BASELINES

ChartBench primarily evaluates MLLMs' ability to understand unannotated charts. We propose two simple yet effective baselines that significantly improve MLLMs' performance.

**ChartCoT.** As shown in Fig. 4, we propose effective baselines based on Chain of Thought Wei et al. (2022) to enhance the visual reasoning capability without model tuning. As shown in Fig. 4b, we design a series of questions that decompose user inquiries and employ prompts to mimic human visual reasoning for chart recognition. Additionally, we enable MLLMs to generate their own CoT (Fig. 4c) or seek assistance from stronger LLMs to generate CoTs (Fig. 4d). This approach significantly aids MLLMs in understanding charts, particularly in cases where visual logic is more complicated.

**Supervised Fine-tuning.** We conduct a two-stage supervised fine-tuning (SFT) based on Qwen-VL-Chat and Internlm-XComposer-v2. In the first stage, we perform alignment training with chart and CSV pairs to update the connector parameters. In the second stage, we utilize instruction and chart pairs to fine-tune the LLM branch with LoRA Hu et al. (2021). Considering that charts are not complex images compared to natural images, we keep the visual encoder frozen during the SFT process. Please refer to Appendix D for detailed experimental settings.

Table 3: The zero-shot performance on ChartQA and our proposed ChartBench. We report average *Acc++* for 4 yes-or-no tasks and GPT-acc for NQA task. Regular: line, pie, and bar plots. Extra: additional chart in Tab. 2. ChartBench is more challenging for more unannotated charts.

| Models | ChartBench | | | | | | | | ChartQA | | | |
| | Regular Type | | | Extra Type | | | Avg. | Rank | Human | Aug. | Avg. | Rank |
| | *Acc++* | NQA | Avg. | *Acc++* | NQA | Avg. | | | | | | |
| *Open source MLLMs* | | | | | | | | | | | | |
| *General Purpose Models* | | | | | | | | | | | | |
| VisualGLM Du et al. (2022) | 3.46 | 1.83 | 3.13 | 4.22 | 4.84 | 4.35 | 3.68 | #21 | 18.96 | 6.80 | 12.88 | #15 |
| Shikra Chen et al. (2023b) | 8.59 | 2.35 | 7.34 | 7.50 | 9.05 | 7.81 | 7.55 | #20 | 16.24 | 7.28 | 11.76 | #18 |
| InstructBLIP Dai et al. (2023) | 17.96 | 0.87 | 14.55 | 5.50 | 5.37 | 5.47 | 10.43 | #18 | 15.92 | 7.92 | 11.92 | #17 |
| Internlm-XComposer Zhang et al. (2023) | 19.70 | 1.22 | 16.01 | 10.11 | 5.79 | 9.25 | 12.94 | #16 | 13.20 | 7.84 | 10.52 | #19 |
| CogVLM-Chat Wang et al. (2023b) | 14.41 | 12.96 | 14.12 | 11.89 | 13.68 | 12.25 | 13.26 | #15 | 34.24 | 28.56 | 31.40 | #12 |
| SPHINX Lin et al. (2023) | 17.87 | 6.17 | 15.54 | 17.92 | 12.74 | 16.89 | 16.13 | #14 | 21.44 | 11.20 | 16.32 | #14 |
| BLIP2 Li et al. (2023c) | 21.65 | 0.96 | 17.53 | 18.44 | 4.84 | 15.74 | 16.70 | #13 | 13.52 | 6.00 | 9.76 | #20 |
| DeepSeek-VL-Chat Lu et al. (2024) | 15.68 | 20.00 | 16.54 | 18.51 | 29.73 | 20.74 | 18.42 | #12 | 44.88 | 76.56 | 60.72 | #9 |
| MiniGPT-v2 Chen et al. (2023a) | 22.37 | 2.43 | 18.40 | 25.06 | 5.26 | 21.11 | 19.61 | #10 | 15.60 | 8.48 | 12.04 | #16 |
| LLaVA-v1.5 Liu et al. (2023e) | 25.61 | 8.09 | 22.12 | 27.39 | 15.26 | 24.97 | 23.39 | #7 | 22.64 | 13.04 | 17.84 | #13 |
| Qwen-VL-Chat Bai et al. (2023b) | 29.46 | 23.57 | 28.28 | 26.56 | 21.05 | 25.46 | 26.98 | #6 | 42.48 | 75.20 | 58.84 | #10 |
| Mini-Gemini Li et al. (2024) | 39.57 | 25.57 | 36.78 | 31.81 | 25.79 | 30.61 | 33.96 | #4 | 44.32 | 57.04 | 50.68 | #11 |
| InternVL2 Chen et al. (2024b) | 40.91 | **50.00** | 42.72 | 36.12 | 47.59 | 38.40 | 40.73 | #3 | - | - | **83.30** | #1 |
| Internlm-XComposer-v2 Dong et al. (2024) | 57.89 | 40.96 | 54.52 | 41.75 | 31.58 | 39.73 | 47.78 | #2 | 63.12 | 81.92 | 72.64 | #4 |
| Qwen2-VL Wang et al. (2024) | **60.45** | **50.00** | **58.37** | **68.99** | **53.30** | **65.87** | **61.70** | #1 | - | - | 83.00 | #2 |
| *OCR Optimized Models* | | | | | | | | | | | | |
| CogAgent Hong et al. (2023) | 20.39 | 26.61 | 21.63 | 14.36 | 25.79 | 16.64 | 19.35 | #11 | 54.08 | 80.56 | 67.32 | #6 |
| mPLUG-Owl-bloomz Ye et al. (2023b) | 27.80 | 2.35 | 22.73 | 25.47 | 6.21 | 21.64 | 22.21 | #8 | 7.84 | 4.88 | 6.36 | #21 |
| DocOwl-v1.5 Hu et al. (2024) | 35.27 | 37.30 | 35.67 | 26.86 | 29.47 | 27.38 | 31.89 | #5 | 48.24 | 86.72 | 67.48 | #5 |
| *Chart Optimized Models* | | | | | | | | | | | | |
| OneChart Chen et al. (2024a) | 12.34 | 2.26 | 10.33 | 8.75 | 3.37 | 7.68 | 9.12 | #19 | **85.30** | 49.10 | 67.20 | #7 |
| ChartVLM Xia et al. (2024) | 8.02 | 43.74 | 15.24 | 5.92 | 18.21 | 8.37 | 12.06 | #17 | 42.08 | 82.48 | 62.28 | #8 |
| ChartLlama Han et al. (2023) | 22.02 | 16.87 | 21.00 | 22.56 | 18.32 | 21.71 | 21.30 | #9 | 58.40 | **93.12** | 75.76 | #3 |
| *Closed source MLLMs* | | | | | | | | | | | | |
| ERNIE BaiDu | 47.39 | 25.74 | 43.08 | 46.39 | 33.37 | 43.82 | 43.37 | #3 | - | - | - | |
| GPT-4V OpenAI (2023) | 53.26 | 33.04 | 49.23 | 55.83 | 40.00 | 52.68 | 50.74 | #2 | - | - | 78.50 | #2 |
| GPT-4O OpenAI (2023) | **65.00** | 40.00 | **60.02** | 63.33 | **41.05** | 58.89 | **59.45** | #1 | - | - | **85.70** | #1 |

## 5 EXPERIMENTS

We evaluate 18 open-sourced and 3 closed-sourced MLLMs (shown in Tab. 3) on ChartBench. Detailed model architectures and configurations are provided in Appendix C.1. Notably, some models exhibited poor performance in certain areas, which may be due to suboptimal instruction prompts. We provide a detailed analysis of the model with this anomaly in Appendix C.2.

**Results on ChartBench.** Tab. 3 compares various MLLMs on the ChartQA and our ChartBench. Overall, MLLMs show consistent trends across both benchmarks, though individual models vary notably. Onechart Chen et al. (2024a) performs well on ChartQA but struggles with ChartBench, extracting incomplete or overly long Python dictionaries, which hampers its LLM (llava-V1.6 Liu et al. (2023e)) from following instructions effectively. Qwen Bai et al. (2023b) and other top-ranked MLLMs demonstrate consistent performance across both metrics, indicating accurate chart comprehension. However, models like BLIP2 and MiniGPT-v2 show significant deviations due to the broader and less standardized output required by NQA compared to *Acc++*, leading to many extraction failures despite filtering by stronger LLMs OpenAI (2023); Bi et al. (2024); Bai et al. (2023a). Unsurprisingly, models generally perform better on regular charts than on extra types, especially those with pre-alignment, such as ChartVLM Xia et al. (2024), DocOwl Hu et al. (2024), and Internlm-XComposer-v2 Dong et al. (2024), since the alignment process primarily uses regular charts. This indicates that pre-alignment and SFT with chart data effectively enhance chart comprehension abilities.

**Results w.r.t. Task Types.** Tab. 4 presents the performance of MLLMs on 5 type tasks, which are introduced in Sec. 3.2. All MLLMs perform exceptionally well on the easiest CR task, demonstrating their ability to recognize basic chart types effectively. LLaVA-v1.5 Liu et al. (2023e), mPLUG-Owl Ye et al. (2023b), and Qwen-VL-Chat Bai et al. (2023b) demonstrate significant advantages in the VC and GC conception tasks, benefiting from their chart-tuning data. VE is the most challenging task, which is the key distinction between ChartBench and ChartQA. VE task cannot be resolved merely through basic OCR and demands a series of visual and textual logical reasoning steps to reach the ultimate answer. Despite demonstrating strong overall performance, models like BLIP2 Li et al. (2023c) and ChartLlama Han et al. (2023) struggle with the VE task. This observation suggests that strong text recognition abilities are insufficient for high chart reasoning capabilities. Closed-source

Table 4: The zero-shot performance w.r.t. task types, i.e., Chart Recognition (CR), Value Extraction (VE), Value Comparison (VC), Global Conception (GC), and Number QA (NQA). ↑ / ↓ indicates that higher/lower is the better, respectively.

| Models | CR | | VE | | VC | | GC | | NQA↑ | Avg.↑ |
|---|---|---|---|---|---|---|---|---|---|---|
| | Acc++↑ | CoR↓ | Acc++↑ | CoR↓ | Acc++↑ | CoR↓ | Acc++↑ | CoR↓ | | |
| *Open source MLLMs* | | | | | | | | | | |
| *General Purpose Models* | | | | | | | | | | |
| VisualGLM Du et al. (2022) | 16.29 | 79.19 | 0.00 | 99.67 | 0.00 | 99.81 | 0.00 | 99.71 | 3.19 | 3.68 |
| Shikra Chen et al. (2023b) | 2.10 | 93.57 | 11.90 | 80.71 | 10.62 | 87.71 | 7.86 | 82.71 | 5.38 | 7.55 |
| InstructBLIP Dai et al. (2023) | 49.57 | 36.67 | 0.00 | 100.00 | 0.05 | 99.81 | 0.00 | 99.90 | 2.90 | 10.43 |
| Internlm-XComposer Zhang et al. (2023) | 42.29 | 56.95 | 6.86 | 85.14 | 2.48 | 96.57 | 9.67 | 78.48 | 3.29 | 12.94 |
| CogVLM-Chat Wang et al. (2023b) | 29.14 | 69.33 | 2.81 | 94.29 | 14.19 | 78.86 | 7.33 | 90.14 | 13.29 | 13.26 |
| SPHINX Lin et al. (2023) | 38.48 | 51.38 | 10.38 | 80.67 | 14.33 | 77.38 | 9.62 | 80.90 | 9.14 | 16.13 |
| BLIP2 Li et al. (2023c) | 60.05 | 37.05 | 4.24 | 89.29 | 14.05 | 78.86 | 3.86 | 90.00 | 2.71 | 16.70 |
| DeepSeek-VL-Chat Lu et al. (2024) | 51.43 | 58.10 | 3.81 | 95.24 | 5.24 | 92.38 | 4.29 | 95.24 | 22.86 | 18.42 |
| MiniGPT-v2 Chen et al. (2023a) | 29.05 | 49.24 | 22.00 | **55.14** | 24.29 | 53.33 | 18.10 | 61.76 | 3.71 | 19.35 |
| LLaVA-v1.5 Liu et al. (2023e) | 47.86 | 36.24 | 15.81 | 66.24 | 26.05 | 56.48 | 16.52 | 66.57 | 11.33 | 23.39 |
| Qwen-VL-Chat Bai et al. (2023b) | 51.67 | 42.71 | 11.14 | 84.57 | 27.29 | 63.14 | 21.71 | 74.86 | 22.43 | 26.98 |
| Mini-Gemini Li et al. (2024) | 80.52 | 17.86 | 17.62 | 70.43 | 26.00 | 59.38 | 22.00 | 71.10 | 25.67 | 33.96 |
| InternVL2 Chen et al. (2024b) | 48.60 | 42.99 | 29.44 | 56.54 | 35.68 | 49.30 | 30.19 | 56.60 | 42.45 | 40.73 |
| Internlm-XComposer-v2 Dong et al. (2024) | 68.29 | 30.24 | 36.63 | 57.71 | 54.63 | 27.71 | 45.80 | 51.46 | 36.71 | 47.78 |
| Qwen2-VL Wang et al. (2024) | **81.17** | **10.31** | **43.05** | 55.16 | **66.67** | **15.32** | **55.86** | **40.54** | **47.75** | **61.70** |
| *OCR Optimized Models* | | | | | | | | | | |
| CogAgent Hong et al. (2023) | 62.57 | 37.10 | 1.19 | 94.90 | 7.33 | 88.24 | 1.19 | 94.76 | 26.24 | 19.61 |
| mPLUG-Owl-bloomz Ye et al. (2023b) | 32.33 | 51.24 | 23.14 | 76.76 | 25.33 | 69.29 | 26.48 | 71.00 | 4.10 | 22.21 |
| DocOwl-v1.5 Hu et al. (2024) | 30.43 | 65.05 | 34.48 | 58.24 | 31.10 | 55.19 | 30.48 | 63.19 | 33.76 | 31.89 |
| *Chart Optimized Models* | | | | | | | | | | |
| OneChart Chen et al. (2024a) | 3.71 | 94.33 | 15.48 | 82.14 | 17.57 | 73.71 | 11.38 | 85.67 | 2.76 | 9.12 |
| ChartVLM Xia et al. (2024) | 0.00 | 100.00 | 9.05 | 85.48 | 10.05 | 83.81 | 8.52 | 86.19 | 32.19 | 12.06 |
| ChartLlama Han et al. (2023) | 49.86 | 44.19 | 8.38 | 84.14 | 20.43 | 69.48 | 10.67 | 83.81 | 17.52 | 21.30 |
| *Closed source MLLMs* | | | | | | | | | | |
| ERNIE BaiDu | 65.24 | 19.52 | **44.76** | **44.76** | 32.86 | 41.43 | 47.14 | 47.62 | 29.24 | 43.37 |
| GPT-4V OpenAI (2023) | 96.19 | 2.86 | 30.95 | 63.33 | 48.57 | 34.76 | 46.19 | 47.62 | 36.19 | 50.74 |
| GPT-4O OpenAI (2023) | **97.62** | **1.43** | 43.33 | **44.76** | 66.19 | 16.19 | 53.33 | 41.43 | 40.48 | 59.45 |

Table 5: The zero-shot *CoR* (%) performance w.r.t. chart types. Higher *CoR* means more severe hallucinations. *CoR* and *Acc++* exhibit a negative correlation.

| Models | Regular Type | | | | Extra Type | | | | | | | CoR |
|---|---|---|---|---|---|---|---|---|---|---|---|---|
| | Line | Bar | Pie | Avg. | Area | Box | Radar | Scatter | Node | Combin. | Avg. | |
| *Open source MLLMs* | | | | | | | | | | | | |
| *General Purpose Models* | | | | | | | | | | | | |
| VisualGLM Du et al. (2022) | 89.20 | 98.04 | 99.38 | 96.27 | 93.50 | 90.50 | 97.50 | 91.33 | 80.50 | 94.62 | 92.39 | 94.60 |
| Shikra Chen et al. (2023b) | 85.80 | 82.19 | 98.25 | 85.93 | 84.83 | 85.00 | 86.00 | 84.33 | 72.00 | 95.38 | 85.89 | 86.18 |
| InstructBLIP Dai et al. (2023) | 75.50 | 82.58 | 79.50 | 80.41 | 88.33 | 85.50 | 91.00 | 86.00 | 90.50 | 89.62 | 88.58 | 84.10 |
| CogVLM-Chat Wang et al. (2023b) | 87.20 | 83.38 | 79.38 | 83.52 | 85.33 | 86.67 | 77.88 | 84.17 | 79.50 | 89.88 | 84.13 | 83.62 |
| DeepSeek-VL-Chat Lu et al. (2024) | 73.00 | 88.46 | 83.75 | 84.09 | 88.33 | 86.67 | 66.25 | 86.67 | 72.50 | 90.00 | 79.73 | 82.74 |
| Internlm-XComposer Zhang et al. (2023) | 79.40 | 73.92 | 68.62 | 74.20 | 93.33 | 79.83 | 77.00 | 84.17 | 91.00 | 92.25 | 85.84 | 79.29 |
| BLIP2 Li et al. (2023c) | 66.40 | 79.96 | 72.75 | 75.57 | 92.50 | 85.83 | 78.12 | 73.17 | 16.00 | 66.88 | 71.92 | 73.80 |
| SPHINX Lin et al. (2023) | 73.80 | 75.73 | 58.00 | 72.07 | 82.00 | 86.17 | 71.00 | 73.17 | 63.50 | 65.25 | 73.47 | 72.58 |
| Qwen-VL-Chat Bai et al. (2023b) | 56.00 | 73.62 | 57.50 | 66.08 | 68.67 | 66.67 | 57.25 | 74.50 | 74.00 | 66.25 | 66.92 | 66.32 |
| LLaVA-v1.5 Liu et al. (2023e) | 51.20 | 59.69 | 54.87 | 56.89 | 61.67 | 58.50 | 60.00 | 59.17 | 29.00 | 56.00 | 55.79 | 56.38 |
| MiniGPT-v2 Chen et al. (2023a) | 52.20 | 57.35 | 56.75 | 56.07 | 57.17 | 56.00 | 52.75 | 51.50 | 47.00 | 54.25 | 53.47 | 54.87 |
| Mini-Gemini Li et al. (2024) | 55.70 | 53.92 | 51.25 | 53.84 | 53.50 | 62.67 | 57.75 | 50.83 | 61.00 | 57.88 | 56.92 | 54.69 |
| InternVL2 Chen et al. (2024b) | 49.00 | 42.31 | 52.50 | 45.68 | 56.16 | 70.00 | 58.75 | 58.33 | 40.00 | 60.00 | 53.25 | 51.35 |
| Internlm-XComposer-v2 Dong et al. (2024) | 27.40 | 44.65 | 32.50 | 38.52 | 55.33 | 58.33 | 47.88 | 43.17 | 29.00 | 39.75 | 47.22 | 41.78 |
| Qwen2-VL Wang et al. (2024) | 25.00 | 34.62 | 23.75 | 30.45 | 33.64 | 55.00 | 35.00 | 11.67 | 10.00 | 25.00 | 25.57 | **30.34** |
| *OCR Optimized Models* | | | | | | | | | | | | |
| CogAgent Hong et al. (2023) | 81.40 | 76.00 | 89.00 | 79.59 | 84.33 | 90.12 | 87.50 | **7.00** | | 84.00 | 81.50 | 78.75 |
| mPLUG-Owl-bloomz Ye et al. (2023b) | 69.20 | 79.54 | 76.12 | 76.57 | 82.50 | 78.50 | 80.00 | 77.83 | 70.00 | 77.50 | 78.24 | 77.35 |
| DocOwl-v1.5 Hu et al. (2024) | 47.10 | 63.69 | 63.62 | 59.91 | 80.50 | 61.33 | 64.62 | 59.67 | 53.00 | 52.00 | 62.44 | 60.42 |
| *Chart Optimized Models* | | | | | | | | | | | | |
| ChartVLM Xia et al. (2024) | 85.80 | 87.46 | 92.25 | 87.95 | 88.00 | 90.33 | 89.88 | 91.17 | 91.00 | 89.50 | 89.83 | 88.87 |
| OneChart Chen et al. (2024a) | 80.10 | 84.46 | 89.38 | 84.36 | 89.83 | 87.33 | 93.62 | 90.17 | 33.50 | 89.38 | 87.08 | 83.96 |
| ChartLlama Han et al. (2023) | 65.60 | 74.27 | 74.50 | 72.34 | 81.50 | 78.83 | 72.62 | 66.00 | 28.50 | 68.62 | 68.47 | 70.40 |
| *Closed source MLLMs* | | | | | | | | | | | | |
| ERNIE BaiDu | 34.00 | 41.15 | 27.50 | 37.05 | **46.67** | 45.00 | 51.25 | 33.33 | 25.00 | 33.75 | 40.26 | 38.33 |
| GPT-4V OpenAI (2023) | 21.00 | 52.69 | 37.50 | 42.73 | 58.33 | 38.33 | 23.75 | 25.00 | **0.00** | 33.75 | 31.32 | 37.14 |
| GPT-4O OpenAI (2023) | **9.00** | **37.31** | **20.00** | **27.73** | 50.00 | **28.33** | **20.00** | 16.67 | **0.00** | 28.75 | **25.26** | **25.95** |

models outperform open-source models, partly due to their larger size and broader data coverage. Additionally, they utilize supplementary recognition tools instead of relying solely on end-to-end inference, as further detailed in Appendix E.6.

**Error Analysis.** Tab. 5 presents the results on *CoR*, which reflects the MLLM's failure to utilize chart information. We find that existing MLLMs tend to give identical answers to similar questions about charts. Internlm-XComposer-v2 Dong et al. (2024) shows the lowest CoR (41.78%), which means nearly half of the responses fail to distinguish between positive and negative questions. This indicates that random guessing without the chart is common among open-source models due to their inability to utilize chart information. *CoR* generally shows a negative correlation with performance, although there are exceptions. Qwen Bai et al. (2023b) demonstrates better *Acc++* compared to MiniGPT-v2 Chen et al. (2023a) with higher *CoR*. For closed-source MLLMs, although GPT-4V OpenAI

Table 6: The performance on with and without annotation charts. *w/i* and *w/o* indicate with and without annotation, respectively. †: *Acc++*. ‡: GPT-acc. MLLMs are better with annotated charts.

| Models | CR† | | VE† | | VC† | | GC† | | NQA‡ | | Avg. | | |
|---|---|---|---|---|---|---|---|---|---|---|---|---|---|
| | w/i | w/o | w/i | w/o | w/i | w/o | w/i | w/o | w/i | w/o | w/i | w/o | Δ |
| *Open source MLLMs* | | | | | | | | | | | | | |
| *General Purpose Models* | | | | | | | | | | | | | |
| Internlm-XComposer Zhang et al. (2023) | 30.50 | 53.00 | 8.00 | 7.50 | 1.00 | 2.75 | 10.00 | 8.50 | 10.60 | 1.75 | 12.02 | 14.70 | -2.68 |
| Shikra Chen et al. (2023b) | 1.25 | 1.00 | 6.00 | 10.25 | 2.25 | 6.50 | 5.00 | 8.50 | 15.80 | 1.50 | 6.06 | 5.55 | +0.51 |
| MiniGPT-v2 Chen et al. (2023a) | 31.25 | 31.00 | 24.50 | 22.25 | 27.25 | 26.50 | 16.50 | 19.75 | 7.80 | 2.75 | 21.46 | 20.45 | +1.01 |
| InstructBLIP Dai et al. (2023) | 59.50 | 54.75 | 0.00 | 0.00 | 0.25 | 0.00 | 0.00 | 0.00 | 10.40 | 1.00 | 14.03 | 11.15 | +2.88 |
| BLIP2 Li et al. (2023c) | 78.25 | 69.00 | 4.00 | 5.00 | 26.50 | 21.50 | 5.00 | 7.50 | 6.80 | 1.75 | 24.11 | 20.95 | +3.16 |
| VisualGLM Du et al. (2022) | 24.75 | 16.25 | 0.00 | 0.00 | 0.00 | 0.00 | 0.00 | 0.00 | 9.20 | 0.75 | 6.79 | 3.40 | +3.39 |
| SPHINX Lin et al. (2023) | 43.75 | 41.00 | 11.75 | 12.25 | 18.50 | 17.00 | 15.00 | 8.75 | 23.00 | 5.25 | 22.40 | 16.85 | +5.55 |
| LLaVA-v1.5 Liu et al. (2023e) | 55.25 | 43.50 | 17.75 | 16.00 | 28.50 | 31.50 | 15.50 | 16.25 | 31.80 | 5.50 | 29.76 | 22.55 | +7.21 |
| CogVLM-Chat Wang et al. (2023b) | 31.25 | 27.00 | 3.50 | 2.00 | 22.75 | 19.25 | 14.00 | 9.00 | 37.40 | 5.75 | 21.78 | 12.60 | +9.18 |
| Mini-Gemini Li et al. (2024) | 79.33 | 74.00 | 20.00 | 16.75 | 32.89 | 33.75 | 30.89 | 22.00 | 59.20 | 14.75 | 44.46 | 32.25 | +12.21 |
| DeepSeek-VL-Chat Lu et al. (2024) | 68.00 | 46.25 | 10.00 | 1.88 | 18.00 | 1.25 | 10.00 | 2.50 | 48.00 | 15.00 | 26.50 | 13.25 | +13.25 |
| InternVL2 Chen et al. (2024b) | 56.00 | 46.34 | 46.00 | 24.39 | 52.00 | 30.67 | 38.00 | 27.78 | 76.00 | 34.65 | 48.00 | 33.46 | +14.54 |
| Qwen-VL-Chat Bai et al. (2023b) | 68.00 | 53.50 | 26.50 | 7.50 | 47.75 | 35.00 | 31.50 | 33.50 | 54.80 | 14.00 | 45.71 | 28.70 | +17.01 |
| Internlm-XComposer-v2 Dong et al. (2024) | 83.00 | 64.25 | **75.25** | 39.75 | 70.00 | **66.00** | 67.75 | **66.25** | 69.80 | 37.75 | 73.16 | 54.80 | +18.36 |
| Qwen2-VL Wang et al. (2024) | **96.00** | 76.88 | 74.00 | 34.10 | **76.00** | 63.95 | **76.00** | 50.00 | **70.00** | 43.30 | **80.50** | **59.06** | **+21.44** |
| *OCR Optimized Models* | | | | | | | | | | | | | |
| mPLUG-Owl-bloomz Ye et al. (2023b) | 37.50 | 41.50 | 22.50 | 27.50 | 27.25 | 30.25 | 27.50 | 29.25 | 9.40 | 3.75 | 24.83 | 26.45 | -1.62 |
| DocOwl-v1.5 Hu et al. (2024) | 47.11 | 49.25 | 60.89 | **42.25** | 43.11 | 41.50 | 38.22 | 43.75 | 61.60 | 40.75 | 50.19 | 43.50 | +6.69 |
| CogAgent Hong et al. (2023) | 64.67 | 64.75 | 2.89 | 0.00 | 16.00 | 13.25 | 2.44 | 0.25 | 61.60 | 11.50 | 29.52 | 17.95 | +11.57 |
| *Chart Optimized Models* | | | | | | | | | | | | | |
| ChartVLM Xia et al. (2024) | 0.00 | 0.00 | 12.22 | 16.00 | 9.33 | 11.00 | 12.44 | 10.25 | 57.00 | **46.50** | 18.20 | 15.55 | +2.65 |
| OneChart Chen et al. (2024a) | 4.00 | 3.50 | 36.67 | 14.50 | 21.78 | 16.00 | 25.11 | 9.25 | 4.40 | 2.25 | 18.39 | 9.10 | +9.29 |
| ChartLlama Han et al. (2023) | 57.00 | 53.50 | 15.75 | 7.00 | 33.00 | 24.25 | 20.00 | 13.00 | 42.20 | 12.75 | 33.59 | 22.10 | +11.49 |
| *Closed source MLLMs* | | | | | | | | | | | | | |
| ERNIE BaiDu | 67.50 | 72.50 | 32.50 | **45.00** | 42.50 | 37.50 | 52.50 | 52.50 | 52.20 | 7.25 | 49.44 | 42.95 | +6.49 |
| GPT-4O OpenAI (2023) | **95.00** | 95.00 | **87.50** | 37.50 | **72.50** | 80.00 | **87.50** | 60.00 | 74.00 | **32.50** | **83.30** | **61.00** | +22.30 |
| GPT-4V OpenAI (2023) | 92.50 | **97.50** | 72.50 | 7.50 | 67.50 | 57.50 | 72.50 | 37.50 | **82.00** | 15.00 | 77.40 | 43.00 | **+34.40** |

Table 7: Performance gain of chart chain of thought on various MLLMs. CoTs have proven to be simple and effective ways to improve ChartBench's performance. †: *Acc++*. ‡: GPT-acc.

| Models | Method | w/i | w/o | Δ | CR† | VE† | VC† | GC† | NQA‡ | Avg. |
|---|---|---|---|---|---|---|---|---|---|---|
| MiniGPT-v2 | Base | 21.46 | 20.45 | 1.01 | 29.02 | 22.29 | 24.59 | 18.29 | 3.71 | 19.58 |
| | CoT-fix | 25.25$_{+3.79}$ | 21.33$_{+0.88}$ | 3.92$_{+2.91}$ | 36.76$_{+7.74}$ | 29.22$_{+6.93}$ | 25.14$_{+0.55}$ | 26.37$_{+8.08}$ | 5.20$_{+1.49}$ | 24.54$_{+4.96}$ |
| | CoT-self | 22.44$_{+0.98}$ | 20.12$_{-0.33}$ | 2.32$_{+1.31}$ | 34.52$_{+5.50}$ | 27.83$_{+5.54}$ | 26.02$_{+1.43}$ | 24.44$_{+6.15}$ | 4.40$_{+0.69}$ | 23.44$_{+3.86}$ |
| | CoT-GPT | 26.66$_{+5.20}$ | 21.52$_{+1.07}$ | 5.14$_{+4.13}$ | 37.72$_{+8.70}$ | 29.31$_{+7.02}$ | 26.66$_{+2.07}$ | 27.62$_{+9.33}$ | 5.55$_{+1.84}$ | 25.37$_{+5.79}$ |
| Qwen-VL-Chat | Base | 45.71 | 28.70 | 17.01 | 52.54 | 10.78 | 27.46 | 21.95 | 22.43 | 27.03 |
| | CoT-fix | 50.12$_{+4.42}$ | 29.80$_{+1.10}$ | 20.32$_{+3.31}$ | 64.54$_{+12.00}$ | 15.85$_{+5.07}$ | 28.44$_{+0.98}$ | 29.22$_{+7.27}$ | 24.98$_{+2.55}$ | 32.61$_{+5.58}$ |
| | CoT-self | 47.77$_{+2.07}$ | 26.74$_{-1.96}$ | 21.03$_{+4.02}$ | 56.52$_{+3.98}$ | 11.24$_{+0.46}$ | 26.42$_{-1.04}$ | 24.33$_{+2.38}$ | 22.64$_{+0.21}$ | 28.23$_{+1.20}$ |
| | CoT-GPT | 51.22$_{+5.52}$ | 30.02$_{+1.32}$ | 21.20$_{+4.19}$ | 66.64$_{+14.10}$ | 16.02$_{+5.24}$ | 29.33$_{+1.87}$ | 28.82$_{+6.87}$ | 26.72$_{+4.29}$ | 33.51$_{+6.48}$ |
| Internlm-XComposer-v2 | Base | 73.16 | 54.80 | 18.36 | 68.29 | 36.63 | 54.63 | 45.80 | 36.71 | 48.41 |
| | CoT-fix | 75.22$_{+2.06}$ | 55.74$_{+0.94}$ | 19.48$_{+1.12}$ | 69.22$_{+0.93}$ | 36.76$_{+0.13}$ | 58.23$_{+3.60}$ | 46.11$_{+0.31}$ | 36.52$_{-0.19}$ | 49.37$_{+0.96}$ |
| | CoT-self | 73.54$_{+0.38}$ | 54.62$_{-0.18}$ | 18.92$_{+0.56}$ | 69.92$_{+1.63}$ | 35.32$_{-1.31}$ | 55.21$_{+0.58}$ | 46.02$_{+0.22}$ | 36.32$_{-0.39}$ | 48.56$_{+0.15}$ |
| | CoT-GPT | 76.23$_{+3.07}$ | 55.12$_{+0.32}$ | 21.11$_{+2.75}$ | 70.92$_{+2.63}$ | 37.33$_{+0.70}$ | 58.82$_{+4.19}$ | 47.46$_{+1.66}$ | 37.22$_{+0.51}$ | 50.35$_{+1.94}$ |

(2023) outperforms ERNIE BaiDu in terms of *Acc++*, their *CoR* are similar. More granular analysis reveals that ERNIE performs better on challenging VE tasks, which is the weakest area for GPT-4V.

**Results w.r.t. Data-point Annotations.** Tab. 6 presents the MLLMs' performance on annotated and unannotated charts. We report only the comparison results *between the w/i and w/o chart versions from the same table* to ensure fair comparisons. Almost all models perform better on annotated charts. As MLLM capabilities increase, the performance gap between annotated and unannotated charts widens significantly, such as Internlm-XComposer-v2 (+18.36%) and GPT-4V (+34.40%). This is because OCR on annotated charts is an easier task for advanced MLLMs, while their performance on unannotated charts is limited. To further enhance MLLM capabilities, more unannotated charts are needed to highlight the importance of our ChartBench.

**CoT Performance.** Tab. 7 shows the performance of the CoT-based baseline, which generally improves performance without parameter updates. Because many models encounter difficulties in following instructions, we show the results on MiniGPT-v2, Qwen-VL-Chat, and Internlm-XComposer-v2. The fixed prompt ameliorates all tasks, especially for weaker models like MiniGPT-v2 and Qwen-VL-Chat. CoT-self is less effective because the quality and length of the self-generated CoT are uncontrollable, which hinders models from following instructions. CoT-GPT ensures CoT quality and is customized for each question type and thus performs the best. See chain of thought examples in Fig. 4.

**SFT Performance.** Tab. 8 shows the performance of the SFT-based baseline. Each model undergoes 2 epochs of alignment and 1 epoch of SFT with a learning rate of $1e-5$. Due to the commonality of chart images, we freeze the visual encoder and update only the connector and LLM branch using LoRA Hu et al. (2021). We balance *NQA* and *Acc++* instructions to avoid predictive bias. The

Table 8: Performance gain of supervised fine-tuning on Qwen-VL-Chat and Internlm-XComposer-v2.

| Models | w/i | w/o | Δ | Regular | | | Extra | | | Avg. |
|---|---|---|---|---|---|---|---|---|---|---|
| | | | | Acc++ | NQA | Avg. | Acc++ | NQA | Avg. | |
| Qwen-VL-Chat | 45.71 | 28.70 | 17.01 | 29.46 | 23.57 | 28.28 | 26.56 | 21.05 | 25.46 | 26.98 |
| Qwen-VL-Chat+SFT | 60.00$_{+14.29}$ | 43.65$_{+14.95}$ | 16.35$_{-0.66}$ | 46.39$_{+16.93}$ | 25.65$_{+2.08}$ | 42.26$_{+13.98}$ | 40.18$_{+13.62}$ | 25.89$_{+4.84}$ | 37.33$_{+11.87}$ | 39.99$_{+13.01}$ |
| Internlm-XComposer-v2 | 73.16 | 54.80 | 18.36 | 57.89 | 40.96 | 54.52 | 41.75 | 31.58 | 39.73 | 47.78 |
| Internlm-XComposer-v2+SFT | 87.16$_{+14.00}$ | 68.20$_{+13.40}$ | 18.96$_{+0.60}$ | 72.66$_{+14.77}$ | 43.81$_{+2.85}$ | 66.91$_{+12.39}$ | 62.74$_{+21.00}$ | 45.37$_{+13.79}$ | 59.28$_{+19.55}$ | 63.40$_{+15.65}$ |

improvement in $Acc++$ is particularly notable. SFT significantly boosts performance on ChartBench (Qwen-VL-Chat +13.01%, Internlm-XComposer-v2 +15.62%) and shows gains on ChartQA as well. Notably, Internlm-XComposer-v2 is the best open-source model on ChartBench and achieves performance on par with the SOTA GPT-4o after alignment and SFT. Furthermore, the model does not lose its general visual recognition capabilities (Tab. 16) and even shows improved performance on other chart benchmarks (Tab. 17). This demonstrates the effectiveness of the ChartBench dataset.

## 6 DISCUSSION

**Instruction Following.** Some models encounter difficulties in following instructions. For instance, mPLUG Ye et al. (2023b) provides overly detailed responses to explain its decision. LLaVA-v1.6 has difficulty accurately understanding the instructions when the dictionaries extracted by OneChart Chen et al. (2024a) are too lengthy. Models like Shikra Chen et al. (2023b) often simply reiterate the original question. Meanwhile, models like CogVLM Wang et al. (2023b) produce hallucinatory responses unrelated to the query. Therefore, instruction design greatly impacts the performance of models because the same model can yield vastly different results with different prompt templates.

**MLLM Performance**. MLLMs exhibit several common deficiencies in chart comprehension. 1) Since MLLMs are typically trained on *images* and *descriptive statements*, they prioritize giving descriptive responses to charts over numbers. This is the opposite of human graph recognition, where specific elements are identified first, followed by the final answer. 2) Some MLLMs fail to effectively follow complex instructions, which hinders their application of intricate CoT strategies. 3) Data hallucinations that occurred in VE and NQA tasks show that the data extracted by models is not yet entirely reliable, leading to errors when answers involve specific numbers.

**CoT v.s. SFT.** Both CoT and SFT effectively improve MLLMs' capabilities, but their impacts vary. CoT shows greater improvement for weaker MLLMs (e.g., 6.48% for Qwen-VL-Chat v.s. 1.94% for Internlm-XComposer-v2 in Tab. 7). The main improvement of CoT comes from unannotated charts, and Qwen-VL-Chat benefits more than Internlm-XComposer-v2. As a result, CoT provides limited improvement for MLLMs that already exhibit high performance on annotated charts. Enhancing performance on unannotated charts through CoT remains a challenging task. In contrast, as shown in Tab. 8, SFT provides more significant improvements for the more powerful model Internlm-XComposer-v2 compared to Qwen-VL-Chat (Avg. gain 15.65% v.s. 13.01%, respectively). The improvements are comparable for both annotated and unannotated charts (Δ -0.66% v.s. +0.60%, respectively). This indicates that existing models are required to enhance the fundamental ability to understand unannotated charts, and researchers should prioritize such data during MLLM training.

**Limitations.** 1) ChartBench is required to evaluate more models, and we will continue to follow the rapidly evolving area. 2) Models are highly sensitive to prompt templates, and thus the best prompt template for each model is required to be explored further. 3) The training methods and model architectures for chart perception and reasoning are worth further exploration.

## 7 CONCLUSION

In this paper, we introduce ChartBench to evaluate the chart comprehension abilities of MLLMs. ChartBench significantly expands chart types and requires MLLMs to infer numbers using visual cues like color or legends. We propose improved $Acc+$ for accurate, automated assessments, avoiding manual effort or costly LLM evaluations. We further offer two effective baselines to show how the chain of thought and supervised fine-tuning ameliorate MLLMs on charts. Our evaluation of 21 mainstream MLLMs reveals their limitations in chart interpretation and provides some insights for further directions. We aim to highlight the MLLM's ability to understand charts without data annotations. ChartBench and its code will be publicly available for research.

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

# ChartBench: A Benchmark for Complex Visual Reasoning in Charts
# Supplementary Materials

# A CHARTBENCH STATISTICS

## A.1 EXPLANATION OF *Visually Grounded* IN TABLE 1

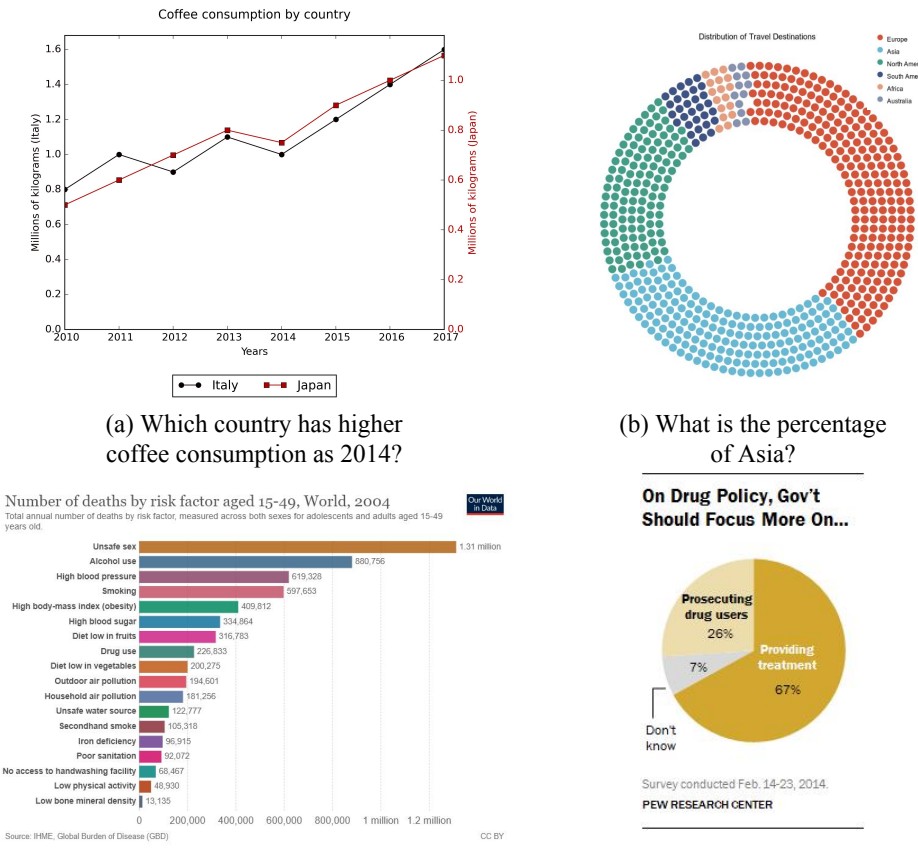

(a) Which country has higher coffee consumption as 2014?

(b) What is the percentage of Asia?

(c) How many people die because of low physical activity?

(d) What is the percentage of Prosecuting drug users?

Figure 5: Examples to illustrate the concept of *Visually Grounded* described in Paper Tab. 1: (a) *Visually Grounded*: The model must understand legends, colors, and dual-coordinate systems to answer the question correctly. Relying solely on spatial relationships is insufficient. (b) *Visually Grounded*: The model needs to count both the number of blue dots and the total number of dots to calculate the proportion representing Asia. (c) Not *Visually Grounded*: The model could perform OCR and find the number closest to the keyword. (d) Not *Visually Grounded*: The model only needs to extract the corresponding text via OCR without visual clues.

## A.2 DESIGN PRINCIPLE

ChartBench has two fundamental design principles. 1) ***Wider range of chart types***. ChartBench expands the 3 common chart types (line, bar, and pie) Masry et al. (2022); Methani et al. (2020); Chen et al. (2024a) to representative 9 chart types in the real world (see Tab. 2 and thumbnails in Appendix H). In the train and test sets, conventional charts account for 61.4% and 54.8%, respectively, while the newly added charts account for 38.6% and 45.2%. ChartBench further divides 9 major categories into 42 subcategories, allowing for a more detailed analysis of MLLM performance. 2) ***More intuitive visual logic***. Unlike existing benchmarks, ChartBench primarily focuses on perception and *visual* logical reasoning. It emphasizes evaluating the ability to extract value from unlabeled charts rather than simple OCR or localization tasks. We assess MLLMs' core visual reasoning skills directly without converting charts into textual descriptions for further textual reasoning. Previous benchmarks mainly provided annotated charts, which led to some approaches extracting tables first and then transforming the problem into purely text-based logic. In contrast, ChartBench includes a larger proportion of unlabeled charts, accounting for 84.96% and 76.20% in train and test splits, respectively, in Tab. 2. MLLMs must accurately extract values based on color or line shape to identify categories and their corresponding coordinate

Table 9: ChartBench training set detailed statistics. We provide statistics based on chart types and more granular image types. Each image will have two kinds of questions: *Acc+* and Number QA.

| Data Split | #Image Number | Chart Type | #Image Number | Image Type | Number | | |
|---|---|---|---|---|---|---|---|
| | | | | | #Image | #*Acc+* QA | #NQA |
| Regular | 40,887 | Line | 7,830 | multi-line plot | 1,744 | 13,952 | 1,744 |
| | | | | multi-line plot (w/i anno) | 1,744 | 13,952 | 1,744 |
| | | | | single line plot | 1,744 | 13,952 | 1,744 |
| | | | | single line plot (w/i anno) | 1,744 | 13,952 | 1,744 |
| | | | | line with error plot | 854 | 6,832 | 854 |
| | | Bar | 24,580 | horizontal single bar plot | 1,891 | 15,128 | 1,891 |
| | | | | horizontal single bar plot (w/i anno) | 1,891 | 15,128 | 1,891 |
| | | | | horizontal multi-bar plot | 1,891 | 15,128 | 1,891 |
| | | | | horizontal stacked bar plot | 1,891 | 15,128 | 1,891 |
| | | | | horizontal stacked bar in percentage plot | 1,890 | 15,120 | 1,890 |
| | | | | vertical single bar plot | 1,891 | 15,128 | 1,891 |
| | | | | vertical single bar plot (w/i anno) | 1,891 | 15,128 | 1,891 |
| | | | | vertical multi-bar plot | 1,891 | 15,128 | 1,891 |
| | | | | vertical stacked bar plot | 1,891 | 15,128 | 1,891 |
| | | | | vertical stacked bar in percentage plot | 1,890 | 15,120 | 1,890 |
| | | | | 3D multi-bar plot | 1,891 | 15,128 | 1,891 |
| | | | | 3D stacked bar plot | 1,891 | 15,128 | 1,891 |
| | | | | 3D stacked bar in percentage plot | 1,890 | 15,120 | 1,890 |
| | | Pie | 8,477 | ring plot | 1,989 | 15,912 | 1,989 |
| | | | | ring plot (w/i anno) | 1,989 | 15,912 | 1,989 |
| | | | | inter sun plot | 521 | 4,168 | 521 |
| | | | | sector plot | 1,989 | 15,912 | 1,989 |
| | | | | pie plot | 1,989 | 15,912 | 1,989 |
| Extra | 25,737 | Area | 5,613 | area plot | 1,871 | 14,968 | 1,871 |
| | | | | area in percentage plot | 1,871 | 14,968 | 1,871 |
| | | | | stacked area plot | 1,871 | 14,968 | 1,871 |
| | | Box | 4,068 | stock plot | 1,356 | 10,848 | 1,356 |
| | | | | vertical box plot | 1,356 | 10,848 | 1,356 |
| | | | | horizontal box plot | 1,356 | 10,848 | 1,356 |
| | | Radar | 3,056 | single radar plot | 764 | 6,112 | 764 |
| | | | | single radar plot (w/i anno) | 764 | 6,112 | 764 |
| | | | | multi-radar plot | 764 | 6,112 | 764 |
| | | | | multi-radar with fill plot | 764 | 6,112 | 764 |
| | | Scatter | 2,046 | 2D scatter plot | 784 | 6,272 | 784 |
| | | | | 2D scatter smooth plot | 784 | 6,272 | 784 |
| | | | | 3D scatter | 478 | 3,824 | 478 |
| | | Node | 3,978 | undirected node plot | 1,989 | 15,912 | 1,989 |
| | | | | directed node plot | 1,989 | 15,912 | 1,989 |
| | | Combination | 6,976 | line & line plot (dual coordinate) | 1,744 | 13,952 | 1,744 |
| | | | | bar & line plot (dual coordinate) | 1,744 | 13,952 | 1,744 |
| | | | | pie & bar combinated plot | 1,744 | 13,952 | 1,744 |
| | | | | pie & pie combinated plot | 1,744 | 13,952 | 1,744 |
| Total | 66,624 | Total | 66,624 | Total | 66,624 | 532,992 | 66,624 |

systems, rather than relying on OCR for answer candidates, which offers a more realistic assessment of MLLMs' visual reasoning abilities of charts.

## A.3 CHART TAXONOMY

ChartBench primarily focuses on the following evaluation aspects: 1) **Bar charts** are the most common and have been the focus of ChartQA and ChartLLaMA. ChartBench includes basic variations such as horizontal and vertical bar orientations, data complexity (single and multiple groups of data), and different representations (regular, percentage, stacked, and 3D bar charts). 2) **Line charts** are commonly used chart types to reflect data trends. ChartBench includes error line charts as well as regular single or multiple groups, with or without annotations line charts. 3) **Pie charts** primarily show the data proportional distribution. ChartBench includes single, nested, doughnut pie charts, and irregular sector charts. 4) **Radar charts** have a straightforward distribution structure and are used to represent multiple attributes of an entity. ChartBench incorporates diverse data complexities (single or multiple groups) and representations (with or without fillings). 5) **Box charts** primarily depict the statistical distribution of a substantial volume of data points. ChartBench collects horizontal and vertical box plots, as well as authentic candlestick charts depicting real stock prices. 6) **Scatter charts** mainly depict the distribution of discrete data. ChartBench includes simple single or multi-group scatter plots, 3D bubble plots, and scatter plots with interpolated smoothing lines. 7) **Area charts** employ color fillings to visually convey the magnitude and distribution of data. ChartBench encompasses single or multiple groups area plots, stacked and percentage stacked area charts. 8) **Node charts** primarily illustrate the logical relationships between

Table 10: ChartBench test set detailed statistics. We provide statistics based on chart types and more granular image types. Each image will have two kinds of questions: *Acc+* and Number QA.

| Data Split | #Image Number | Chart Type | #Image Number | Image Type | Number | | |
|---|---|---|---|---|---|---|---|
| | | | | | #Image | #*Acc+* QA | #NQA |
| Regular | 1,150 | Line | 250 | multi-line plot | 50 | 400 | 50 |
| | | | | multi-line plot (w/i anno) | 50 | 400 | 50 |
| | | | | single line plot | 50 | 400 | 50 |
| | | | | single line plot (w/i anno) | 50 | 400 | 50 |
| | | | | line with error plot | 50 | 400 | 50 |
| | | Bar | 650 | horizontal single bar plot | 50 | 400 | 50 |
| | | | | horizontal single bar plot (w/i anno) | 50 | 400 | 50 |
| | | | | horizontal multi-bar plot | 50 | 400 | 50 |
| | | | | horizontal stacked bar plot | 50 | 400 | 50 |
| | | | | horizontal stacked bar in percentage plot | 50 | 400 | 50 |
| | | | | vertical single bar plot | 50 | 400 | 50 |
| | | | | vertical single bar plot (w/i anno) | 50 | 400 | 50 |
| | | | | vertical multi-bar plot | 50 | 400 | 50 |
| | | | | vertical stacked bar plot | 50 | 400 | 50 |
| | | | | vertical stacked bar in percentage plot | 50 | 400 | 50 |
| | | | | 3D multi-bar plot | 50 | 400 | 50 |
| | | | | 3D stacked bar plot | 50 | 400 | 50 |
| | | | | 3D stacked bar in percentage plot | 50 | 400 | 50 |
| | | Pie | 250 | ring plot | 50 | 400 | 50 |
| | | | | ring plot (w/i anno) | 50 | 400 | 50 |
| | | | | inter sun plot | 50 | 400 | 50 |
| | | | | sector plot | 50 | 400 | 50 |
| | | | | pie plot | 50 | 400 | 50 |
| Extra | 950 | Area | 150 | area plot | 50 | 400 | 50 |
| | | | | area in percentage plot | 50 | 400 | 50 |
| | | | | stacked area plot | 50 | 400 | 50 |
| | | Box | 150 | stock plot | 50 | 400 | 50 |
| | | | | vertical box plot | 50 | 400 | 50 |
| | | | | horizontal box plot | 50 | 400 | 50 |
| | | Radar | 200 | single radar plot | 50 | 400 | 50 |
| | | | | single radar plot (w/i anno) | 50 | 400 | 50 |
| | | | | multi-radar plot | 50 | 400 | 50 |
| | | | | multi-radar with fill plot | 50 | 400 | 50 |
| | | Scatter | 150 | 2D scatter plot | 50 | 400 | 50 |
| | | | | 2D scatter smooth plot | 50 | 400 | 50 |
| | | | | 3D scatter | 50 | 400 | 50 |
| | | Node | 100 | undirected node plot | 50 | 400 | 50 |
| | | | | directed node plot | 50 | 400 | 50 |
| | | Combination | 200 | line & line plot (dual coordinate) | 50 | 400 | 50 |
| | | | | bar & line plot (dual coordinate) | 50 | 400 | 50 |
| | | | | pie & bar combined plot | 50 | 400 | 50 |
| | | | | pie & pie combined plot | 50 | 400 | 50 |
| Total | 2,100 | Total | 2,100 | Total | 2,100 | 16,800 | 2,100 |

nodes. ChartBench includes directed and undirected graphs, as well as simple and complex node-link diagrams. 9) **Combination charts** combine the above-mentioned chart types. ChartBench includes dual coordinate system charts (e.g. line and bar charts), multi-level pie charts, and combinations between bar and pie charts.

## A.4 DATA SPLITTING

Tab. 9 and Tab. 10 show the hierarchical relationship and quantity of each type of chart in detail. The distribution of the train and test set is slightly different because we guarantee that each subclass in the test split has 50 data points. The charts in the test set are all redrawn using real-world plotting websites to ensure they accurately reflect real-world scenarios as much as possible. For each chart, we generate questions on 5 different tasks to evaluate MLLMs' basic performance on perception and cognition. Notice that some categories have two variants, i.e., *w/i* and *w/o* annotations. Although the dataset mainly consists of unannotated charts, we only report the results of comparisons between the *w/i* and *w/o* chart versions derived from the same table in our experiments to ensure fair comparisons.

## B QUALITY INSPECTION

In this section, we discuss the quality control of ChartBench through rule-based, manual, and GPT-powered automated inspections.

### B.1 RULE-BASED INSPECTION

We begin with rule-based checks and filtering. For all charts in ChartBench that are rendered through code execution, we filter out cases with compilation failures or warnings. To ensure image quality, we conduct both manual inspection and automatic checks using GPT. For the instructions, we generate both positive and negative sample queries for each chart based on templates, with the only difference being the ground truth (GT). For different chart types, we establish a set of rules to evaluate the reasonableness of the generated instructions, such as percentage calculations in distribution charts. It's worth noting that the GT for negative samples is randomly sampled from the same meta table, which might result in values that are identical or very similar to the positive one. Therefore, we further filter and adjust it based on the relative differences between the two GTs.

### B.2 GPT-BASED INSPECTION

The rule-based checks can only filter out the potential errors that we have anticipated in advance, so we also use GPT to correct some rigid errors or grammatical issues in the template-generated text. As shown in Fig. 6, we provide the prompt template for refining the query. To conserve API resources, we group the generated questions into batches of 10 and use the following prompt to correct any grammatical errors.

Figure 6: The prompt to polish queries generated by templates.

**Prompt**:
Please fix the grammatical and semantic errors in the following 10 questions:
1. Questions are generated based on templates, please be careful not to modify proper nouns;
2. Make as few changes as possible to avoid changing the original intention of the question;
3. If the content of the question is not understandable, return "FAILED".

Please read the following example:
Question 1: According to this node_link_dir chart, the node Beijing points to node New York.
Question 2: According to this bar chart, the LA has the highest sales volume (in percentage) 62 at month Aug.

**Your answer:**
According to this directed node link chart, the node Beijing points to node New York.
According to this bar chart, the LA has the highest percentage of sales volume 62% at August.

**User Questions:**
{}

**Your Answer:**

With the generated chart and query, we further judge the generated quality and its correlation. We randomly sample a toy subset from the ChartBench and use GPT-4O to evaluate the samples on aspects such as *relevance*, *consistency*, *information richness*, *multimodal synergy*, *ambiguity*, and *overall quality*. The prompt used for scoring is shown in Fig. 7. Each aspect is rated on a scale of 1 to 5, with a particular focus on samples where GPT gives an overall rating below 3. The statistical results of this evaluation are presented in Tab. 11. The results show that approximately 5% of the samples can be considered flawed when we define flaws as scores below 3. Upon manual review, we found that the primary issues are label occlusion or overly dense elements, which do not affect the accuracy of proposed queries.

Notice that GPT-4O fails to achieve perfect performance on ChartBench. However, we chose it as the evaluation model because we provide it with meta tables as additional supplementary information. When evaluating ChartBench, GPT-4O only received the question and the chart as input and was required to provide specific numerical answers. Due to its difficulty in providing precise numerical values, GPT-4O's performance on ChartBench is not perfect. During the quality assessment, GPT-4O was given the chart, the table used to generate the chart, the question, and the answer. In practice, GPT-4O tends to struggle with precise numerical extraction on unannotated charts but performs well in understanding visual markers.

### B.3 MANUAL INSPECTION

The 2,100 charts in the test split have been reviewed by at least three researchers to ensure that they filter out drawing errors, severe label occlusion, mismatches with the questions, etc., which is also confirmed by

Figure 7: The prompt to evaluate the chart, query, and processed meta table with GPT-4O API.

**Prompt**:
Please evaluate the following chart and query pair, and assign scores based on the criteria listed below. For each criterion, provide a score (1 to 5) and a brief justification for your rating. The score descriptions are as follows:
- 1: Very mismatched or unreasonable
- 2: Noticeably mismatched or unreasonable
- 3: Mostly matched, but with some minor issues
- 4: Well-matched, generally reasonable
- 5: Perfectly matched, very reasonable

**Evaluation Criteria**:
1. *Visual Content Relevance*: How closely do the main elements in the chart relate to the content of the query?
2. *Contextual Consistency*: Are the chart and the query consistent? Are the chart and the meta table consistent?
3. *Informational Richness*: Does the chart provide enough information to answer the query, or does it help guide toward an answer?
4. *Multimodal Synergy*: Do the chart and query effectively complement each other, providing a complete understanding?
5. *Ambiguity*: Is there any potential for misunderstanding or ambiguity between the chart and the query? Is chart flawed? A higher score indicates lower ambiguity.

**Evaluation**:
Please score the following input and provide ratings and justifications for each criterion:
- *Chart*: "path/to/chart"
- *Query*: "question and answer"
- *Meta Table*: "table in CSV format"

**Scoring Template**:
- Visual Content Relevance: X/5, \[Justification\]
- Contextual Consistency: X/5, \[Justification\]
- Informational Richness: X/5, \[Justification\]
- Multimodal Synergy: X/5, \[Justification\]
- Ambiguity: X/5, \[Justification\]
- Overall: X/5, \[Justification\]

**Your Answer**:

Table 11: Statistical results of GPT automatically evaluation. The *Flowed Case* means < 3 points.

| Criteria | Relevance | Consistency | Richness | Synergy | Ambiguity | Overall |
|---|---|---|---|---|---|---|
| Flowed Case | 2/200 | 5/200 | 0/200 | 2/200 | 9/200 | 11/200 |
| Average Score | 4.47 | 4.16 | 4.74 | 4.44 | 4.12 | 4.56 |

the automated review result from GPT-4O. Furthermore, ChartBench undergoes human testing (results in Appendix E), during which we collect user feedback and have already made adjustments to it.

ChartBench consists of 42 categories, including samples generated from online charting websites and code-based templates. During the manual inspection, we do not modify the charts from the online websites but make proper adjustments to the plotting code for 10 chart categories. The specific modifications are as follows.

Online website generated charts:

1. *Resolution Concerns*: The images from the online websites have high resolution, making the text difficult to read on smaller user mobile screens during the human surveys. This issue doesn't appear on larger monitors.

2. *Lack of Data Point Labels*: Some comparative questions involving charts without data point labels rely solely on the length of bars for comparison. When the values are close, users find it difficult to make accurate judgments. We believe the model should handle this since the input charts are lossless, allowing the model to determine the absolute size of the bars.

Code generated charts:

1. *Percentage Accumulation Charts*: Some themes are not intuitive, like a percentage distribution chart for temperatures from January to December. Users may misinterpret 12% as 12°C. This issue affects four subsets (50*4). We add percentage information in the titles and along the y-axis to clarify.

2. *Label Obstruction*: Sometimes, label text is obstructed due to length or other factors. This issue appears in four subsets (50*4). We adjust the padding to ensure all text is positioned away from the chart to avoid obstruction.

3. *Dual Axis Charts*: We use color to convey the correspondence between data points and their respective axes. However, in some cases, the contrast is insufficient, making it hard for users to distinguish between them. This issue affects two subsets (50*2). We update the color map, removing low-contrast styles such as *civitas*, *Greys*, and *YlGn*.

# C PARTICIPATING MLLMs

## C.1 ARCHITECTURE

Table 12: Open-sourced model architecture. Note that we classify connector components such as QFormer (Li et al., 2023c) as the visual branch for brevity. Mem.: the maximum GPU memory usage during inference. Time: the average inference time per QA. Due to the multiple visual encoders in SPHINX Lin et al. (2023), which extract more robust visual representations, *mixed* refers to QFormer (Li et al., 2023c), OpenCLIP ViT-L/14 (Ilharco et al., 2021), OpenCLIP ConvNeXt-XXL (Ilharco et al., 2021; Cherti et al., 2023), DINOv2-ViT-g/14 (Oquab et al., 2023) and MLP.

| Models | Total Size | LLM Branch | LLM Size | Visual Branch | Visual Size | Peak Memory (G) | Inference Time (s) |
|---|---|---|---|---|---|---|---|
| BLIP2 Li et al. (2023c) | 12.1B | FlanT5-XXL | 11B | EVA-CLIP-g/14 | 1B | 39.60 | 0.176 |
| CogVLM-Chat Wang et al. (2023b) | 17B | Vicuna-7B | 7B | EVA-02-CLIP-E/14 | 4.4B | 39.60 | 1.455 |
| InstructBLIP Dai et al. (2023) | 8.2B | Vicuna-7B | 7B | EVA-CLIP-g/14 | 1B | 36.50 | 0.895 |
| Internlm-XComposer Zhang et al. (2023) | 8.2B | InternLM-Chat-7B | 7B | EVA-CLIP-g/14 | 1B | 22.20 | 0.707 |
| LLaVA-v1.5 Liu et al. (2023e) | 13.4B | Vicuna-13B | 13B | CLIP ViT-L/14@336px | 304M | 16.50 | 0.534 |
| MiniGPT-v2 Chen et al. (2023a) | 8.1B | LLaMA2-Chat-7B | 7B | EVA-ViT-g/14 | 1B | 17.20 | 0.236 |
| mPLUG-Owl-bloomz Ye et al. (2023b) | 7.4B | Bloomz-7B | 7B | CLIP ViT-L/14 | 304M | 16.00 | 0.284 |
| Qwen-VL-Chat Bai et al. (2023b) | 9.6B | Qwen-7B | 7.7B | OpenCLIP ViT-G/14 | 1.9B | 21.00 | 0.269 |
| Shikra Chen et al. (2023b) | 7.4B | Vicuna-7B | 7B | CLIP ViT-L/14 | 304M | 15.60 | 0.561 |
| SPHINX Lin et al. (2023) | 15.7B | LLaMA-13B | 13B | Mixed | 2.7B | 29.6 * 2 | 0.581 |
| VisualGLM Du et al. (2022) | 7.8B | ChatGLM-6B | 6.2B | EVA-CLIP-g/14 | 1B | 16.00 | 0.201 |
| ChartLlama Han et al. (2023) | 13.4B | Vicuna-13B | 13B | CLIP ViT-L/14@336px | 304M | 29.00 | 0.593 |
| DocOwl-v1.5 Hu et al. (2024) | 8.1B | Bloomz-7B | 7B | CLIP ViT-L/14 | 304M | 37.5 | 0.483 |
| Mini-Gemini Li et al. (2024) | 14B | Vicuna-13B | 13B | ConvNext-L + CLIP ViT-L/14 | 502M | 32.45 | 3.951 |
| Internlm-XComposer-v2 Dong et al. (2024) | 8B | InternLM2-7B | 7B | CLIP ViT-L/14 | 304M | 23.72 | 0.945 |
| OneChart Chen et al. (2024a) | 13.4B | Vicuna-13B | 13B | SAM-base ViT | 304M | 37.62 | 2.201 |
| ChartVLM Xia et al. (2024) | 7.4B | Vicuna-7B | 7B | Pix2Struct-base | 304M | 17.83 | 2.831 |
| CogAgent Hong et al. (2023) | 7.4B | Vicuna-7B | 7B | EVA2-CLIP-L | 304M | 18.82 | 2.548 |

We evaluate 18 main-stream open-sourced and 3 closed-sourced MLLMs on ChartBench. The open-source models include **BLIP2** Li et al. (2023c), **CogVLM-Chat** Wang et al. (2023b), **InstructBLIP** Dai et al. (2023), **InternLM-XComposer** (Zhang et al., 2023), **LLaVA-v1.5** Liu et al. (2023d), **MiniGPT-v2** Chen et al. (2023a), **mPLUG-Owl-bloomz** Ye et al. (2023b), **Qwen-VL-Chat** Bai et al. (2023b), **Shikra** Chen et al. (2023b), **SPHINX** Lin et al. (2023), **VisualGLM** (Du et al., 2022; Ding et al., 2021), **ChartLlama** Han et al. (2023), **DocOwl-v1.5** Hu et al. (2024), **Mini-Gemini** Li et al. (2024), **Internlm-XComposer-v2** Dong et al. (2024), **OneChart** Chen et al. (2024a), **ChartVLM** Xia et al. (2024), **CogAgent** Hong et al. (2023), while the closed-source models contain **Baidu ERNIE** BaiDu, **GPT-4V / GPT-4O** OpenAI (2023). Some close-sourced models do not provide efficient APIs, so we randomly sample a subset for evaluations. Tab. 12 summarizes the visual and LLM branch architecture, along with memory costs and inference latency on NVIDIA A100-40G GPUs.

**BLIP2** (Li et al., 2023c) proposes a lightweight Query Transformer to leverage off-the-shelf frozen image encoders and LLMs, which is pre-trained via a two-stage strategy. We test *BLIP-2 ViT-g FlanT5-xxl* (Fang et al., 2023; Chung et al., 2022).

**CogVLM-Chat** (Wang et al., 2023b) bridges the gap between the frozen vision encoder and LLM by integrating a visual expert module in the transformer block. We test the version *CogVLM-Chat-17B*, which leverages Vicuna-7B finetuned from LLaMA2 (Touvron et al., 2023b) and EVA-02-CLIP-E/14 (Sun et al., 2023) as unimodal encoders.

**InstructBLIP** (Dai et al., 2023) extends the framework of instruction tuning to the BLIP2, and demonstrates its appealing ability of generalization. We carry out evaluations on version *IntructBLIP-7B*, which uses EVA-CLIP-g/14 as vision encoder and Vicuna-7B as text encoder.

**InternLM-XComposer** (Zhang et al., 2023) is an instruction-tuned MLLM based on InternLM (Team, 2023). It is empowered by tuning on extensive multimodal multilingual concepts with carefully crafted strategies. We test the released version of *InternLM-XComposer-7B* with InternLM-Chat-7B (Team, 2023) and EVA-CLIP-g/14.

**LLaVA-v1.5** (Liu et al., 2023d) is a variant of LLaVA (Liu et al., 2023e) with exquisite modifications, such as curated datasets, larger input resolution, modality connector and prompt engineering. We test the version of *LLaVA-v1.5-13B* with Vicuna-13B and CLIP ViT-L/14@336px (Radford et al., 2021).

**MiniGPT-v2** (Chen et al., 2023a) proposes a three-stage training paradigm and uses unique identifiers for different tasks, building a unified interface for multiple vision-language tasks. We test *MiniGPT-v2-7B* version, leveraging LLaMA2-Chat-7B and EVA-ViT-g/14 as unimodal encoders.

**mPLUG-Owl-bloomz** (Ye et al., 2023b) equips LLM with visual abilities by modularized learning of LLM, visual knowledge module, and visual abstractor module. We conduct evaluations on *mPLUG-Owl-bloomz-7B* version with Bloomz-7B (Muennighoff et al., 2022) and CLIP ViT-L/14.

**Qwen-VL-Chat** (Bai et al., 2023b) is trained with alignment techniques, which support more flexible interaction, such as multiple image inputs, multi-round question answering and creative capability. We test the version of

*Qwen-VL-Chat-7B* with Qwen-7B (Bai et al., 2023a) and OpenCLIP ViT-G/14 (Ilharco et al., 2021; Cherti et al., 2023).

*Shikra* (Chen et al., 2023b) proposes to tackle spatial coordinate inputs and outputs in natural language without extra plug-in models or vocabularies. We test the version *Shikra-7B* which uses Vicuna-7B and CLIP ViT-L/14.

*SPHINX* (Lin et al., 2023) showcases the superior capability of multi-modal understanding with a joint mixing of model weights, tuning tasks, visual embeddings, and sub-images of different scales. We conduct the test on version *SPHINX-13B*, whose visual branch (note as mixed in Tab. 12) is a mixture of QFormer, OpenCLIP ViT-L/14, OpenCLIP ConvNeXt-XXL and DINOv2-ViT-g/14 (Oquab et al., 2023) and LLM branch is LLaMA-13B (Touvron et al., 2023a).

*VisualGLM* (Du et al., 2022; Ding et al., 2021) is an open-source, multi-modal dialogue language model. We test *VisualGLM-6B* based on ChatGLM-6B (Du et al., 2022) and EVA-CLIP-g/14.

*ChartLlama* (Han et al., 2023) proposes to endow *LLaVA-v1.5* with the capability of chart understanding and generation. We evaluate *ChartLlama-13B*, which uses Vicuna-13B and CLIP ViT-L/14@336px.

*DocOwl-v1.5* (Hu et al., 2024) propose to merge visual tokens horizontally to handle high-resolution images and align all data with markdown. We evaluate the DocOwl-Omni version in our experiments, which is good at document/webpage parsing and VQA with concise answers.

*Mini-Gemini* (Li et al., 2024) adopt two visual encoders to handle low and high-resolution images. This approach is applicable to a variety of LLMs, and we select the Mini-Gemini-Vicuna-13B for evaluation.

*Internlm-XComposer-v2* Dong et al. (2024) introduces a Partial LoRA approach, applying additional LoRA parameters only to image tokens. This preserves the integrity of the model's pre-trained language knowledge while enabling precise vision understanding and literary-level text composition. Compared to the first version, the performance of Internlm-XComposer-v2 has been significantly improved.

*OneChart* Chen et al. (2024a) introduces an auxiliary token placed at the beginning of the token sequence, along with an additional decoder. This decoder will provide a Python dictionary about chart metadata. OneChart needs to be used in conjunction with other MLLMS, so we choose LLaVA-v1.6, which is the best model in the paper.

*ChartVLM* Xia et al. (2024) extracts metadata of chart based on Pix2Struct Lee et al. (2023). It employs an instruction adapter to dynamically select tasks based on user instructions and provides two decoders for the base and complex queries. ChartVLM has two variants and we select ChartVLM-Base-7.3B for evaluations.

*CogAgent* Hong et al. (2023) is a visual-linguistic model specialized in GUI understanding and planning while retaining strong capabilities across general cross-modal tasks. By leveraging both low and high-resolution image encoders, CogAgent supports input at $1120 \times 1120$ resolution, enabling it to recognize even tiny page elements and text.

## C.2 MODEL PERFORMANCE EXPLANATION

*OneChart* (Chen et al., 2024a) is a hierarchical architecture model. It trains a decoder to convert charts to CSV tables as a prompt for LLaVA-V1.6 to inference. OneChart's performance on ChartBench is abnormal and inconsistent with its performance on ChartQA. Unlike ChartQA, the metadata in ChartBench is longer, and the charts do not have data point annotations. In this case, the Python dictionary extracted by OneChart is inaccurate and results in generally longer table prompts. After analyzing specific cases, we find that OneChart always fails to follow instructions on the cases with longer prompts, even for simple yes-or-no binary outputs.

*ChartVLM* Xia et al. (2024) is a multi-decoder structure. The router selects the corresponding decoder according to the difficulty of the current query. However, ChartVLM shows the opposite performance on *Acc++* and NQA tasks (Tab. 3 8.02% v.s. 43.74% in regular charts and 5.92% v.s. 18.21% in extra charts). Case studies show that ChartVLM tends to generate numbers or phrases, ignoring various yes/no prompt constraints. As a result, the current metric cannot parse the output of ChartVLM. However, it is worth noting that although some of ChartVLM's outputs are not strictly yes or no, they are consistent with the correct answers. While LLMs can be used to correct this bias, we have retained the original results for a fair comparison.

*ChartLlama* Han et al. (2023) is a supervised fine-tuning model with LoRA Hu et al. (2021) based on LLaVA-v1.5 Liu et al. (2023e) with a large number of generated chart instruction data. As shown in Tab. 3, ChartLlama is the best-performing model on ChartQA, but it fails to catch up with LLaVA-v1.5 on ChartBench. Notice that ChartLlama is still better than LLaVA-v1.5 on NQA tasks but performs poorly on *Acc++* tasks that mainly require yes/no answers. This indicates that ChartLlama's ability to extract values is relatively good, but SFT may reduce the model's ability to follow instructions, causing it to consistently provide numerical answers instead of yes/no responses.

***mPLUG-Owl-bloomz*** Ye et al. (2023b) performs well on the ChartBench generally. However, when asked to provide a concise answer consisting of only one word or phrase, it becomes difficult to control the length of the output. It tends to generate descriptive statements, which explains its poor performance on the NQA tasks of ChartBench and ChartQA. Even if we apply LLMs to extract the key information from its output statements, the results are still unsatisfactory. Considering the model's impressive performance on *Acc++* tasks, we believe that mPLUG-Owl-bloomz shares a similar issue with ChartVLM. The excessive emphasis on descriptive summaries during the supervised fine-tuning process hinders the model's ability to generate short and concise content. This limitation arises from the training procedure, which prioritizes detailed and elaborate explanations rather than producing succinct answers. As a result, when tasked with generating brief responses, the model struggles to control the length of its output and tends to generate lengthy and descriptive statements instead. This issue adversely affects its performance on tasks that require concise answers, such as the ChartQA and NQA tasks in ChartBench.

## D EXPERIMENTAL SETTINGS

### D.1 EVALUATION IMPLEMENTATION

We locally deploy 18 open-source MLLMs and conduct evaluations on A100-40G GPUs. To maintain consistency, we strictly utilize a single GPU to evaluate the *Chat* version of each MLLM with the corresponding system prompt. We employ the zero-shot evaluation manner to avoid any potential data leakage and guarantee fair comparisons. It is to highlight that the choice of prompts remarkably influences the MLLMs' response. Hence, we extensively conduct experiments with several prompts and select the one yielding the best performance (see detail in Tab. 13). For NQA task, all models adopt the same constraints as ChartQA, i.e.,

```
user\nAnswer the question using a single word or phrase.  {}\nassistant:
```

Although this prompt is clear enough, some models will not be generated efficiently, so we have made some adjustments to this instruction to guide the output style of models.

### D.2 ZERO-SHOT PROMPT

Table 13: The mapping between the template and the MLLMs is displayed. Different prompt templates can greatly affect the performance. The values we report are the best results in each template. ICL: in-context learning style. Green: system prompt. Pink: *Acc+* instruction. Blue: the judgement based on the corresponding chart. The ground truth in the judgment has been **bolded**.

| Prompt Style | Model | Prompt Example |
|---|---|---|
| BLIP2 style | BLIP2 Li et al. (2023c)
CogVLM Wang et al. (2023b)
MiniGPT-v2 Chen et al. (2023a)
Internlm-Xcomposer Zhang et al. (2023)
ChartVLM Xia et al. (2024)
CogAgent Hong et al. (2023)
DocOwl-v1.5 Hu et al. (2024)
Internlm-Xcomposer-v2 Dong et al. (2024) | Question: According to this chart, the Rainfall in Millimeters of Months Jul is around **100.0**. Please answer yes or no. Answer: |
| LLaVA style | LLaVA-v1.5 Liu et al. (2023d)
ChartLlama Han et al. (2023)
Mini-Gemini Li et al. (2024) | You are a data analyst, good at dealing with chart data. Please determine whether the user's judgments on this chart are correct. You only need to answer [**yes**] or [**no**]. The judgment from the User is: According to this chart, the Rainfall in Millimeters of Months Jul is around **100.0**. Please answer **yes or no**. Your Answer: |
| LLaVA style no or yes | Qwen-VL-Chat Bai et al. (2023b)
SPHINX Lin et al. (2023)
OneChart Chen et al. (2024a) | You are a data analyst, good at dealing with chart data. Please determine whether the user's judgments on this chart are correct. You only need to answer [**no**] or [**yes**]. The judgment from the User is: According to this chart, the Rainfall in Millimeters of Months Jul is around **100.0**. Please answer **no or yes**. Your Answer: |
| LLaVA style ICL | InstructBLIP Dai et al. (2023)
mPLUG-Owl-bloomz Ye et al. (2023b)
Shikra Chen et al. (2023b)
VisualGLM Du et al. (2022) | You are a data analyst, good at dealing with chart data. Please determine whether the user's judgments on this chart are correct. You only need to answer [yes] or [no]. Here is an example:
User: **<image>**
User: The figure is a line chart.
You: yes.

Following the above example:
The query from the User is: According to this chart, the Rainfall in Millimeters of Months Jul is around **100.0**. Your Answer: |

During the evaluation on ChartBench, we observe that the zero-shot performance of MLLMs is heavily influenced by the prompt templates, which indirectly reflects the current lack of robustness in MLLMs. To ensure fairness,

we select the most appropriate templates used by each MLLM's official implementation for testing. In Tab. 13, we provide the corresponding mappings between the MLLMs and the prompt templates that yield the best $Acc++$ metric. We also test more than 10 other prompt templates, but fail to produce the best $Acc++$, which thus are not summarized in the table.

It is worth noting that the MLLMs tend to randomly answer the judgment questions in ChartBench if they cannot accurately comprehend the chart. Specifically, we observe a tendency for these models to favor the first option (e.g., *yes* in a yes-or-no scenario). Therefore, we provide two sets of LLaVA-style prompt templates, differing only in the order of the yes-or-no options. We have performed similar operations on other templates as well, but none of the MLLMs exhibited optimal performance on these prompt templates. Therefore, we did not include specific details about them in Tab. 13.

ICL stands for *In Context Learning*. We only adopt the template format as shown in Tab. 13 to standardize the output of MLLMs. We do not conduct actually ICL for our evaluations. In other words, for *LLaVA-style ICL*, we just adopt a single-turn dialogue, and only the queried chart is provided as the image input.

### D.3 SUPERVISED FINE-TUNING IMPLEMENTATION

Using the ChartBench data, we propose an SFT baseline. Here, we introduce the basic setup of our training process. Considering the imbalance between the $Acc++$ and NQA content in the instruction data, we manually balance these two types of data to prevent the model from developing a prediction bias.

***Qwen-VL-Chat.*** We perform SFT for 3 epochs using instructions. We keep the parameters of the vision encoder frozen and use LoRA to update only the LLM branch. Training is conducted with DeepSpeed's *Zero2 configuration* in half-precision *bf16*, with a weight decay of 0.05. The optimizer is AdamW with adam_beta2 set to 0.98. The input image resolution is $448 \times 448$, the batch size is 1, and the learning rate is $2e-5$. The entire training process consumes 12 A100 GPU days. We do not perform alignment training for the connector because Qwen-VL's connector is small and can be updated along with the LLM parameters.

***Internlm-XComposer-v2.*** We use the chart-CSV pair for alignment training over 2 epochs, freezing the parameters of the ViT Encoder and LLM, and only updating the connector. Then, we perform 1 epoch of supervised fine-tuning using the chart instruction data, updating both the connector and the LLM branch with LoRA. We set a learning rate of $1e-5$ and the AdamW optimizer (adam_beta2=0.95). DeepSpeed's *Zero2 configuration* is employed, with half-precision *bf16* for parameter updates. The input image resolution is $490 \times 490$, and the batch size is set to 1. This experiment approximately requires 15 A100-GPU days.

## E ADDITIONAL RESULTS

In this section, we 1) expand the discussion to include the model's $Acc++$ (Tab. 14) and *NQA* (Tab. 15) performance on each chart type, details of FixedCoT (Fig. 8), and the relationship between model performance and image resolution (Fig. 9); 2) provide results using accuracy as a metric (Tab. 18 & 19); 3) show evaluation results on ChartQA by image type (Tab. 20 & 21); 4) present human evaluation results on ChartBench (Tab. 22); 5) offer specific evaluation samples (Fig. 10 & 11); and 6) provide sample analyses of SOTA, i.e., GPT-4 (Fig. 12).

### E.1 FURTHER STUDY

**Results w.r.t. Chart Types.** Tab. 14 & 15 illustrate the performance of $Acc++$ and *GPT-acc* w.r.t. chart types. In general, the current MLLMs demonstrate limited proficiency in chart recognition and encounter significant challenges. For certain chart types (e.g., radar or combination chart), some MLLMs achieve close to 0% $Acc++$, indicating their inability to extract key information from charts and insensitivity to both positive and negative interrogations. Note that the $Acc++$ metric approaches 0% under random guessing, as discussed in Sec. 3.4. We also provide results of the vanilla accuracy metric in Appendix E.2, where the baseline should be 50%.

Specifically, some MLLMs like Qwen-VL-Chat and mPLUG-Owl demonstrate satisfying chart recognition capabilities, which may be attributed to their instruction tuning on chart data. The corresponding performance is lower than their reported results in ChartQA (Masry et al., 2022; Han et al., 2023), primarily because their chart recognition depends on OCR capability rather than robust visual logical reasoning. In ChartBench, the proportion of annotated charts is notably low (about 20% in Tab. 2). The majority of queries demand MLLMs to employ visual, logical reasoning, which is quite challenging for these models. VisualGLM and Shikra perform poorly, possibly due to their smaller LLM sizes and weaker visual encoding branches. MLLMs exhibit satisfactory performance on regular charts, but there is still substantial potential for improvement when it comes to handling more intricate graphics.

Table 14: The zero-shot *Acc++* (%) performance w.r.t. chart types.

| Models | Regular Type | | | | Extra Type | | | | | | | *Acc++* |
|---|---|---|---|---|---|---|---|---|---|---|---|---|
| | Line | Bar | Pie | Avg. | Area | Box | Radar | Scatter | Node | Combin. | Avg. | |
| *Open source MLLMs* | | | | | | | | | | | | |
| VisualGLM Du et al. (2022) | 10.80 | 1.96 | 0.00 | 3.46 | 1.17 | 8.50 | 0.25 | 3.33 | 15.50 | 5.13 | 4.22 | 3.79 |
| ChartVLM Xia et al. (2024) | 10.70 | 8.04 | 4.62 | 8.02 | 7.67 | 6.67 | 5.25 | 5.50 | 0.00 | 6.50 | 5.92 | 6.90 |
| Shikra Chen et al. (2023b) | 7.40 | 10.62 | 4.50 | 8.59 | 6.00 | 11.33 | 11.88 | 4.17 | 8.50 | 3.63 | 7.50 | 8.11 |
| OneChart Chen et al. (2024a) | 15.10 | 12.27 | 9.12 | 12.34 | 7.00 | 7.33 | 2.75 | 6.33 | 53.50 | 7.75 | 8.75 | 12.04 |
| InstructBLIP Dai et al. (2023) | 24.40 | 15.04 | 19.10 | 17.96 | 4.33 | 7.33 | 2.00 | 12.50 | 9.00 | 2.38 | 5.50 | 12.49 |
| CogVLM Wang et al. (2023b) | 10.50 | 14.58 | 17.90 | 14.41 | 12.50 | 9.67 | 16.00 | 14.33 | 16.00 | 6.13 | 11.89 | 13.30 |
| Internlm-XComposer Zhang et al. (2023) | 16.00 | 20.42 | 21.50 | 19.70 | 4.50 | 14.50 | 15.00 | 12.00 | 8.50 | 5.13 | 10.11 | 15.49 |
| SPHINX Lin et al. (2023) | 18.40 | 15.54 | 23.40 | 17.87 | 12.00 | 8.17 | 19.00 | 17.17 | 31.00 | 25.88 | 17.92 | 17.89 |
| CogAgent Hong et al. (2023) | 18.60 | 23.96 | 11.00 | 20.39 | 15.67 | 16.50 | 9.38 | 11.67 | 27.50 | 15.50 | 14.36 | 18.07 |
| BLIP2 Li et al. (2023c) | 29.60 | 17.35 | 24.90 | 21.65 | 6.17 | 10.67 | 17.63 | 22.00 | 33.00 | 28.00 | 18.44 | 20.24 |
| ChartLlama Han et al. (2023) | 28.90 | 19.35 | 22.10 | 22.02 | 16.50 | 13.33 | 25.00 | 28.50 | 25.50 | 26.38 | 22.56 | 22.26 |
| MiniGPT-v2 Chen et al. (2023a) | 26.70 | 21.54 | 20.20 | 22.37 | 21.67 | 24.67 | 25.88 | 28.17 | 15.50 | 27.13 | 25.06 | 23.55 |
| LLaVA-v1.5 Liu et al. (2023e) | 34.40 | 24.73 | 19.10 | 25.61 | 26.83 | 25.67 | 28.63 | 26.00 | 41.50 | 27.38 | 27.39 | 26.39 |
| mPLUG-Owl-bloomz Ye et al. (2023b) | 37.50 | 24.73 | 26.10 | 27.80 | 21.33 | 25.83 | 26.50 | 24.17 | 28.50 | 27.50 | 25.47 | 26.78 |
| Qwen-VL-Chat Bai et al. (2023b) | 41.00 | 20.96 | 40.00 | 29.46 | 28.83 | 24.17 | 35.00 | 19.50 | 18.50 | 25.50 | 26.56 | 28.18 |
| DocOwl-v1.5 Hu et al. (2024) | 49.10 | 31.08 | 31.62 | 36.27 | 12.17 | 24.00 | 20.50 | 35.33 | 26.00 | 40.25 | 26.86 | 31.62 |
| Mini-Gemini Li et al. (2024) | 37.60 | 40.19 | 40.00 | 39.57 | **36.83** | 26.50 | 30.00 | 37.17 | 43.00 | 27.00 | 31.81 | 36.54 |
| Internlm-XComposer-v2 Dong et al. (2024) | 70.60 | 51.50 | 62.75 | 57.89 | 30.17 | **31.33** | 43.50 | 52.00 | 52.50 | **46.12** | 41.75 | 51.34 |
| *Closed source MLLMs* | | | | | | | | | | | | |
| ERNIE BaiDu | 44.00 | 45.00 | 57.00 | 47.39 | **45.00** | 30.00 | 40.00 | 51.67 | 70.00 | 56.25 | 46.39 | 46.95 |
| GPT-4V OpenAI (2023) | 74.00 | 41.54 | 63.00 | 53.26 | 33.30 | 46.67 | **57.50** | 70.00 | **100.00** | 56.25 | 55.83 | 54.39 |
| GPT-4O OpenAI (2023) | **86.00** | **51.92** | **78.00** | **65.00** | 36.67 | **63.33** | **57.50** | **83.33** | **100.00** | **65.00** | **63.33** | **64.27** |

Table 15: The zero-shot *NQA* (%) performance w.r.t. chart types.

| Models | Regular Type | | | | Extra Type | | | | | | | *NQA* |
|---|---|---|---|---|---|---|---|---|---|---|---|---|
| | Line | Bar | Pie | Avg. | Area | Box | Radar | Scatter | Node | Combin. | Avg. | |
| *Open source MLLMs* | | | | | | | | | | | | |
| BLIP2 Li et al. (2023c) | 0.80 | 1.38 | 0.00 | 0.96 | 0.00 | 0.67 | 4.00 | 2.67 | 31.00 | 1.00 | 4.84 | 2.71 |
| OneChart Chen et al. (2024a) | 1.20 | 2.31 | 3.20 | 2.26 | 0.00 | 1.33 | 0.50 | 10.67 | 6.00 | 3.50 | 3.37 | 2.76 |
| InstructBLIP Dai et al. (2023) | 0.40 | 1.23 | 0.40 | 0.87 | 1.33 | 0.67 | 0.50 | 0.00 | 46.00 | 0.50 | 5.37 | 2.90 |
| VisualGLM Du et al. (2022) | 1.20 | 2.77 | 0.00 | 1.83 | 0.00 | 0.67 | 0.50 | 2.67 | 38.00 | 1.00 | 4.84 | 3.19 |
| Internlm-XComposer Zhang et al. (2023) | 0.80 | 1.54 | 0.80 | 1.22 | 2.67 | 0.00 | 2.00 | 1.33 | 43.00 | 1.00 | 5.79 | 3.29 |
| MiniGPT-v2 Chen et al. (2023a) | 2.80 | 1.85 | 3.60 | 2.43 | 2.00 | 0.67 | 3.00 | 3.33 | 30.00 | 2.50 | 5.26 | 3.71 |
| mPLUG-Owl-bloomz Ye et al. (2023b) | 0.40 | 2.77 | 3.20 | 2.35 | 0.00 | 0.67 | 11.00 | 0.67 | 33.00 | 1.00 | 6.21 | 4.10 |
| Shikra Chen et al. (2023b) | 2.40 | 1.85 | 3.60 | 2.35 | 2.00 | 2.00 | 8.50 | 2.67 | 52.00 | 3.50 | 9.05 | 5.38 |
| SPHINX Lin et al. (2023) | 4.80 | 6.31 | 7.20 | 6.17 | 2.00 | 0.67 | 15.00 | 13.33 | **53.00** | 7.00 | 12.74 | 9.14 |
| LLaVA-v1.5 Liu et al. (2023e) | 8.00 | 7.38 | 10.00 | 8.09 | 1.33 | 2.00 | 23.00 | 13.33 | 50.00 | 12.00 | 15.26 | 11.33 |
| CogVLM Wang et al. (2023b) | 9.60 | 12.46 | 17.60 | 12.96 | 3.33 | 1.33 | 26.00 | 14.67 | 23.00 | 13.00 | 13.68 | 13.29 |
| ChartLlama Han et al. (2023) | 18.40 | 16.77 | 15.60 | 16.87 | 5.33 | 6.67 | 21.50 | 24.67 | 29.00 | 23.50 | 18.32 | 17.52 |
| Qwen-VL-Chat Bai et al. (2023b) | 26.00 | 19.69 | 31.20 | 23.57 | 6.00 | 7.33 | 26.00 | 29.33 | 23.00 | 30.50 | 21.05 | 22.43 |
| Mini-Gemini Li et al. (2024) | 24.00 | 19.85 | **42.00** | 25.57 | 8.67 | 10.67 | **33.00** | 27.33 | 46.00 | 31.50 | 25.79 | 25.67 |
| CogAgent Hong et al. (2023) | 39.20 | 18.92 | 34.00 | 26.61 | 3.33 | 11.33 | 27.50 | 50.67 | 21.00 | 35.50 | 25.79 | 26.24 |
| ChartVLM Xia et al. (2024) | **66.80** | **38.62** | 34.00 | **43.74** | 6.67 | 12.67 | 19.00 | 17.33 | 27.00 | 26.50 | 18.21 | 32.19 |
| DocOwl-v1.5 Hu et al. (2024) | 51.60 | 34.15 | 31.20 | 37.30 | 12.67 | **20.67** | 30.50 | 39.33 | 44.00 | 33.00 | 29.47 | 33.76 |
| Internlm-XComposer-v2 Dong et al. (2024) | 58.40 | 37.69 | 32.00 | 40.96 | **16.67** | 1.33 | 26.50 | **56.67** | 42.00 | **46.50** | 31.58 | **36.71** |
| *Closed source MLLMs* | | | | | | | | | | | | |
| ERNIE BaiDu | 36.00 | 19.23 | 32.42 | 25.74 | 5.32 | 13.33 | 20.00 | 60.00 | **100.00** | 30.00 | 33.47 | 29.24 |
| GPT-4V OpenAI (2023) | 48.00 | 24.62 | **40.00** | 33.04 | 6.67 | 26.67 | 25.00 | 66.67 | 80.00 | 50.00 | 40.00 | 36.19 |
| GPT-4O OpenAI (2023) | **72.00** | **29.00** | 36.00 | **40.00** | **7.00** | **47.00** | **35.00** | **73.00** | 20.00 | **60.00** | **41.05** | **40.48** |

**Fixed Chart CoT.** In Fig. 4, we mention using a fixed template for CoT, with detailed content shown in Fig. 8. Thanks to the expanded chart types, we can summarize some common approaches to understanding each type of chart. For example, we can identify the main subject of the question and the objects being queried, then guide the model to focus on the locations and spatial relationships of these objects. Although we cannot specify the exact logical relationships between these elements (as they depend on the specific content of each chart), guiding the model to prioritize commonly occurring logic can still enhance overall performance.

**Chart Resolution.** The visual branch of MLLMs typically scales images to a fixed pixel size, e.g., Qwen-VL-Chat is 448px, and LLaVA-v1.5 is 336px by default. To investigate the impact of resolution, we select a part of annotated regular charts from ChartBench and adjust them to 5-level resolutions using *Matplotlib* while keeping the font size unchanged. We ensure that each resolution is clear and legible for humans. Fig. 9 illustrates the performance of Qwen-VL-Chat and LLaVA-v1.5 at different resolutions. As the resolution increases, the scaled annotations gradually become unreadable for OCR, resulting in a decline in MLLMs' performance. Qwen-VL-Chat exhibits larger performance drops than LLaVA-v1.5, indicating a greater reliance on OCR.

**Performance of Supervised Fine-tuned Models on General Question Answering.** The results in Tab. 8 demonstrate that supervised fine-tuning of existing MLLMs with a small amount of chart data labeled without data points can significantly enhance their performance on ChartBench. To further illustrate the scalability,

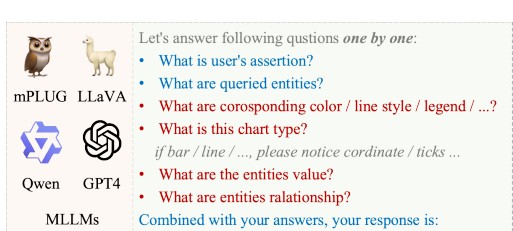

Figure 8: The proposed FixedCoT. Blue and red color questions indicate textual and visual reasoning, respectively.

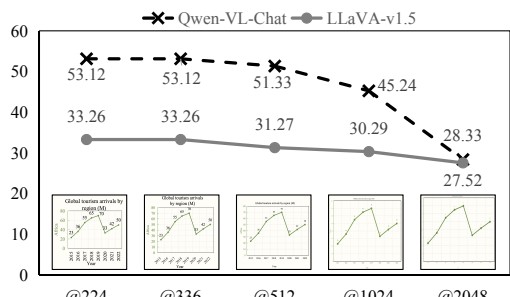

Figure 9: The zero-shot $Acc++$ (%) w.r.t. query chart resolution.

Table 16: Performance on general tasks. Results of InternLM-XC-v2+SFT (supervised fine-tuning with the ChartBench trainset split) on 6 public benchmarks. Data comes from the arxiv paper of InternLM-XC-v2. Evaluations are conducted using the scripts provided by InternLM-XC-v2's code repository. We only present results for benchmarks that could be evaluated locally due to time constraints. We adopt the DeepSeek-v2 API to replace the GPT4 for benchmarks that require LLM evaluation. Given the similarity in evaluation manners, the SFT version significantly improves the baseline on benchmarks like MME. Besides, the SFT version does not show any noticeable degradation in performance for descriptive evaluations like $LLaVA^W$.

| Method | LLM | $MME^P$ | $MME^C$ | MMB | $SEED^I$ | $LLaVA^W$ | $QBench^T$ |
|---|---|---|---|---|---|---|---|
| BLIP-2 | FLAN-T5 | 1,293.8 | 290.0 | - | 46.4 | 38.1 | - |
| InstructBLIP | Vicuna-7B | - | - | 36.0 | 53.4 | 60.9 | 55.9 |
| IDEFICS-80B | LLaMA-65B | - | - | 54.5 | 52.0 | 56.9 | - |
| Qwen-VL-Chat | Qwen-7B | 1,487.5 | 360.7 | 60.6 | 58.2 | 67.7 | 61.7 |
| LLaVA | Vicuna-7B | 807.0 | 247.9 | 34.1 | 25.5 | 63.0 | 54.7 |
| LLaVA-1.5 | Vicuna-13B | 1,531.3 | 295.4 | 67.7 | 68.2 | 70.7 | 61.4 |
| ShareGPT4V | Vicuna-7B | 1,567.4 | 376.4 | 68.8 | 69.7 | 72.6 | - |
| CogVLM-17B | Vicuna-7B | - | - | 65.8 | 68.8 | 73.9 | - |
| LLaVA-XTuner | InernLM2-20B | - | - | 75.1 | 70.2 | 63.7 | - |
| Monkey-10B | Qwen-7B | 1,522.4 | 401.4 | 72.4 | 68.9 | 33.5 | - |
| InternLM-XC | InernLM-7B | 1,528.4 | 391.1 | 74.4 | 66.1 | 53.8 | 64.4 |
| InternLM-XC-v2 | InernLM2-7B | 1,712.0 | 530.7 | 79.6 | 75.9 | 81.8 | 72.5 |
| InternLM-XC-v2+SFT | InernLM2-7B | 1,743.0 | 533.6 | 79.4 | 76.6 | 82.4 | 73.2 |
| Performance Gain | – | +31.0 | +2.9 | -0.2 | +0.7 | +0.6 | +0.7 |

applicability, and effectiveness of the proposed data, Tab. 16 presents the performance of SFT models on general capability benchmarks. Notably, the fine-tuning results in minimal general capability loss, while achieving significant performance improvements on ChartBench with around only 30K fine-tuning data points.

**Performance of Proposed Methods on Other Chart Tasks.** In Tab. 7 & 8, we provide how the CoT and SFT ameliorate model performance on ChartBench. In Tab. 17, we further provide the model performance of proposed methods on ChartQA. As illustrated in this table, Internlm-XC-v2 achieves remarkable improvements with our proposals. The Internlm-XC-v2 SFT version achieved a 3.1% overall increase and a 5.44% boost on the augmented part. This demonstrates the effectiveness of the ChartBench training set.

Table 17: Performance on other chart tasks. Results on ChartQA&ChartX for our proposals, i.e., CoT strategies and InternLM-XC-v2+SFT, supervised fine-tuning with the ChartBench trainset. Note that none of the methods use ChartQA's train set. The SFT improves the performance of the InternLM-XC-v2 by 3.1%, especially the *Augmented* part, which increased significantly by 5.44%. This demonstrates the versatility of ChartBench in improving MLLM chart understanding.

| Dataset | InternLM-XC-v2 | CoT-fix | Δ | CoT-self | Δ | CoT-GPT | Δ | InternLM-XC-v2+SFT | Δ |
|---|---|---|---|---|---|---|---|---|---|
| ChartQA$_{Human}$ | 62.72 | 63.12 | +0.40 | 61.06 | -1.66 | 63.33 | +0.61 | 63.48 | +0.76 |
| ChartQA$_{Augmented}$ | 81.28 | 83.14 | +1.86 | 81.44 | +0.16 | 84.52 | +3.24 | 86.72 | +5.44 |
| ChartQA$_{Average}$ | 72.00 | 73.13 | +1.13 | 71.25 | -0.75 | 73.93 | +1.93 | 75.10 | +3.10 |
| ChartX$_{QA}$ | 42.18 | - | - | - | - | - | - | 53.35 | +11.17 |

Table 18: The zero-shot *Accuracy* (%) performance w.r.t. chart types in ChartBench. We report the results of the best-performing prompt for each MLLM.

| Models | Regular Type | | | | Extra Type | | | | | | | Avg. |
|---|---|---|---|---|---|---|---|---|---|---|---|---|
| | Line | Bar | Pie | Avg. | Area | Box | Radar | Scatter | Node | Combin. | Avg. | |
| *Open source MLLMs* | | | | | | | | | | | | |
| mPLUG-Owl-bloomz Ye et al. (2023b) | 56.55 | 49.87 | 49.19 | 51.26 | 46.75 | 48.50 | 50.44 | 47.58 | 47.25 | 49.06 | 48.47 | 49.94 |
| Shikra Chen et al. (2023b) | 50.35 | 50.75 | 50.00 | 50.52 | 51.33 | 50.17 | 49.94 | 47.17 | 47.75 | 49.94 | 49.53 | 50.05 |
| MiniGPT-v2 Chen et al. (2023a) | 52.80 | 50.21 | 48.88 | 50.56 | 50.25 | 52.67 | 52.25 | 53.92 | 39.00 | 54.25 | 51.29 | 50.79 |
| ChartVLM Xia et al. (2024) | 53.60 | 51.77 | 50.75 | 52.00 | 51.67 | 51.83 | 50.19 | 51.08 | 45.50 | 51.25 | 50.83 | 51.34 |
| VisualGLM Du et al. (2022) | 55.40 | 50.98 | 49.69 | 51.75 | 47.92 | 53.75 | 49.00 | 49.00 | 55.75 | 52.44 | 51.01 | 51.37 |
| OneChart Chen et al. (2024a) | 55.15 | 54.50 | 53.81 | 54.52 | 51.92 | 51.00 | 49.56 | 51.42 | **70.25** | 52.44 | 52.29 | 54.02 |
| InstructBLIP Dai et al. (2023) | 62.15 | 56.33 | 59.13 | 58.16 | 48.50 | 50.08 | 47.50 | 55.50 | 54.25 | 47.19 | 49.97 | 54.45 |
| SPHINX Lin et al. (2023) | 55.40 | 53.40 | 52.25 | 53.65 | 53.00 | 51.25 | 54.50 | 53.75 | 62.75 | 58.50 | 55.34 | 54.51 |
| LLaVA-v1.5 Liu et al. (2023d) | 60.00 | 54.58 | 47.06 | 54.44 | 57.67 | 54.92 | 58.63 | 55.58 | 48.00 | 55.38 | 55.61 | 54.75 |
| Internlm-XComposer Zhang et al. (2023) | 55.70 | 57.38 | 55.31 | 56.62 | 51.17 | 54.42 | 53.50 | 54.08 | 54.00 | 51.25 | 52.95 | 54.96 |
| CogVLM Wang et al. (2023b) | 54.40 | 56.27 | 56.50 | 55.89 | 55.50 | 53.08 | 55.25 | 56.42 | 55.25 | 51.19 | 54.28 | 55.26 |
| BLIP2 Li et al. (2023c) | 62.80 | 57.33 | 60.38 | 59.13 | 52.42 | 53.58 | 56.69 | 58.58 | 41.00 | 61.44 | 55.17 | 57.45 |
| CogAgent Hong et al. (2023) | 59.30 | 61.96 | 55.50 | 60.18 | 57.83 | 57.83 | 54.44 | 55.42 | 31.00 | 57.50 | 55.11 | 57.45 |
| ChartLlama Han et al. (2023) | 61.70 | 56.48 | 57.67 | 57.85 | 57.25 | 52.75 | 61.31 | 61.50 | 39.75 | 60.69 | 56.95 | 57.54 |
| Qwen-VL-Chat Bai et al. (2023b) | 69.00 | 57.77 | 66.50 | 61.91 | **63.17** | 57.50 | 63.62 | 56.75 | 55.50 | 58.63 | 59.59 | 61.11 |
| DocOwl-v1.5 Hu et al. (2024) | 72.65 | 62.92 | 63.44 | 65.23 | 52.42 | 54.67 | 52.81 | 65.17 | 52.50 | **66.25** | 58.08 | 61.83 |
| Mini-Gemini Li et al. (2024) | 65.15 | 65.42 | 66.12 | 65.49 | 62.75 | 57.33 | 58.38 | 61.67 | 66.25 | 55.81 | 59.35 | 62.86 |
| Internlm-XComposer-v2 Dong et al. (2024) | 84.30 | 73.83 | 79.00 | 77.15 | 57.83 | 60.50 | 67.44 | 73.58 | 67.00 | 66.00 | **65.36** | 72.23 |
| *Closed source MLLMs* | | | | | | | | | | | | |
| ERNIE BaiDu | 61.00 | 65.58 | 71.25 | 65.57 | **68.33** | 52.50 | 65.62 | 68.33 | 82.50 | 73.12 | 67.76 | 66.67 |
| GPT-4V OpenAI (2023) | 84.50 | 68.08 | 78.75 | 73.75 | 62.50 | 65.83 | **69.38** | 82.50 | 100.00 | 73.12 | 73.82 | 74.11 |
| GPT-4O OpenAI (2023) | **90.50** | 70.58 | 82.50 | 77.27 | 61.67 | 77.50 | 67.50 | **91.67** | 100.00 | 79.38 | 77.89 | 78.10 |

Table 19: The zero-shot *Accuracy* (%) performance w.r.t. chart tasks in ChartBench. We report the results of the best-performing prompt for each MLLM.

| Models | Task Type | | | | | Avg. |
|---|---|---|---|---|---|---|
| | CR | VE | VC | GC | NQA | |
| *Open source MLLMs* | | | | | | |
| mPLUG-Owl-bloomz Ye et al. (2023b) | 50.43 | 50.05 | 49.83 | 49.45 | 4.10 | 40.77 |
| Shikra Chen et al. (2023b) | 49.98 | 50.31 | 50.14 | 49.79 | 5.38 | 41.12 |
| MiniGPT-v2 Chen et al. (2023a) | 53.67 | 49.57 | 50.95 | 48.98 | 3.71 | 41.38 |
| VisualGLM Du et al. (2022) | 55.88 | 49.83 | 49.90 | 49.86 | 3.19 | 41.73 |
| OneChart Chen et al. (2024a) | 50.88 | 56.55 | 54.43 | 54.21 | 2.76 | 43.77 |
| InstructBLIP Dai et al. (2023) | 67.90 | 50.00 | 49.95 | 49.95 | 2.90 | 44.14 |
| Internlm-XComposer Zhang et al. (2023) | 70.76 | 49.43 | 50.76 | 48.90 | 3.29 | 44.63 |
| SPHINX Lin et al. (2023) | 64.21 | 50.71 | 53.02 | 50.07 | 9.14 | 45.43 |
| LLaVA-v1.5 Liu et al. (2023d) | 65.98 | 48.93 | 54.29 | 49.81 | 11.33 | 46.07 |
| BLIP2 Li et al. (2023c) | 78.57 | 48.88 | 53.48 | 48.86 | 2.71 | 46.50 |
| CogVLM Wang et al. (2023b) | 64.07 | 49.98 | 54.57 | 52.40 | 13.29 | 46.86 |
| ChartVLM Xia et al. (2024) | 50.00 | 51.79 | 51.95 | 51.62 | 32.19 | 47.51 |
| ChartLlama Han et al. (2023) | 71.95 | 50.45 | 55.17 | 52.57 | 17.52 | 49.53 |
| CogAgent Hong et al. (2023) | 81.12 | 48.64 | 51.45 | 48.57 | 26.24 | 51.20 |
| Qwen-VL-Chat Bai et al. (2023b) | 73.02 | 53.43 | 58.86 | 59.14 | 22.43 | 53.38 |
| Mini-Gemini Li et al. (2024) | **88.95** | 52.17 | 55.48 | 54.83 | 25.67 | 55.42 |
| DocOwl-v1.5 Hu et al. (2024) | 62.95 | 63.60 | 58.69 | 62.07 | 33.76 | 56.21 |
| Internlm-XComposer-v2 Dong et al. (2024) | 83.41 | **65.49** | **68.49** | **71.54** | **36.71** | **65.13** |
| *Closed source MLLMs* | | | | | | |
| ERNIE BaiDu | 75.00 | 67.14 | 53.57 | 70.95 | 16.19 | 56.57 |
| GPT-4V OpenAI (2023) | 97.62 | 62.86 | 65.95 | 70.00 | 36.19 | 66.52 |
| GPT-4O OpenAI (2023) | **98.33** | 65.71 | **74.29** | **74.05** | **40.48** | **70.57** |

## E.2 RESULTS OF ACCURACY METRIC

Accuracy is the most widely used evaluation criterion for true/false or multiple-choice questions, but it has inherent limitations. Firstly, for difficult questions, accuracy struggles to distinguish between genuine answers and random guesses, both of which can yield performance close to the baseline (e.g., 50% for true/false questions, 25% for four-choice questions). Secondly, accuracy places high demands on data scale. In the case of the accuracy metric, if the test sample approaches infinity, the performance of random guessing would converge to the baseline. Conversely, with a small data scale, random guessing might produce results significantly higher than the baseline. Although ChartBench provides *16.8K* judgment QA pairs (consisting of *8.4K* original questions and their counterparts), this quantity still cannot completely eliminate the occurrence of the situations above (e.g., the accuracy of MiniGPT-v2 on Node chart in Tab. 18).

In Tab. 18 and Tab. 19, we present the results using Accuracy (abbreviated as *Acc.*) as the metric. Overall, Internlm-Xcomposer-v2 continues to demonstrate the best performance, consistent with the trend shown by *Acc++* in Tab. 3. However, there are differences between accuracy and *Acc++* in terms of specific details. InternLM-Xcomposer achieves 55.70% accuracy in Tab 18, while its *Acc++* performance is just 15.49% (Tab. 3), indicating that a significant portion of its correct answers are the result of random guessing. This is further

confirmed by the *CoR* metric in Tab. 5. From Tab. 19, it can be observed that accuracy does not effectively differentiate between tasks of varying difficulty, as it shows results close to the baseline of 50% across all 5 tasks. Compared with Tab. 4, it is evident that the VE and GC tasks are notably more challenging, as they require MLLMs to rely on more visual cues for reasoning. The above analysis demonstrates that the improved *Acc++* metric enables more robust evaluations.

Our improved metric, *Acc++*, effectively addresses the two limitations of accuracy mentioned above. The *Acc++* metric requires MLLMs to provide accurate judgments for both positive and negative perspectives regarding the base assertions. This innovative metric offers two distinct advantages. Firstly, it ensures consistency between positive and negative queries, with the only difference being the Ground Truth value. This precautionary approach reduces the chance of lucky guesses resulting from random choices, as MLLMs may produce identical responses for both query types if they fail to comprehend the chart. Secondly, the GT values for negative queries are derived from other data within the same chart, eliminating unrealistic scenarios and enhancing the validity of the evaluation process. Generally, the expected probability of random guessing is 25% for vanilla *Acc++*. However, for the MLLM that has insufficient chart recognition capabilities, *the CoR tends to be 100%, and thus the* Acc++ *tends to be 0% instead of 25% baseline*. This characteristic enables *Acc++* to accurately reflect the model's chart comprehension ability even when the dataset is small in size.

### E.3 RESULTS OF CHARTQA

Table 20: The zero-shot *Acc* (%) performance w.r.t. chart types in ChartQA. For bar chart, We report the average score of horizontal and vertical bars in ChartQA.

| Models | Human | | | | Augmented | | | | Acc. |
|---|---|---|---|---|---|---|---|---|---|
| | Line | Bar | Pie | Avg. | Line | Bar | Pie | Avg. | |
| BLIP2 Li et al. (2023c) | 14.34 | 9.69 | 7.24 | 10.40 | 6.20 | 5.18 | 0.00 | 5.20 | 7.80 |
| mPLUG-Owl-bloomz Ye et al. (2023b) | 22.79 | 10.53 | 6.58 | 12.72 | 7.75 | 5.72 | 5.00 | 5.92 | 9.32 |
| Shikra Chen et al. (2023b) | 25.00 | 13.68 | 13.82 | 16.16 | 8.53 | 7.27 | 0.00 | 7.28 | 11.72 |
| InstructBLIP Dai et al. (2023) | 29.78 | 11.86 | 10.53 | 15.60 | 10.08 | 9.81 | 10.00 | 9.84 | 12.72 |
| VisualGLM Du et al. (2022) | 32.35 | 14.89 | 7.24 | 17.76 | 9.30 | 7.81 | 5.00 | 7.92 | 12.84 |
| Internlm-XComposer Zhang et al. (2023) | 31.99 | 13.20 | 9.21 | 16.80 | 9.30 | 9.17 | 20.00 | 9.36 | 13.08 |
| MiniGPT-v2 Chen et al. (2023a) | 33.09 | 16.22 | 11.18 | 19.28 | 9.30 | 10.99 | 10.00 | 10.80 | 15.04 |
| SPHINX Lin et al. (2023) | 35.66 | 17.68 | 16.45 | 21.44 | 10.08 | 11.35 | 10.00 | 11.20 | 16.32 |
| LLaVA-v1.5 Liu et al. (2023d) | 39.71 | 19.01 | 16.45 | 23.20 | 9.30 | 14.26 | 15.00 | 13.76 | 18.48 |
| CogVLM Wang et al. (2023b) | 48.90 | 29.41 | 34.21 | 34.24 | 17.83 | 29.88 | 25.00 | 28.56 | 31.40 |
| Mini-Gemini Li et al. (2024) | 55.88 | 40.68 | 43.42 | 44.32 | 43.41 | 58.31 | 75.00 | 57.04 | 50.68 |
| Qwen-VL-Chat Bai et al. (2023b) | 54.41 | 38.38 | 43.42 | 42.48 | 55.04 | 77.48 | 80.00 | 75.20 | 58.84 |
| ChartVLM Xia et al. (2024) | 48.90 | 39.59 | 43.42 | 42.08 | 69.77 | 83.92 | 85.00 | 82.48 | 62.28 |
| OneChart Chen et al. (2024a) | - | - | - | 85.30 | - | - | - | 49.10 | 67.20 |
| CogAgent Hong et al. (2023) | 65.44 | 49.88 | 56.58 | 54.08 | 62.02 | 82.74 | 80.00 | 80.56 | 67.32 |
| DocOwl-v1.5 Hu et al. (2024) | 57.72 | 44.79 | 50.00 | 48.24 | 68.22 | 88.92 | 85.00 | 86.72 | 67.48 |
| Internlm-XComposer-v2 Dong et al. (2024) | 65.81 | **61.38** | **67.76** | 63.12 | 78.29 | 82.11 | 95.00 | 81.92 | 72.64 |
| ChartLlama Han et al. (2023) | **68.75** | 53.63 | 65.79 | 58.40 | **79.84** | **94.55** | **100.00** | **93.12** | **75.76** |

Table 21: The zero-shot *Acc++* (%) and *Acc* (%) performance in ChartBench and ChartQA respectively w.r.t *regular* chart types. We report the results of the best-performing prompt for each MLLM.

| Models | Line | | Bar | | Pie | | Avg. | |
|---|---|---|---|---|---|---|---|---|
| | ChartBench | ChartQA | ChartBench | ChartQA | ChartBench | ChartQA | ChartBench | ChartQA |
| Shikra Chen et al. (2023b) | 7.40 | 22.19 | 10.62 | 9.81 | 4.50 | 9.30 | 7.51 | 13.77 |
| MiniGPT-v2 Chen et al. (2023a) | 26.70 | 21.70 | 21.54 | 10.33 | 20.20 | 8.72 | 22.81 | 13.58 |
| VisualGLM Du et al. (2022) | 10.80 | 23.44 | 1.96 | 10.90 | 0.00 | 10.47 | 4.25 | 14.94 |
| SPHINX Lin et al. (2023) | 18.40 | 27.43 | 15.54 | 14.06 | 23.40 | 15.70 | 19.11 | 19.06 |
| InstructBLIP Dai et al. (2023) | 24.40 | 22.44 | 15.04 | 9.81 | 19.10 | 11.05 | 19.51 | 14.43 |
| LLaVA-v1.5 Liu et al. (2023d) | 34.40 | 29.68 | 24.73 | 15.31 | 19.10 | 18.60 | 26.08 | 21.20 |
| ChartLlama Han et al. (2023) | 28.90 | **72.32** | 19.35 | **77.01** | 22.10 | 69.77 | 23.45 | **73.03** |
| CogVLM Wang et al. (2023b) | 10.50 | 38.90 | 14.58 | 29.68 | 17.90 | 33.14 | 14.33 | 33.91 |
| Internlm-XComposer Zhang et al. (2023) | 16.00 | 16.96 | 20.42 | 9.24 | 21.50 | 9.89 | 19.30 | 12.03 |
| BLIP2 Li et al. (2023c) | 29.60 | 18.20 | 17.35 | 8.35 | 24.90 | 5.81 | 23.95 | 10.79 |
| mPLUG-Owl-bloomz Ye et al. (2023b) | 37.50 | 10.47 | 24.73 | 5.81 | 26.10 | 2.91 | 29.44 | 6.40 |
| Qwen-VL-Chat Bai et al. (2023b) | 41.00 | 54.61 | 20.96 | 60.72 | 40.00 | 47.67 | 33.99 | 54.33 |
| Mini-Gemini Li et al. (2024) | 37.60 | 51.87 | 40.19 | 50.75 | 40.00 | 47.09 | 39.57 | 49.90 |
| ChartVLM Xia et al. (2024) | 10.70 | 55.61 | 8.04 | 64.92 | 4.62 | 48.26 | 8.02 | 56.26 |
| DocOwl-v1.5 Hu et al. (2024) | 49.10 | 61.10 | 31.08 | 70.01 | 31.62 | 54.07 | 35.27 | 61.73 |
| Internlm-XComposer-v2 Dong et al. (2024) | **70.60** | 69.83 | **51.50** | 73.22 | **62.75** | **70.93** | **57.89** | 71.33 |

ChartQA Masry et al. (2022) is a canonical benchmark utilized in prior research to appraise the competency of multimodal models to comprehend chart data. It comprises two subsets, namely *Human* and *Augmented*, and encompasses solely three chart types, viz., line, bar, and pie. To ascertain the indispensability of ChartBench and the rationality of our benchmark design and evaluation, we initially scrutinize the vanilla accuracy (*Acc.*) on ChartQA. We employ the test-split in ChartQA for evaluation, circumventing the prompt engineering process, and directly utilizing the original query without any modification as the prompt input to MLLMs. Thereafter,

we evaluate the correctness of the results utilizing rule-based and regular expression matching. For numerical questions, we employ the relax accuracy metric akin to ChartQA, signifying that the difference between the model's answer and the ground truth is within 5% to be regarded as correct. As tabulated in Tab. 20, we report the zero-shot *Acc* regarding chart types and dataset split. Conspicuously, for bar charts, we report the average accuracy of MLLMs on horizontal and vertical bars.

Tab. 20 evinces that despite the relatively simple chart understanding task with specific data point annotations in ChartQA, most of the MLLMs remain woefully deficient in this regard. However, it is evident that incorporating chart data in training augments the ability of MLLMs to comprehend charts, as demonstrated by the relatively superior performance of ChartLlama and Qwen-VL-Chat in Tab. 20. In contrast to the results in Tab. 18, which show a specific baseline, Tab. 20 does not converge to a baseline despite using basic accuracy as the evaluation metric. It is attributable to the question-answer pairs' design in ChartQA, which employs annotated metadata and open-ended answers instead of the binary yes/no format. While this design ostensibly appears to appraise the model's ability to comprehend charts, we contend that it is fraught with several inconveniences. 1) open-ended answers render the verification of MLLM's correctness excessively laborious, sometimes necessitating third-party (human or GPT) intervention. However, the ChartBench design we propose only necessitates the model to answer yes/no, streamlining the judgment process while enhancing efficiency and accuracy. 2) the chart data in ChartQA entail specific numerical annotations, which may prompt MLLMs to rely solely on OCR-based visual judgments instead of utilizing other implicit information in the chart (e.g., color coordinates and legends) for logical inference. This inevitably reduces the complexity of tasks. The performance of ChartLlama in Tab. 18 & 20 clearly illustrates ChartQA's predisposition to MLLMs that rely heavily on OCR. 3) ChartQA's design constraints necessitate the utilization of less-convincing metrics such as vanilla accuracy and BLEU score to assess MLLMs' ability to comprehend charts.

## E.4 RESULTS OF HUMAN EVALUATION

Table 22: Human evaluation results on the ChartBench via random questionnaire. We provide the performance of Qwen-VL-Chat (open-sourced) and GPT-4V (closed-sourced) for easy comparisons.

| Models | Regular Type | | | Extra Type | | | | | | Acc++ |
|---|---|---|---|---|---|---|---|---|---|---|
| | Line | Bar | Pie | Area | Box | Radar | Scatter | Node | Combin. | |
| Internlm-XComposer-v2 | 70.60 | 51.50 | 62.75 | 30.17 | 31.33 | 43.50 | 52.00 | 52.50 | 46.12 | 51.34 |
| GPT-4V | 74.00 | 41.54 | 63.00 | 33.30 | 46.67 | 57.50 | 70.00 | 100.00 | 56.25 | 54.39 |
| Human Evaluation | 90.63 | 88.69 | 87.86 | 86.61 | 84.56 | 89.86 | 89.29 | 88.75 | 85.64 | 88.46 |

| Models | Task Type (*Acc++*) | | | | | Task Type (*CoR*) | | | | |
|---|---|---|---|---|---|---|---|---|---|---|
| | CR | VE | VC | GC | ALL | CR | VE | VC | GC | ALL |
| Internlm-XComposer-v2 | 68.29 | 36.63 | 54.63 | 45.80 | 51.34 | 30.24 | 57.71 | 27.71 | 51.46 | 41.78 |
| GPT-4V | 96.10 | 29.27 | 47.32 | 44.88 | 54.39 | 2.93 | 64.88 | 35.61 | 48.78 | 38.05 |
| Human Evaluation | 93.68 | 84.56 | 88.68 | 86.91 | 88.46 | 1.34 | 5.82 | 4.72 | 3.52 | 3.85 |

The motivation behind ChartBench is to evaluate the understanding capability of MLLMs regarding charts. While MLLMs have exhibited high performance on previous benchmarks, they still encounter significant hallucination issues in practical applications due to the unreliable nature of the data they extract from charts. ChartBench aims to truly reflect MLLM's ability to interpret visual data and approach or even surpass human-level performance. Therefore, we have provided evaluation results of human performance on ChartBench.

To ensure a fair and objective evaluation, we conduct an online survey, which consists of 10 randomly selected subcategories from ChartBench for each questionnaire. 1 chart and 4 assertions are selected from each sub-category for respondents to assess their accuracy. To obtain reliable evaluation results, the survey participants mainly consist of undergraduate and graduate students with chart reading ability, as well as other researchers in the campus and company. We encourage participants to use large-screen devices for better chart display and kindly request their patient and diligent responses. On average, it takes approximately *15 minutes* and *17 seconds* to complete each survey. To avoid cases of random guessing, we still employ the *Acc++* evaluation metric. Incomplete responses are discarded, and we ensure that each subcategory has valid answers. In total, we have collected 68 valid surveys.

Tab. 22 presents the results of human evaluations, revealing some insightful observations. Firstly, the VE task appears to be more challenging compared to other tasks. The human eye faces challenges in determining the values of unmarked data points. While the coordinate system offers potential inference, excessively fine granularity can diminish respondents' confidence. Secondly, there is not a significant variation in human performance across different chart types. Once individuals grasp the correct interpretation methods for charts, they can demonstrate similar proficiency across each chart category. Thirdly, even in some relatively straightforward tasks, such as identifying chart types, humans are unable to achieve 100% accuracy. This limitation could be attributed to constraints within our survey methodology. For instance, certain descriptions may have confused the respondents, or the length of the test might have led to hastily completed surveys.

## E.5 CASE STUDY OF CHARTBENCH

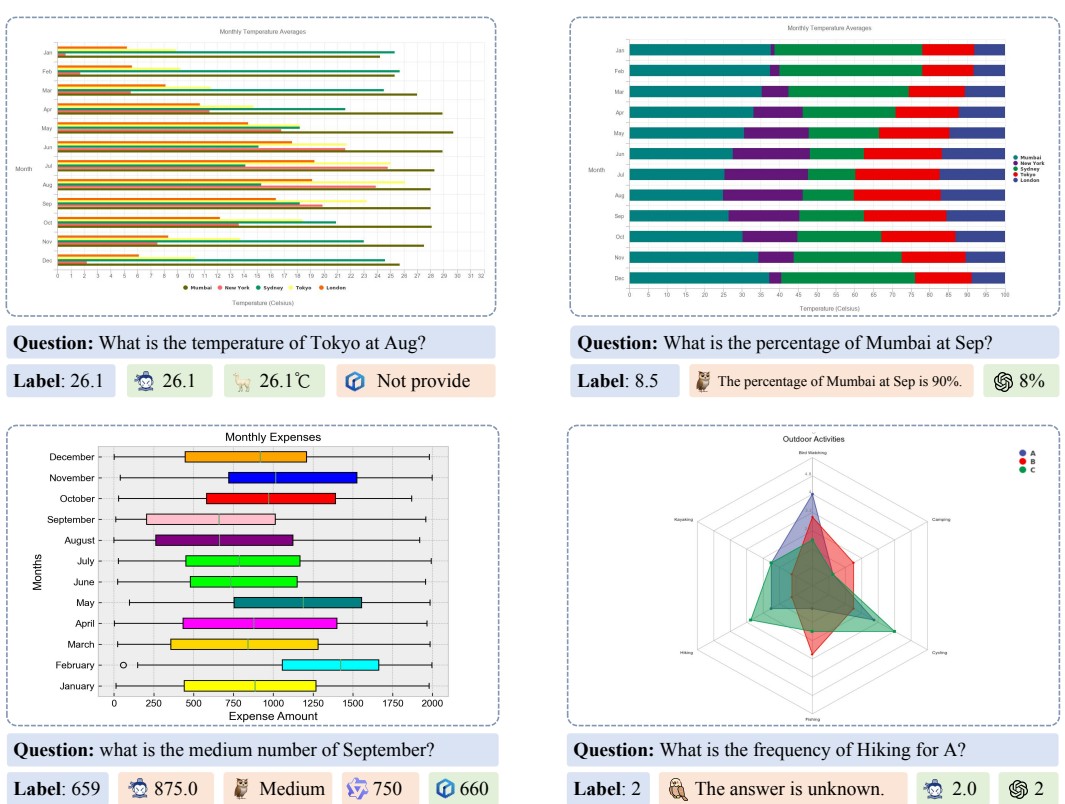

Figure 10: NQA cases with unannotated charts from the ChartBench Test Split. Red indicates incorrect answers, and green indicates correct answers.

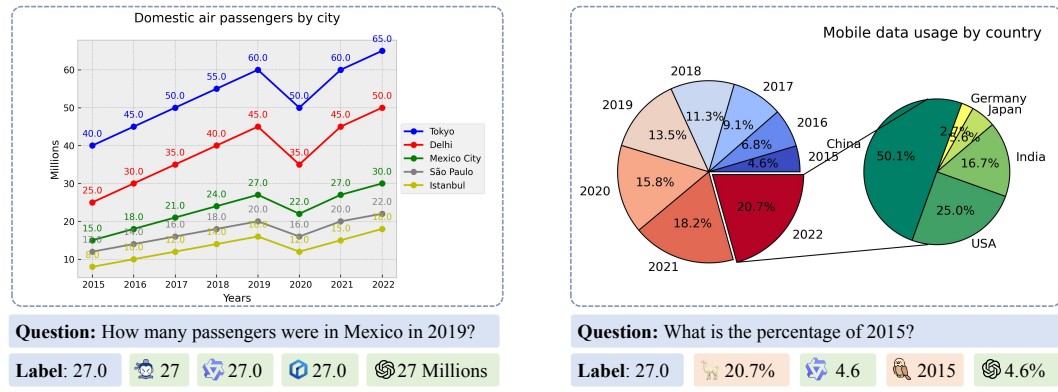

Figure 11: NQA cases with annotated charts from the ChartBench Test Split. Red indicates incorrect answers, and green indicates correct answers.

Fig. 10 & 11 visualize some NQA task examples with/without annotations from the ChartBench test split. We adopt ChartQA's relaxed matching and correct it from two perspectives: 1) Considering the robustness issue of string matching, we only ask questions about numbers. 2) To account for a 5% margin of error, we avoid questions about entities like years or months. Considering the varying instruction-following capabilities of different models, we use LLMs to extract numerical values from the model responses.

## E.6   Case study of GPT-4

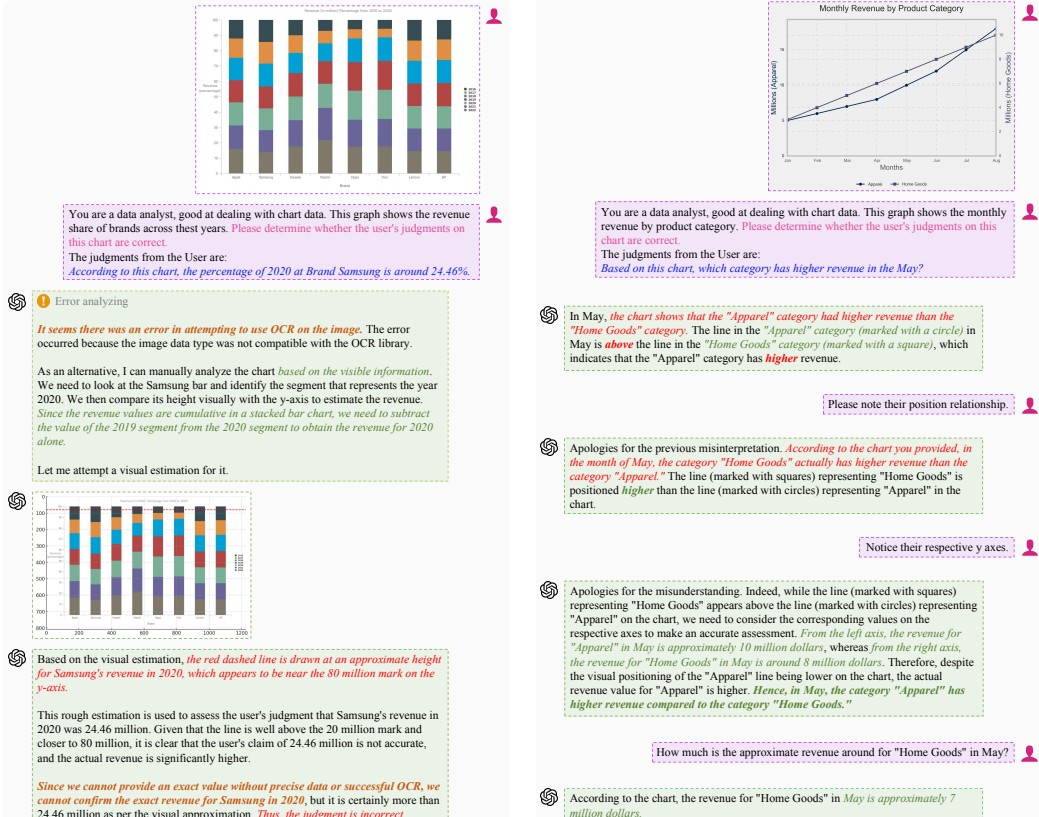

(a) GPT-4V combines several APIs (e.g., OCR), for chart understanding but fails on unannotated charts.

(b) The GPT-4V requires multiple manual instructions to achieve the correct answer for unannotated charts.

Figure 12: Specific examples of GPT-4V in chart comprehension. Pink: user requirement. Blue: user assertion. Orange: GPT-4V ensembles APIs to assist chart comprehension. Green: the correct visual clues. Red: the misperceptions or misjudgments.

As the top-performing proprietary model, Fig. 12 showcases some characteristics of GPT-4V in chart comprehension. Firstly, GPT-4V goes beyond a single end-to-end MLLM by integrating multiple APIs to aid in chart cognition (highlight orange in Fig. 12a). The performance of GPT-4V is inherently influenced by the output of these APIs, thereby imposing limitations. For instance, when OCR results are unavailable, its ability to interpret visual information significantly declines. Secondly, GPT-4V can proactively acknowledge its limitations, such as recognizing its inability to determine specific values solely based on visual information. Thirdly, while GPT-4V possesses strong chart comprehension capabilities, it requires multi-step guidance from humans (Fig. 12b). This accounts for its shortcomings in zero-shot performance on ChartBench.

## F   Ethical Statement

This study upholds rigorous ethical standards to ensure the credibility and confidentiality of the findings. All data underwent thorough de-identification procedures to protect privacy and maintain anonymity. The study followed ethical guidelines and obtained informed consent from participants while prioritizing their rights and autonomy. Transparency and accountability were maintained throughout the research process to minimize biases and conflicts of interest. No academic ethical issues or misconduct were encountered, and the authors affirm their unwavering commitment to upholding ethical research practices and promptly addressing any unintentional errors or oversights.

# G  LEADERBOARDS

In this section, we devise several leaderboards to evaluate the performance of diverse MLLMs across multiple task types to obtain a more nuanced insight into their perceptual capacities in the context of varied chart categories.

In Tab. 23 & 24 & 25 & 26, we present the leaderboards of MLLMs on ChartBench, which includes **3** regular types of charts and **6** extra types of charts, utilizing the $Acc++$ metric. Additionally, we showcase the $Acc++$ and $CoR$ leaderboards of MLLMs for **4** chart comprehension tasks while also displaying their rankings on *w/i* and *w/o* annotation data.

## G.1  LEADERBOARDS ON CHART TYPE

Tab. 23 presents an overview of MLLMs' performance across various chart types, along with the overall $Acc++$ metric. Generally, the current MLLMs exhibit a constrained ability in chart recognition, encountering notable challenges. For specific chart types, such as radar or combination charts, certain MLLMs achieve close to 0% in $Acc++$, signaling their difficulty in extracting crucial information from charts and their insensitivity to both positive and negative queries. It's essential to highlight that the $Acc++$ metric tends toward 0% in situations of random guessing, as elaborated in Sec. 3.4. Particularly, Qwen-VL-Chat and mPLUG-Owl-bloomz showcase commendable proficiency in recognizing charts, a capability likely attributed to their precise tuning with chart data. However, their performance in this aspect falls below what has been reported in ChartQA. This discrepancy can be traced back to their reliance on OCR skills rather than robust visual logical reasoning. In the context of ChartBench, where the proportion of annotated charts is notably low, these models face a significant challenge. The majority of queries in ChartBench necessitate MLLMs to employ visual logical reasoning, a task that proves quite demanding for models like Qwen-VL-Chat and mPLUG-Owl-bloomz. On the other hand, VisualGLM and Shikra exhibit subpar performance, potentially due to their smaller LLM size and less robust visual encoding branch. While MLLMs generally demonstrate satisfactory performance on regular charts, there remains considerable room for improvement, particularly in handling more intricate graphics.

## G.2  LEADERBOARDS ON TASK TYPE

Tab. 24 outlines the performance of MLLMs on perception and conception tasks introduced in Sec. 3.2. Most MLLMs exhibit notable success in the CR task, showcasing their proficiency in recognizing fundamental chart types. Notably, LLaVA-v1.5, mPLUG-Owl-bloomz, and Qwen-VL-Chat demonstrate substantial advantages in the VC and GC conception tasks, leveraging their chart-tuned data. The most challenging task, VE, serves as a key distinction between ChartBench and ChartQA. Unlike basic OCR, the VE task requires a series of visual and textual logical reasoning steps to arrive at the correct answer. Despite strong overall performance, models such as BLIP2 and ChartLlama face difficulties in the VE task. This underscores the importance of prioritizing and enhancing the visual logical reasoning capabilities of these MLLMs. In terms of model comparison, closed-source models outperform their open-source counterparts, partly attributed to their larger model size and broader data coverage.

## G.3  LEADERBOARDS ON *CoR* METRIC

Tab. 25 showcases the *CoR* metric, which signifies the portion of the chart that the MLLM fails to comprehend entirely. Qwen-VL-Chat exhibits the highest $Acc++$, albeit with a lower *CoR* compared to models like MiniGPT-v2. The top-performing MiniGPT-v2 demonstrates a *CoR* of 55.06%, underscoring the prevalence of random guessing cases for open-source models due to their challenges in accurately understanding charts. In the case of closed-source MLLMs, although GPT-4V outperforms ERNIE in terms of $Acc++$, their *CoR* values are similar. A more detailed examination reveals that ERNIE excels in challenging VE tasks, which happen to be the weaker area for GPT-4V.

## G.4  LEADERBOARDS ON WITH/WITHOUT ANNOTATED CHARTS

The rationale behind ChartBench is to assess the comprehension of unlabeled charts by MLLMs. In Tab. 26, the performance of all MLLMs on both annotated and unannotated charts is presented. It is important to note that: 1) Virtually all models exhibit significantly superior performance on annotated charts when compared to unannotated ones. This discrepancy arises because MLLMs heavily depend on OCR to acquire answer candidates, thereby enhancing answer accuracy—an advantage not applicable to unannotated charts. 2) The larger the performance gap between models, such as Qwen-VL-Chat (+16.00%) and GPT-4V (+31.39%), the more favorable their overall performance. This suggests that the $Acc++$ of MLLMs is primarily elevated by annotated charts, while unannotated charts notably intensify the challenge presented by ChartBench.

| No. | Model | Acc++ |
|---|---|---|
| 1 | GPT-4O | 86.00 |
| 2 | GPT-4V | 74.00 |
| 3 | InternLM-v2 | 70.60 |
| 4 | DocOwl-v1.5 | 49.10 |
| 5 | ERNIE | 44.00 |
| 6 | Qwen-VL | 41.00 |
| 7 | Mini-Gemini | 37.60 |
| 8 | mPLUG-Owl | 37.50 |
| 9 | LLaVA-v1.5 | 34.40 |
| 10 | BLIP2 | 29.60 |
| 11 | ChartLlama | 28.90 |
| 12 | MiniGPT-v2 | 26.70 |
| 13 | InstructBLIP | 24.40 |
| 14 | CogAgent | 18.60 |
| 15 | SPHINX | 18.40 |
| 16 | InternLM | 16.00 |
| 17 | OneChart | 15.10 |
| 18 | VisualGLM | 10.80 |
| 19 | ChartVLM | 10.70 |
| 20 | CogVLM | 10.50 |
| 21 | Shikra | 7.40 |

(a) *Line Chart*

| No. | Model | Acc++ |
|---|---|---|
| 1 | GPT-4O | 51.92 |
| 2 | InternLM-v2 | 51.50 |
| 3 | ERNIE | 45.00 |
| 4 | GPT-4V | 41.54 |
| 5 | Mini-Gemini | 40.19 |
| 6 | DocOwl-v1.5 | 31.08 |
| 7 | LLaVA-v1.5 | 24.73 |
| 8 | mPLUG-Owl | 24.73 |
| 9 | CogAgent | 23.96 |
| 10 | MiniGPT-v2 | 21.54 |
| 11 | Qwen-VL | 20.96 |
| 12 | InternLM | 20.42 |
| 13 | ChartLlama | 19.35 |
| 14 | BLIP2 | 17.35 |
| 15 | SPHINX | 15.54 |
| 16 | InstructBLIP | 15.04 |
| 17 | CogVLM | 14.58 |
| 18 | OneChart | 12.27 |
| 19 | Shikra | 10.62 |
| 20 | ChartVLM | 8.04 |
| 21 | VisualGLM | 1.96 |

(b) *Bar Chart*

| No. | Model | Acc++ |
|---|---|---|
| 1 | GPT-4O | 78.00 |
| 2 | GPT-4V | 63.00 |
| 3 | InternLM-v2 | 62.75 |
| 4 | ERNIE | 57.00 |
| 5 | Qwen-VL | 40.00 |
| 6 | Mini-Gemini | 40.00 |
| 7 | DocOwl-v1.5 | 31.62 |
| 8 | mPLUG-Owl | 26.10 |
| 9 | BLIP2 | 24.90 |
| 10 | SPHINX | 23.40 |
| 11 | ChartLlama | 22.10 |
| 12 | InternLM | 21.50 |
| 13 | MiniGPT-v2 | 20.20 |
| 14 | InstructBLIP | 19.10 |
| 15 | LLaVA-v1.5 | 19.10 |
| 16 | CogVLM | 17.90 |
| 17 | CogAgent | 11.00 |
| 18 | OneChart | 9.12 |
| 19 | ChartVLM | 4.62 |
| 20 | Shikra | 4.50 |
| 21 | VisualGLM | 0.00 |

(c) *Pie Chart*

| No. | Model | Acc++ |
|---|---|---|
| 1 | ERNIE | 45.00 |
| 2 | Mini-Gemini | 36.83 |
| 3 | GPT-4O | 36.67 |
| 4 | GPT-4V | 33.30 |
| 5 | InternLM-v2 | 30.17 |
| 6 | Qwen-VL | 28.83 |
| 7 | LLaVA-v1.5 | 26.83 |
| 8 | MiniGPT-v2 | 21.67 |
| 9 | mPLUG-Owl | 21.33 |
| 10 | ChartLlama | 16.50 |
| 11 | CogAgent | 15.67 |
| 12 | CogVLM | 12.50 |
| 13 | DocOwl-v1.5 | 12.17 |
| 14 | SPHINX | 12.00 |
| 15 | ChartVLM | 7.67 |
| 16 | OneChart | 7.00 |
| 17 | BLIP2 | 6.17 |
| 18 | Shikra | 6.00 |
| 19 | InternLM | 4.50 |
| 20 | InstructBLIP | 4.33 |
| 21 | VisualGLM | 1.17 |

(d) *Area Chart*

| No. | Model | Acc++ |
|---|---|---|
| 1 | GPT-4O | 63.33 |
| 2 | GPT-4V | 46.67 |
| 3 | InternLM-v2 | 31.33 |
| 4 | ERNIE | 30.00 |
| 5 | Mini-Gemini | 26.50 |
| 6 | mPLUG-Owl | 25.83 |
| 7 | LLaVA-v1.5 | 25.67 |
| 8 | MiniGPT-v2 | 24.67 |
| 9 | Qwen-VL | 24.17 |
| 10 | DocOwl-v1.5 | 24.00 |
| 11 | CogAgent | 16.50 |
| 12 | InternLM | 14.50 |
| 13 | ChartLlama | 13.33 |
| 14 | Shikra | 11.33 |
| 15 | BLIP2 | 10.67 |
| 16 | CogVLM | 9.67 |
| 17 | VisualGLM | 8.50 |
| 18 | SPHINX | 8.17 |
| 19 | OneChart | 7.33 |
| 20 | InstructBLIP | 7.33 |
| 21 | ChartVLM | 6.67 |

(e) *Box Chart*

| No. | Model | Acc++ |
|---|---|---|
| 1 | GPT-4V | 57.50 |
| 2 | GPT-4O | 57.50 |
| 3 | InternLM-v2 | 43.50 |
| 4 | ERNIE | 40.00 |
| 5 | Qwen-VL | 35.00 |
| 6 | Mini-Gemini | 30.00 |
| 7 | LLaVA-v1.5 | 28.63 |
| 8 | mPLUG-Owl | 26.50 |
| 9 | MiniGPT-v2 | 25.88 |
| 10 | ChartLlama | 25.00 |
| 11 | DocOwl-v1.5 | 20.50 |
| 12 | SPHINX | 19.00 |
| 13 | BLIP2 | 17.63 |
| 14 | CogVLM | 16.00 |
| 15 | InternLM | 15.00 |
| 16 | Shikra | 11.88 |
| 17 | CogAgent | 9.38 |
| 18 | ChartVLM | 5.25 |
| 19 | OneChart | 2.75 |
| 20 | InstructBLIP | 2.00 |
| 21 | VisualGLM | 0.25 |

(f) *Radar Chart*

| No. | Model | Acc++ |
|---|---|---|
| 1 | GPT-4O | 83.33 |
| 2 | GPT-4V | 70.00 |
| 3 | InternLM-v2 | 52.00 |
| 4 | ERNIE | 51.67 |
| 5 | Mini-Gemini | 37.17 |
| 6 | DocOwl-v1.5 | 35.33 |
| 7 | ChartLlama | 28.50 |
| 8 | MiniGPT-v2 | 28.17 |
| 9 | LLaVA-v1.5 | 26.00 |
| 10 | mPLUG-Owl | 24.17 |
| 11 | BLIP2 | 22.00 |
| 12 | Qwen-VL | 19.50 |
| 13 | SPHINX | 17.17 |
| 14 | CogVLM | 14.33 |
| 15 | InstructBLIP | 12.50 |
| 16 | InternLM | 12.00 |
| 17 | CogAgent | 11.67 |
| 18 | OneChart | 6.33 |
| 19 | ChartVLM | 5.50 |
| 20 | Shikra | 4.17 |
| 21 | VisualGLM | 3.33 |

(g) *Scatter Chart*

| No. | Model | Acc++ |
|---|---|---|
| 1 | GPT-4V | 100.0 |
| 2 | GPT-4O | 100.0 |
| 3 | ERNIE | 70.00 |
| 4 | OneChart | 53.50 |
| 5 | InternLM-v2 | 52.50 |
| 6 | Mini-Gemini | 43.00 |
| 7 | LLaVA-v1.5 | 33.50 |
| 8 | BLIP2 | 33.00 |
| 9 | SPHINX | 31.00 |
| 10 | mPLUG-Owl | 28.50 |
| 11 | CogAgent | 27.50 |
| 12 | DocOwl1.5 | 26.00 |
| 13 | ChartLlama | 25.50 |
| 14 | Qwen-VL | 18.50 |
| 15 | CogVLM | 16.00 |
| 16 | VisualGLM | 15.50 |
| 17 | MiniGPT-v2 | 15.50 |
| 18 | InstructBLIP | 9.00 |
| 19 | Shikra | 8.50 |
| 20 | InternLM | 8.50 |
| 21 | ChartVLM | 0.00 |

(h) *Node Chart*

| No. | Model | Acc++ |
|---|---|---|
| 1 | GPT-4O | 65.00 |
| 2 | ERNIE | 56.25 |
| 3 | GPT-4V | 56.25 |
| 4 | InternLM-v2 | 46.12 |
| 5 | DocOwl-v1.5 | 40.25 |
| 6 | BLIP2 | 28.00 |
| 7 | mPLUG-Owl | 27.50 |
| 8 | LLaVA-v1.5 | 27.38 |
| 9 | MiniGPT-v2 | 27.13 |
| 10 | Mini-Gemini | 27.00 |
| 11 | ChartLlama | 26.38 |
| 12 | SPHINX | 25.88 |
| 13 | Qwen-VL | 25.50 |
| 14 | CogAgent | 15.50 |
| 15 | OneChart | 7.75 |
| 16 | ChartVLM | 6.50 |
| 17 | CogVLM | 6.13 |
| 18 | VisualGLM | 5.13 |
| 19 | InternLM | 5.13 |
| 20 | Shikra | 3.63 |
| 21 | InstructBLIP | 2.38 |

(i) *Combination Chart*

| No. | Model | Acc++ |
|---|---|---|
| 1 | GPT-4O | 65.00 |
| 2 | InternLM-v2 | 57.89 |
| 3 | GPT-4V | 53.26 |
| 4 | ERNIE | 47.39 |
| 5 | Mini-Gemini | 39.57 |
| 6 | DocOwl-v1.5 | 35.27 |
| 7 | Qwen-VL | 29.46 |
| 8 | mPLUG-Owl | 27.80 |
| 9 | LLaVA-v1.5 | 25.61 |
| 10 | MiniGPT-v2 | 22.37 |
| 11 | ChartLlama | 22.02 |
| 12 | BLIP2 | 21.65 |
| 13 | CogAgent | 20.39 |
| 14 | InternLM | 19.70 |
| 15 | InstructBLIP | 17.96 |
| 16 | SPHINX | 17.87 |
| 17 | CogVLM | 14.41 |
| 18 | OneChart | 12.34 |
| 19 | Shikra | 8.59 |
| 20 | ChartVLM | 8.02 |
| 21 | VisualGLM | 3.46 |

(j) *Regular Type*

| No. | Model | Acc++ |
|---|---|---|
| 1 | GPT-4O | 63.33 |
| 2 | GPT-4V | 55.83 |
| 3 | ERNIE | 46.39 |
| 4 | InternLM-v2 | 41.75 |
| 5 | Mini-Gemini | 31.81 |
| 6 | LLaVA-v1.5 | 27.39 |
| 7 | DocOwl-v1.5 | 26.86 |
| 8 | Qwen-VL | 26.56 |
| 9 | mPLUG-Owl | 25.47 |
| 10 | MiniGPT-v2 | 25.06 |
| 11 | ChartLlama | 22.56 |
| 12 | BLIP2 | 18.44 |
| 13 | SPHINX | 17.92 |
| 14 | CogAgent | 14.36 |
| 15 | CogVLM | 11.89 |
| 16 | InternLM | 10.11 |
| 17 | OneChart | 8.75 |
| 18 | Shikra | 7.50 |
| 19 | ChartVLM | 5.92 |
| 20 | InstructBLIP | 5.50 |
| 21 | VisualGLM | 4.22 |

(k) *Extra Type*

| No. | Model | Acc++ |
|---|---|---|
| 1 | GPT-4O | 64.27 |
| 2 | GPT-4V | 54.39 |
| 3 | InternLM-v2 | 51.34 |
| 4 | ERNIE | 46.95 |
| 5 | Mini-Gemini | 36.54 |
| 6 | DocOwl-v1.5 | 31.62 |
| 7 | Qwen-VL | 28.18 |
| 8 | mPLUG-Owl | 26.78 |
| 9 | LLaVA-v1.5 | 26.39 |
| 10 | MiniGPT-v2 | 23.55 |
| 11 | ChartLlama | 22.26 |
| 12 | BLIP2 | 20.24 |
| 13 | CogAgent | 18.07 |
| 14 | SPHINX | 17.89 |
| 15 | InternLM | 15.49 |
| 16 | CogVLM | 13.30 |
| 17 | InstructBLIP | 12.49 |
| 18 | OneChart | 12.04 |
| 19 | Shikra | 8.11 |
| 20 | ChartVLM | 6.90 |
| 21 | VisualGLM | 3.79 |

(l) *Average*

Table 23: Leaderboards of tasks, dataset splits and average **Acc++** (%) performance on ChartBench. We report the results of the best-performing prompt for each MLLM.

| No. | Model | Acc++ | No. | Model | Acc++ | No. | Model | Acc++ | No. | Model | Acc++ | No. | Model | Acc++ |
|---|---|---|---|---|---|---|---|---|---|---|---|---|---|---|
| 1 | GPT-4O | 97.62 | 1 | ERNIE | 44.76 | 1 | GPT-4O | 66.19 | 1 | GPT-4O | 53.33 | 1 | GPT-4O | 40.48 |
| 2 | GPT-4V | 96.19 | 2 | GPT-4O | 43.33 | 2 | InternLM-v2 | 54.63 | 2 | ERNIE | 47.14 | 2 | InternLM-v2 | 36.71 |
| 3 | Mini-Gemini | 80.52 | 3 | InternLM-v2 | 36.63 | 3 | GPT-4V | 48.57 | 3 | GPT-4V | 46.19 | 3 | GPT-4V | 36.19 |
| 4 | InternLM-v2 | 68.29 | 4 | DocOwl1.5 | 34.48 | 4 | ERNIE | 32.86 | 4 | InternLM-v2 | 45.80 | 4 | DocOwl1.5 | 33.76 |
| 5 | ERNIE | 65.24 | 5 | GPT-4V | 30.95 | 5 | DocOwl1.5 | 31.10 | 5 | DocOwl1.5 | 30.48 | 5 | SPHINX | 32.19 |
| 6 | ChartLlama | 62.57 | 6 | LLaVA-v1.5 | 23.14 | 6 | Qwen-VL | 27.29 | 6 | LLaVA-v1.5 | 26.48 | 6 | ERNIE | 29.24 |
| 7 | CogAgent | 60.05 | 7 | BLIP2 | 22.00 | 7 | mPLUG-Owl | 26.05 | 7 | Mini-Gemini | 22.00 | 7 | ChartLlama | 26.24 |
| 8 | Qwen-VL | 51.67 | 8 | Mini-Gemini | 17.62 | 8 | Mini-Gemini | 26.00 | 8 | Qwen-VL | 21.71 | 8 | Mini-Gemini | 25.67 |
| 9 | MiniGPT-v2 | 49.86 | 9 | mPLUG-Owl | 15.81 | 9 | LLaVA-v1.5 | 25.33 | 9 | BLIP2 | 18.10 | 9 | Qwen-VL | 22.43 |
| 10 | OneChart | 49.57 | 10 | Shikra | 15.48 | 10 | BLIP2 | 24.29 | 10 | mPLUG-Owl | 16.52 | 10 | MiniGPT-v2 | 17.52 |
| 11 | mPLUG-Owl | 47.86 | 11 | ChartVLM | 11.90 | 11 | MiniGPT-v2 | 20.43 | 11 | Shikra | 11.38 | 11 | CogVLM | 13.29 |
| 12 | InstructBLIP | 42.29 | 12 | Qwen-VL | 11.14 | 12 | Shikra | 17.57 | 12 | MiniGPT-v2 | 10.67 | 12 | mPLUG-Owl | 11.33 |
| 13 | Internlm | 38.48 | 13 | Internlm | 10.38 | 13 | Internlm | 14.33 | 13 | InstructBLIP | 9.67 | 13 | Internlm | 9.14 |
| 14 | LLaVA-v1.5 | 32.33 | 14 | SPHINX | 9.05 | 14 | CogVLM | 14.19 | 14 | Internlm | 9.62 | 14 | ChartVLM | 5.38 |
| 15 | DocOwl1.5 | 30.43 | 15 | MiniGPT-v2 | 8.38 | 15 | CogAgent | 14.05 | 15 | SPHINX | 8.52 | 15 | LLaVA-v1.5 | 4.10 |
| 16 | CogVLM | 29.14 | 16 | InstructBLIP | 6.86 | 16 | ChartVLM | 10.62 | 16 | ChartVLM | 7.86 | 16 | BLIP2 | 3.71 |
| 17 | BLIP2 | 29.05 | 17 | CogAgent | 4.24 | 17 | SPHINX | 10.05 | 17 | CogVLM | 7.33 | 17 | InstructBLIP | 3.29 |
| 18 | VisualGLM | 16.29 | 18 | CogVLM | 2.81 | 18 | ChartLlama | 7.33 | 18 | CogAgent | 3.86 | 18 | VisualGLM | 3.19 |
| 19 | Shikra | 3.71 | 19 | ChartLlama | 1.19 | 19 | InstructBLIP | 2.48 | 19 | ChartLlama | 1.19 | 19 | OneChart | 2.90 |
| 20 | ChartVLM | 2.10 | 20 | VisualGLM | 0.00 | 20 | OneChart | 0.05 | 20 | VisualGLM | 0.00 | 20 | Shikra | 2.76 |
| 21 | SPHINX | 0.00 | 21 | OneChart | 0.00 | 21 | VisualGLM | 0.00 | 21 | OneChart | 0.00 | 21 | CogAgent | 2.71 |
| | (a) *CR.* | | | (b) *VE.* | | | (c) *VC.* | | | (d) *GC.* | | | (e) *Number QA.* | |

Table 24: Leaderboards of different chart tasks on ChartBench. We report zero-shot **Acc++** (%) performance of the best-performing prompt for each MLLM.

| No. | Model | CoR | No. | Model | CoR | No. | Model | CoR | No. | Model | CoR |
|---|---|---|---|---|---|---|---|---|---|---|---|
| 1 | GPT-4O | 1.43 | 1 | ERNIE | 44.76 | 1 | GPT-4O | 16.19 | 1 | GPT-4O | 41.43 |
| 2 | GPT-4V | 2.86 | 2 | GPT-4O | 44.76 | 2 | InternLM-v2 | 27.71 | 2 | ERNIE | 47.62 |
| 3 | Mini-Gemini | 17.86 | 3 | BLIP2 | 55.14 | 3 | GPT-4V | 34.76 | 3 | GPT-4V | 47.62 |
| 4 | ERNIE | 19.52 | 4 | InternLM-v2 | 57.71 | 4 | ERNIE | 41.43 | 4 | InternLM-v2 | 51.46 |
| 5 | InternLM-v2 | 30.24 | 5 | DocOwl1.5 | 58.24 | 5 | BLIP2 | 53.33 | 5 | BLIP2 | 61.76 |
| 6 | mPLUG-Owl | 36.24 | 6 | GPT-4V | 63.33 | 6 | DocOwl1.5 | 55.19 | 6 | DocOwl1.5 | 63.19 |
| 7 | OneChart | 36.67 | 7 | mPLUG-Owl | 66.24 | 7 | mPLUG-Owl | 56.48 | 7 | mPLUG-Owl | 66.57 |
| 8 | CogAgent | 37.05 | 8 | Mini-Gemini | 70.43 | 8 | Mini-Gemini | 59.38 | 8 | LLaVA-v1.5 | 71.00 |
| 9 | ChartLlama | 37.10 | 9 | LLaVA-v1.5 | 76.67 | 9 | Qwen-VL | 63.14 | 9 | Mini-Gemini | 71.10 |
| 10 | Qwen-VL | 42.71 | 10 | Internlm | 80.67 | 10 | LLaVA-v1.5 | 69.29 | 10 | Qwen-VL | 74.86 |
| 11 | MiniGPT-v2 | 44.19 | 11 | ChartVLM | 80.71 | 11 | MiniGPT-v2 | 69.48 | 11 | InstructBLIP | 78.48 |
| 12 | BLIP2 | 49.24 | 12 | Shikra | 82.14 | 12 | Shikra | 73.71 | 12 | Internlm | 80.90 |
| 13 | LLaVA-v1.5 | 51.24 | 13 | MiniGPT-v2 | 84.14 | 13 | Internlm | 77.38 | 13 | ChartVLM | 82.71 |
| 14 | Internlm | 51.38 | 14 | Qwen-VL | 84.57 | 14 | CogAgent | 78.86 | 14 | MiniGPT-v2 | 83.81 |
| 15 | InstructBLIP | 56.95 | 15 | InstructBLIP | 85.14 | 15 | CogVLM | 80.71 | 15 | Shikra | 85.67 |
| 16 | DocOwl1.5 | 65.05 | 16 | SPHINX | 85.48 | 16 | SPHINX | 83.81 | 16 | SPHINX | 86.19 |
| 17 | CogVLM | 69.33 | 17 | CogAgent | 89.29 | 17 | ChartVLM | 87.71 | 17 | CogAgent | 90.00 |
| 18 | VisualGLM | 79.19 | 18 | CogVLM | 94.29 | 18 | ChartLlama | 88.24 | 18 | CogVLM | 90.14 |
| 19 | ChartVLM | 93.57 | 19 | ChartLlama | 94.90 | 19 | InstructBLIP | 96.57 | 19 | ChartLlama | 94.76 |
| 20 | Shikra | 94.33 | 20 | VisualGLM | 99.67 | 20 | VisualGLM | 99.81 | 20 | VisualGLM | 99.71 |
| 21 | SPHINX | 100.0 | 21 | OneChart | 100.0 | 21 | OneChart | 99.81 | 21 | OneChart | 100.0 |
| | (a) *Chart Recognition.* | | | (b) *Value Extraction.* | | | (c) *Value Comparison.* | | | (d) *Global Conception.* | |

Table 25: Leaderboards of different chart tasks on ChartBench. We report zero-shot **CoR** (%) performance of the best-performing prompt for each MLLM.

| No. | Model | Acc++ | No. | Model | Acc++ | No. | Model | CoR | No. | Model | CoR |
|---|---|---|---|---|---|---|---|---|---|---|---|
| 1 | GPT-4O | 83.30 | 1 | GPT-4O | 61.00 | 1 | GPT-4O | 10.62 | 1 | GPT-4O | 23.75 |
| 2 | GPT-4V | 77.40 | 2 | InternLM-v2 | 54.80 | 2 | GPT-4V | 18.75 | 2 | InternLM-v2 | 33.69 |
| 3 | InternLM-v2 | 73.16 | 3 | DocOwl-v1.5 | 43.50 | 3 | InternLM-v2 | 20.88 | 3 | ERNIE | 35.62 |
| 4 | DocOwl-v1.5 | 50.19 | 4 | GPT-4V | 43.00 | 4 | ERNIE | 35.00 | 4 | GPT-4V | 41.25 |
| 5 | ERNIE | 49.44 | 5 | ERNIE | 42.95 | 5 | DocOwl-v1.5 | 44.50 | 5 | DocOwl-v1.5 | 50.12 |
| 6 | Qwen-VL | 45.71 | 6 | Mini-Gemini | 32.25 | 6 | Qwen-VL | 51.00 | 6 | Mini-Gemini | 52.56 |
| 7 | Mini-Gemini | 44.46 | 7 | Qwen-VL | 28.70 | 7 | Mini-Gemini | 51.94 | 7 | MiniGPT-v2 | 54.31 |
| 8 | ChartLlama | 33.59 | 8 | mPLUG-Owl | 26.45 | 8 | MiniGPT-v2 | 53.37 | 8 | LLaVA-v1.5 | 58.06 |
| 9 | LLaVA-v1.5 | 29.76 | 9 | LLaVA-v1.5 | 22.55 | 9 | LLaVA-v1.5 | 54.81 | 9 | Qwen-VL | 62.31 |
| 10 | CogAgent | 29.52 | 10 | ChartLlama | 22.10 | 10 | ChartLlama | 63.31 | 10 | mPLUG-Owl | 63.19 |
| 11 | mPLUG-Owl | 24.83 | 11 | BLIP2 | 20.95 | 11 | mPLUG-Owl | 65.44 | 11 | BLIP2 | 69.56 |
| 12 | BLIP2 | 24.11 | 12 | MiniGPT-v2 | 20.45 | 12 | BLIP2 | 66.00 | 12 | ChartLlama | 71.00 |
| 13 | SPHINX | 22.40 | 13 | CogAgent | 17.95 | 13 | SPHINX | 67.31 | 13 | SPHINX | 71.12 |
| 14 | CogVLM | 21.78 | 14 | SPHINX | 16.85 | 14 | CogAgent | 71.06 | 14 | InternLM | 76.38 |
| 15 | MiniGPT-v2 | 21.46 | 15 | ChartVLM | 15.55 | 15 | OneChart | 73.94 | 15 | CogAgent | 80.06 |
| 16 | OneChart | 18.39 | 16 | InternLM | 14.70 | 16 | CogVLM | 78.00 | 16 | CogVLM | 82.25 |
| 17 | ChartVLM | 18.20 | 17 | CogVLM | 12.60 | 17 | InstructBLIP | 81.06 | 17 | InstructBLIP | 82.81 |
| 18 | InstructBLIP | 14.03 | 18 | InstructBLIP | 11.15 | 18 | InternLM | 82.62 | 18 | OneChart | 86.44 |
| 19 | InternLM | 12.02 | 19 | OneChart | 9.10 | 19 | ChartVLM | 88.50 | 19 | ChartVLM | 87.31 |
| 20 | VisualGLM | 6.79 | 20 | Shikra | 5.55 | 20 | VisualGLM | 93.31 | 20 | Shikra | 91.75 |
| 21 | Shikra | 6.06 | 21 | VisualGLM | 3.40 | 21 | Shikra | 95.25 | 21 | VisualGLM | 95.44 |
| | (a) *With Annotations.* | | | (b) *Without Annotations.* | | | (c) *With Annotations.* | | | (d) *Without Annotations.* | |

Table 26: Leaderboards w.r.t. data annotations of **Acc++** (%) and **CoR** (%) performance on ChartBench.

## H CHART TYPE THUMBNAILS

Previous benchmarks Masry et al. (2022); Methani et al. (2020); Kantharaj et al. (2022a;b); Chen et al. (2024a) mainly focus on the line, bar, and pie charts. To enlarge chart diversity, ChartBench provides 9 major categories and 42 subcategories of charts, including regular and specialized ones. We provide thumbnails of all chart types for visualizations in Fig. 13 & 14.

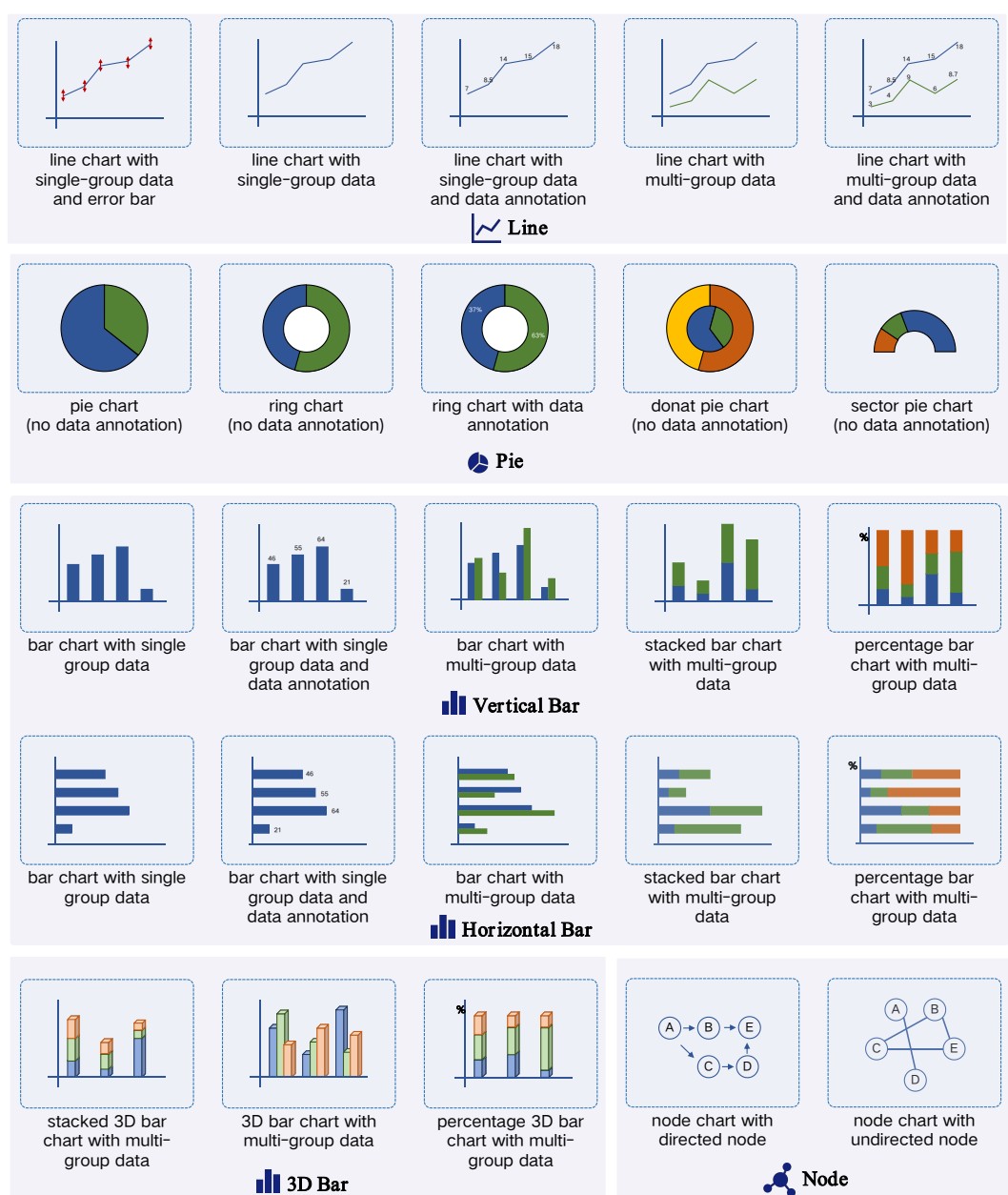

Figure 13: The categories and thumbnail examples of ChartBench (Part 1). We strive to avoid direct labeling of chart data to encourage MLLMs to understand charts using human-like visual reasoning and ensure the credibility of the data. The example charts are provided as thumbnail representations of the corresponding chart features.

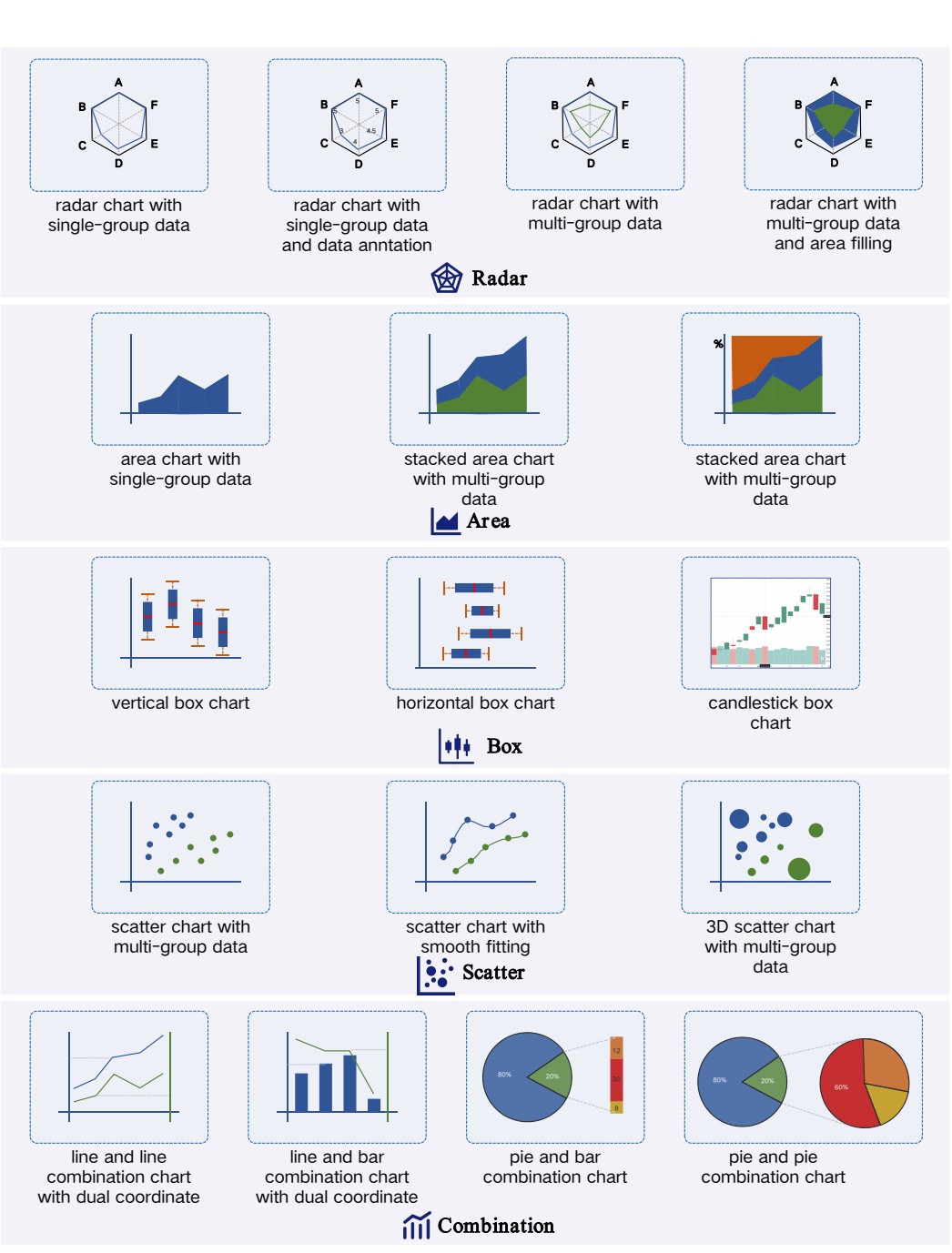

Figure 14: The categories and thumbnail examples of ChartBench (Part 2). We strive to avoid direct labeling of chart data to encourage MLLMs to understand charts using human-like visual reasoning and ensure the credibility of the data. The example charts are provided as thumbnail representations of the corresponding chart features.

