# OpenReview forum: "ChartBench: A Benchmark for Complex Visual Reasoning in Charts"
_ICLR.cc/2025/Conference — Submitted to ICLR 2025_

### Official Review · Reviewer_uiYC · 2024-10-28

**Soundness:** 3
**Presentation:** 3
**Contribution:** 2
**Rating:** 6
**Confidence:** 4

**Summary:**

The paper presents ChartBench, a chart-related benchmark that is created to evaluate the chart comprehension abilities of the existing VLM models. It comprises 42 categories, 66.6K charts, and 600K question-answer pairs, with a focus on complex visual reasoning, especially for charts without data point annotations. Besides, the authors introduce an evaluation metric, Acc++, to accurately evaluate the performance of VLM models. Experiments are conducted on the proposed ChartBench and ChartQA datasets to show the effectiveness of the proposed dataset and the chain of thought and supervised fine-tuning baselines.

**Strengths:**

- ChartBench contains a vast amount of visual chart data and chart QA data.

- This manuscript is well-organized and well-presented. Besides, the appendix of this paper is also well-written, showcasing numerous visual examples, quality inspection, additional experimental results and analyses, making the content comprehensive.

- The authors employ 200 question templates when constructing QA pairs to ensure diversity, which is an advantage.

**Weaknesses:**

- 1) This work is clearly written and organized, but I feel that its biggest drawback is its technical contribution: a) The Acc++ proposed in this paper mainly derives from MME's Acc+, and the main difference between Acc+ and Acc++ lies in the random replacement of the ground truth. I feel that the improvement seems insufficient to serve as a standalone technical contribution for ICLR submission; b) The chain of thought and supervised fine-tuning baselines proposed by the authors are also common techniques in LLM and VLM community, and please provide the detailed comparisons and design considerations between state-of-the-art techniques and the proposed chain of thought or supervised fine-tuning baselines during the rebuttal period.

- 2) The authors emphasize that the chart dataset they constructed does not have data point annotations. However, I believe that data point annotations for chart-related tasks are important, due to the following reasons: a) Data point annotations can serve as auxiliary information for QA tasks, thereby improving the interpretability of QA task predictions; b) Data point annotations can facilitate research on chart information extraction tasks, such as the recent GOT [Ref A]. Please explain the reason for excluding data point annotations.

[Ref A] General OCR Theory: Towards OCR-2.0 via a Unified End-to-end Model

- 3) Considering that the authors reported the results on ChartQA dataset, it is necessary to validate the effectiveness of the proposed ChartBench **training set** on the ChartQA dataset. For example, you could conduct an additional experiment by comparing a selected VLM (such as InternLM-XComposer-V2) and the VLM fine-tuned your proposed training set (such as InternLM-XComposer-V2 fine-tuned results), and the models should be tested on ChartQA testset as reported in Table 3.

- 4) It is recommended to include some of the latest work on VLM models for comparison in Tables 3 and 4, such as Qwen2-VL, or DeepSeek-VL, etc.

- 5) It is suggested to discuss the usefulness of the ChartBench dataset in real-world scenarios, such as in time series prediction tasks.

**Questions:**

- 1) By observing Figure 3, there is a significant difference in the distribution of charts, CSVs, and queries between ChartBench and ChartQA. Is this difference due to the fact that the ChartBench dataset is primarily simulation-based? Please explain this reason during the rebuttal.

---

> ### Author Response · Authors · 2024-11-20
>
> Thank you for your review and detailed comments! Please let us know if you have any further concerns or suggestions.
>
> ---
>
> **Q1: About the Contribution of ChartBench**
>
> In addition to expanding chart types and providing a convenient ACC++ evaluation metric, ChartBench's more significant contribution lies in its **focus on charts without data point annotations**. In many real-world scenarios, charts provide rich data but often lack text annotations for all data points. This leads to MLLMs heavily relying on textual information within the charts rather than developing true visual understanding ability. The introduction of ChartBench aims to encourage the community to pay attention to the visual logical reasoning capabilities of MLLMs in handling chart data, ensuring that they can provide truly reliable numerical data for more rigorous chart analysis.
>
> ---
>
> **Q2: Why No Data Annotation?**
>
> > However, I believe that data point annotations for chart-related tasks are important. ... Please explain the reason for excluding data point annotations.
>
> Yes, data point annotations are crucial for the accurate understanding of charts. However, in real-world scenarios, many charts lack data point annotations and require MLLMs to reason using visual cues in the chart, such as stock K-line charts or information-dense line charts. In these cases, only a few positions have data point annotations, and for QAs about other positions, MLLMs always provide unreliable numerical values, leading to untrustworthy analyses based on it. This is the motivation behind ChartBench: to evaluate whether MLLMs can truly infer reliable numerical values using visual cues rather than relying solely on their OCR capabilities.
>
> ---
>
> **Q3: Comparison on ChartQA**
>
> > Considering that the authors reported the results on ChartQA dataset, it is necessary to validate the effectiveness of the proposed ChartBench training set on the ChartQA dataset.
>
> Thanks for your reasonable suggestion. In fact, we have provided the results for InternLM-XC-v2 and its version after SFT on the ChartBench training set. These results are currently in Appendix E.1, specifically in Table 16 (on MME, SEED, QBench, etc.) and Table 17 (ChartQA). Given the importance of these results for readers' understanding of ChartBench, we will move them to the main body of the paper.
>
> ---
>
> **Q4: More Experiment on Recent Models**
>
> Yes, we have added the results for recent top-tier VLMs (i.e., DeepSeek, InternLM-v2, Qwen2-VL) as shown in the table below:
>
> | ChartBench        | Acc+  | NQA   | Regular | Acc+  | NQA   | Extra | All  |
> |-------------------|-------|-------|---------|-------|-------|-------|------|
> | DeepSeek-VL-7B    | 15.68 | 20.00 | 16.54   | 18.51 | 29.73 | 20.74 | 18.42 |
> | InternLM-v2-8B    | 40.91 | 50.00 | 42.72   | 36.12 | 47.59 | 38.40 | 40.73 |
> | Qwen2-VL-7B       | 60.45 | 50.00 | 58.37   | 68.99 | 53.30 | 65.87 | 61.70 |
>
> We also provide the performance of these models on ChartQA to validate the claims of ChartBench:
>
> | ChartQA        | Human | Aug.  | Avg.  |
> |----------------|-------|-------|-------|
> | DeepSeek-VL-7B | 44.88 | 76.56 | 60.72 |
> | InternLM-v2-8B | -     | -     | 83.30 |
> | Qwen2-VL-7B    | -     | -     | 83.00 |
>
> We will include this content in the paper and continue to keep it updated. Thank you for your suggestions, which have made our paper more comprehensive.

---

> ### Author Response · Authors · 2024-11-20
>
> **Q5: Application on Real-world**
>
> > It is suggested to discuss the usefulness of the ChartBench dataset in real-world scenarios, such as in time series prediction tasks.
>
> This is an good suggestion! ChartBench contains many real-time series charts, such as stock K-line charts in the `box` type, which can support prediction tasks. However, existing models have low accuracy and reliability in numerical extraction, leading to low credibility in predictions based on these models. We will incorporate this suggestion as one of the evaluation tasks in ChartBench.
>
> ---
>
> **Q6: About the Distribution of ChartBench**
>
> > By observing Figure 3, there is a significant difference in the distribution of charts, CSVs, and queries between ChartBench and ChartQA. Is this difference due to the fact that the ChartBench dataset is primarily simulation-based? Please explain this reason during the rebuttal.
>
> Not exactly. Many of the datasets we compare against also contain a large number of simulation-based instances, such as PlotQA, ChartX, and UniChart. The primary reason for the differences in charts is the variety of chart types. We have expanded the types of charts as much as possible, and during clustering, we observe that charts of the same type tend to cluster together.
>
> The main reason for the differences in CSVs is the different data sources. For example, ChartQA data comes from the web, whereas our data primarily comes from Kaggle and GPT. Additionally, our tables contain an average of `87.62` numbers, compared to an average of `22.11` numbers in ChartQA. The queries are similar because our main goal is to compare with ChartQA. By using similar questions to evaluate MLLMs on both datasets, we can assess whether MLLMs possess visual reasoning capabilities despite the differences in charts and CSVs.
>
> ---
>
> We hope to have addressed your concerns!

---

> ### Author Response · Authors · 2024-11-24
> **Awaiting Your Confirmation**
>
> Dear Reviewer uiYC,
>
> With the discussion phase ending in **3 days**, we would greatly appreciate any further guidance or confirmation from your side. We wanted to follow up on our previous response to your valuable suggestions. Your insights are very important to us, and we are eager to ensure that we have addressed all concerns appropriately.
>
> Warm Regards, :-)

---

> > ### Comment · Reviewer_uiYC · 2024-12-01
> > **Thanks for your response**
> >
> > Thanks for the authors' response. I have carefully reviewed your feedback, which has addressed my concerns in experimental verification and evaluation details during the rebuttal period. Thanks

---

> > > ### Author Response · Authors · 2024-12-01
> > >
> > > Thank you! Wishing you a wonderful day!

---

### Official Review · Reviewer_kV3u · 2024-11-01

**Soundness:** 3
**Presentation:** 2
**Contribution:** 2
**Rating:** 6
**Confidence:** 3

**Summary:**

This paper addresses an important gap in multimodal evaluation by introducing ChartBench, a large-scale chart comprehension benchmark.  The benchmark encompasses more diverse chart types than existing data, covering both annotated and unannotated charts. The authors also refine the Acc+ metric, enabling efficient and stricter evaluation. The authors conduct comprehensive evaluations using various models and provide detailed analysis of the results.

**Strengths:**

- The authors focus on an important aspect MLLM evaluation that existing works lack, and address the problem by proposing a new dataset covering more diverse chart types
- The authors conduct thorough evaluations using various MLLMs on different chart version - the effort is noteworthy. The insightful analysis on model performances also provides valuable insights of capabilities and limitations of current models.
- The authors also enhance the existing Acc+ metrics by proposing Acc++, making the metric more robust.

**Weaknesses:**

- I'm not fully convinced that the designed question types and data collection methods are substantially different compared to existing works. It would be nice to see some arguments on why evaluating on this benchmark substantially address the limitations of existing datasets.

**Questions:**

- How evaluating on this benchmark directly address the limitations on existing datasets? For example, any side-by-side comparison examples?
- Is there any effort in ensuring that the collected questions/instructions is diverse?
- Since the benchmark is relatively large, any comments on the compute/time resources required to conduct the evaluation? Are there any potential subsets that could be representative in any way of the full dataset?

---

> ### Author Response · Authors · 2024-11-20
>
> Thank you for your review and detailed comments! We are glad to see your recognition of our contribution. Please let us know if you have any further concerns or suggestions.
>
> ---
>
> **Q1: The Contribution of ChartBench**
>
> > I'm not fully convinced that the designed question types and data collection methods are substantially different compared to existing works. It would be nice to see some arguments on why evaluating on this benchmark substantially address the limitations of existing datasets.
>
> ChartBench's significant contribution lies in its **focus on charts without data point annotations**. In many real-world scenarios, charts provide rich data but often lack text annotations for all data points. Existing benchmarks often overlook the issue of data point annotations and may degrade chart understanding tasks to trivial OCR problems. This leads to MLLMs heavily relying on textual information within the charts rather than developing true visual understanding. ChartBench is the first benchmark to propose evaluating whether MLLMs can provide reliable numerical data and analysis using only visual cues in the absence of detailed data point annotations. The introduction of ChartBench aims to encourage the community to pay attention to the visual logical reasoning capabilities of MLLMs in handling chart data, ensuring that they can provide truly reliable numerical data for more rigorous chart analysis.
>
> ---
>
> **Q2: Difference with Other Datasets Examples**
>
> > How evaluating on this benchmark directly address the limitations on existing datasets? For example, any side-by-side comparison examples?
>
> Thank you for your suggestion! We provide a comparison of samples from ChartBench and ChartQA in Appendix A.1 and visualize the responses of different MLLMs on ChartBench in Appendix E.5. We will supplement the visualization of the same MLLM's performance on both ChartBench and ChartQA in subsequent updates. In fact, the results of ChartBench in Table 6, which show the performance with and without annotations, highlight the main claim of ChartBench: to evaluate whether MLLMs can provide correct answers using only visual logic without relying on annotations.
>
> ---
>
> **Q3: Diversity of Questions/Instructions**
>
> > Is there any effort in ensuring that the collected questions/instructions is diverse?
>
> To facilitate comparison with the model rankings on ChartQA, we strive to maintain consistency in the way questions are asked. Figure 2 shows the t-SNE distribution of charts, tables, and questions. As shown, our tables and charts differ significantly from those in ChartQA, but the questions are similar, meaning the instructions are consistent with those in ChartQA (especially on the QA task). This allows the differential performance of models on these two benchmarks to highlight that many models overly rely on OCR to solve chart problems rather than possessing true visual chart understanding capabilities.
>
> ---
>
> **Q4: Evaluation requirement**
>
> > Since the benchmark is relatively large, any comments on the compute/time resources required to conduct the evaluation? Are there any potential subsets that could be representative in any way of the full dataset?
>
> Thank you for your suggestion! We provide the inference speed and memory usage for each MLLM during inference in Table 12. For the entire benchmark, we have 2100 charts and 18,900 QAs (Table 10) in the test set, so some MLLMs indeed take a longer time to complete the evaluation. To address this, We have introduced a 20% toy QA set (2100 chart, 3780 QA) via random sampling to facilitate quick testing of MLLMs on ChartBench. We ensure that random sampling is performed evenly on different task types and chart types to construct a representative test subset.

---

> > ### Author Response · Authors · 2024-11-24
> > **Awaiting Your Confirmation**
> >
> > Dear Reviewer kV3u,
> >
> > With the discussion phase ending in **3 days**, we would greatly appreciate any further guidance or confirmation from your side. We wanted to follow up on our previous response to your valuable suggestions. Your insights are very important to us, and we are eager to ensure that we have addressed all concerns appropriately.
> >
> > Warm Regards, :-)

---

> > > ### Comment · Reviewer_kV3u · 2024-12-02
> > >
> > > Thank you for your detailed response, I have no further questions.

---

### Official Review · Reviewer_hEAv · 2024-11-01

**Soundness:** 2
**Presentation:** 3
**Contribution:** 2
**Rating:** 6
**Confidence:** 4

**Summary:**

This paper introduce a large-scale dataset, which contains over 66,000 charts across 42 different types and 600k qa pairs, without data point annotations, which requires models to derive values through visual elements like colors, legends, and coordinate systems . The benchmark introduces an enhanced evaluation metric called Acc++ to assess models' true understanding while avoiding lucky guesses. The paper also provides valuable baselines through Chain-of-Thought (CoT) and supervised fine-tuning approaches  to improve model performance on unannotated charts.

**Strengths:**

1、By introducing a large-scale dataset with unannotated charts, this work advances visual understanding research that more closely aligns with real-world applications.

2、The proposed strategies of CoT and SFT are effective in the tasks.

3、The experiment of the paper is comprehensive.

**Weaknesses:**

1、There seems to be little innovation or discovery in this pipeline, the proposed baseline methods (Chain-of-Thought and supervised fine-tuning) are well-established techniques in the field.

2、How does the data generation pipeline differ from other chart-related datasets?

3、Have you considered testing the models' generalization ability on charts from completely different domains or visual styles not present in the training data?

4、Some recent strong baseline models do not appear to have been evaluated，such as InternVL2、Qwen-VL2.

**Questions:**

Please see Weaknesses

---

> ### Author Response · Authors · 2024-11-20
>
> Thank you for your review and detailed comments! Here is our response.
>
> ---
>
> **Q1: About Paper Novelty and Chart Generation Pipeline**
>
> > How does the data generation pipeline differ from other chart-related datasets?
>
> ChartBench's significant contribution lies in its **focus on charts without data point annotations**. In many real-world scenarios, charts provide rich data but often lack text annotations for all data points. This leads to MLLMs heavily relying on textual information within the charts rather than developing true visual understanding. The introduction of ChartBench aims to encourage the community to pay attention to the visual logical reasoning capabilities of MLLMs in handling chart data, ensuring that they can provide truly reliable numerical data for more rigorous chart analysis. Besides, we also expand chart types and provide a convenient ACC++ metric for chart evaluation.
>
> ---
>
> **Q2: About Data Generalization**
>
> > Have you considered testing the models' generalization ability on charts from completely different domains or visual styles not present in the training data?
>
> Yes, in fact, we have provided the general QA results for InternLM-XC-v2 and its version after supervised fine-tuning on the ChartBench training set. These results are currently in Appendix E.1, specifically in Table 16 (GeneralQA: MME, SEED, QBench, etc.) and Table 17 (Chart-related: ChartQA+ChartX). We performed supervised fine-tuning on InternLM-XC-v2 using the ChartBench training set and evaluated it on ChartQA and ChartX. The significant performance improvements clearly demonstrate the effectiveness of ChartBench (ChartQA +3.1%, ChartX +11.17%).
>
> | Dataset     | InternLM-XC-v2 | InternLM-XC-v2+ChartBench SFT | Performance Gain         |
> |-----------------------------|-----------------|---------------------|-----------|
> | ChartQA_Human               | 62.72           | 63.48               | +0.76     |
> | ChartQA_Augmented           | 81.28           | 86.72               | +5.44     |
> | ChartQA_Average             | 72.00           | 75.10               | +3.10     |
> | ChartX_QA                   | 42.18           | 53.35               | +11.17    |
>
> Given the importance of these results for readers' understanding of ChartBench, we will move them to the main body of the paper.
>
> ---
>
> **Q3: About additional Experiment Results**
>
> Yes, we have added the results for recent top-tier VLMs (i.e., DeepSeek, InternLM-v2, Qwen2-VL) as shown in the table below:
>
> | ChartBench        | Acc+  | NQA   | Regular | Acc+  | NQA   | Extra | All  |
> |-------------------|-------|-------|---------|-------|-------|-------|------|
> | DeepSeek-VL-7B    | 15.68 | 20.00 | 16.54   | 18.51 | 29.73 | 20.74 | 18.42 |
> | InternLM-v2-8B    | 40.91 | 50.00 | 42.72   | 36.12 | 47.59 | 38.40 | 40.73 |
> | Qwen2-VL-7B       | 60.45 | 50.00 | 58.37   | 68.99 | 53.30 | 65.87 | 61.70 |
>
> We also provide the performance of these models on ChartQA to validate the claims of ChartBench:
>
> | ChartQA        | Human | Aug.  | Avg.  |
> |----------------|-------|-------|-------|
> | DeepSeek-VL-7B | 44.88 | 76.56 | 60.72 |
> | InternLM-v2-8B | -     | -     | 83.30 |
> | Qwen2-VL-7B    | -     | -     | 83.00 |
>
> We will include this content in the paper and continue to keep it updated. Thank you for your suggestions, which have made our paper more comprehensive.
>
> ---
>
> **Reference:**
>
> [1] ChartX & ChartVLM: A Versatile Benchmark and Foundation Model for Complicated Chart Reasoning, Arxiv 2024.
>
> [2] MMC: Advancing Multimodal Chart Understanding with LLM Instruction Tuning, NAACL 2024

---

> ### Author Response · Authors · 2024-11-24
> **Awaiting Your Confirmation**
>
> Dear Reviewer hEAv,
>
> With the discussion phase ending in **3 days**, we would greatly appreciate any further guidance or confirmation from your side. We wanted to follow up on our previous response to your valuable suggestions. Your insights are very important to us, and we are eager to ensure that we have addressed all concerns appropriately.
>
> Warm Regards, :-)

---

> > ### Comment · Reviewer_hEAv · 2024-12-02
> > **Thanks for your response**
> >
> > Thank you for your explanation. I've changed my score.

---

> > > ### Author Response · Authors · 2024-12-02
> > >
> > > Thank you for your recognition of ChartBench. Wishing you a wonderful day!

---

### Official Review · Reviewer_QY2B · 2024-11-02

**Soundness:** 3
**Presentation:** 4
**Contribution:** 3
**Rating:** 6
**Confidence:** 5

**Summary:**

This paper makes several contributions to the field of chart understanding in VLMs. The introduction of ChartBench represents a substantial advancement in benchmarking capabilities. Its comprehensive coverage of 42 chart types, 66k charts, and 600k instruction pairs significantly expands upon existing benchmarks such as ChartQA, MMC, ChartX, etc. What sets ChartBench apart is its emphasis on unannotated charts, which forces models to rely on visual reasoning through elements like colors, legends, and coordinate systems rather than simple OCR - a scenario more aligned with real-world applications.

The authors' refinement of the evaluation metric, introducing ACC++, bridges the gap between existing assessment methods. By generating negative queries through random value substitution from the same meta table, they've developed a more robust evaluation system that effectively reduces the likelihood of lucky guesses.

The paper also contributes through two baseline methods: chain of thought and supervised fine-tuning. These approaches provide a foundation for improving models' understanding of unannotated charts, offering valuable insights for future research directions. The extensive experimental evaluation, covering 18 open-source and three closed-source models, provides comprehensive insights into the current limitations of VLMs in chart comprehension.

**Strengths:**

1. This paper is well-structured and presented, with a logical flow that makes it accessible to readers across different expertise levels.
2. The paper's major contribution lies in its comprehensive dataset construction. The inclusion of 9 major categories and 42 subcategories of charts represents an advancement in chart understanding. This extensive coverage substantially surpasses existing benchmarks and provides a more realistic evaluation framework for VLMs.

**Weaknesses:**

Besides the strength, I find several aspects of this paper warrant discussion.

1. While the work expands the range of chart types and introduces the ACC++ evaluation metric, the technical innovation appears somewhat limited beyond these contributions. This raises questions about the proposed benchmark's overall novelty.
2. A more fundamental concern relates to the dataset generation approach. The authors utilize standard chart plotting libraries, which generate relatively straightforward visualizations. However, real-world charts and infographics often feature sophisticated design elements and custom modifications. This disparity raises essential questions about the dataset's representativeness and alignment with actual chart distributions encountered in practice.
3. The core challenge in understanding tasks, particularly chart comprehension, lies in generalization capabilities. The sustantial gap between synthetic and real data distribution poses a critical question: Can the robust generalization abilities of VLMs effectively bridge this gap? The heavy reliance on synthetic data potentially diminishes the benchmark's effectiveness in evaluating real-world performance.
4. To address these concerns, I recommend incorporating additional experiments that demonstrate the practical utility of this work. Specifically, experiments showing how models trained on ChartBench's synthetic data perform on real-world charts would be valuable. Such experiments would help validate whether the synthetic training data enhances models' ability to tackle real-world chart understanding problems.
5. Some personal opinions:
	- a. The baseline models should be more systematically categorized. Specialized models optimized for OCR and document understanding (like mPLUG-Owl and DocOwl) warrant separate evaluation from general-purpose VLMs (such as Qwen-VL).
	- b. the experimental evaluation would be more comprehensive with the inclusion of *recent* top-tier VLMs like InternVL

**Questions:**

Please check previous sections

---

> ### Author Response · Authors · 2024-11-20
>
> Thanks for your detailed comments! We provide detailed responses to your questions and concerns as follows. We hope the following discussion can address your concerns!
>
> ---
>
> **Q1: About Technical Novelty**
>
> > While the work expands the range of chart types and introduces the ACC++ evaluation metric, the technical innovation appears somewhat limited beyond these contributions. This raises questions about the proposed benchmark's overall novelty.
>
> In addition to expanding chart types and providing a convenient ACC++ evaluation metric, ChartBench's more significant contribution lies in its **focus on charts without data point annotations**. In many real-world scenarios, charts provide rich data but usually lack text annotations for all data points. This leads to MLLMs heavily relying on textual information within the charts rather than developing true visual understanding. ChartBench aims to encourage the community to pay attention to **the visual logical reasoning capabilities of MLLMs in handling chart data**, ensuring that they can provide truly reliable numerical data for more rigorous chart analysis.
>
> ---
>
> **Q2: Concern between Synthetic and Real-world Chart**
>
> > The substantial gap between synthetic and real data distribution poses a critical question: Can the robust generalization abilities of VLMs effectively bridge this gap? The heavy reliance on synthetic data potentially diminishes the benchmark's effectiveness in evaluating real-world performance.
>
> It is a reasonable concern. Recent work has shown that using synthetic data to enhance model performance can be highly effective, as demonstrated by ChartVLM[1] and MMC[2]. When diversity and data quality are ensured, models can effectively improve on real evaluation data. To address your concerns further, we performed supervised finetuning on InternLM-XC-v2 using the ChartBench training set and evaluated it on ChartQA and ChartX. The significant performance improvements clearly demonstrate the effectiveness of ChartBench (ChartQA +3.1%, ChartX +11.17%).
>
> | Dataset                     | InternLM-XC-v2 | InternLM-XC-v2+ChartBench SFT | Performance Gain         |
> |-----------------------------|-----------------|---------------------|-----------|
> | ChartQA_Human               | 62.72           | 63.48               | +0.76     |
> | ChartQA_Augmented           | 81.28           | 86.72               | +5.44     |
> | ChartQA_Average             | 72.00           | 75.10               | +3.10     |
> | ChartX_QA                   | 42.18           | 53.35               | +11.17    |
>
> We will include this content in the main body of the paper to make it more comprehensive.
>
> ---
>
> **Q3: More Experiment Result**
>
> > The baseline models should be more systematically categorized.  Specialized models optimized for OCR and document understanding (like mPLUG-Owl and DocOwl) warrant separate evaluation from general-purpose VLMs (such as Qwen-VL).
>
> Thank you very much for your reminder. We will categorize the models being evaluated as follows:
>
> 1. OCR Optimized Models: `mPLUG`, `DocOwl`, `CogAgent`
> 2. Chart Optimized Models: `ChartVLM`, `OneChart`, `ChartLlama`
> 3. General Purpose Models: `Internlm-XComposer`, `Internlm-XComposer-v2`, `Shikra`, `MiniGPT-v2`, `BLIP2`, `InstructBLIP`, `VisualGLM`, `SPHINX`, `LLaVA`, `CogVLM`, `Mini-Gemini`, `Qwen-VL`
>
> > The experimental evaluation would be more comprehensive with the inclusion of recent top-tier VLMs like InternVL
>
> Yes, we have added the results for recent top-tier VLMs (i.e., DeepSeek, InternLM-v2, Qwen2-VL) as shown in the table below:
>
> | ChartBench        | Acc+  | NQA   | Regular | Acc+  | NQA   | Extra | All  |
> |-------------------|-------|-------|---------|-------|-------|-------|------|
> | DeepSeek-VL-7B    | 15.68 | 20.00 | 16.54   | 18.51 | 29.73 | 20.74 | 18.42 |
> | InternLM-v2-8B    | 40.91 | 50.00 | 42.72   | 36.12 | 47.59 | 38.40 | 40.73 |
> | Qwen2-VL-7B       | 60.45 | 50.00 | 58.37   | 68.99 | 53.30 | 65.87 | 61.70 |
>
> We also provide the performance of these models on ChartQA to validate the claims of ChartBench:
>
> | ChartQA        | Human | Aug.  | Avg.  |
> |----------------|-------|-------|-------|
> | DeepSeek-VL-7B | 44.88 | 76.56 | 60.72 |
> | InternLM-v2-8B | -     | -     | 83.30 |
> | Qwen2-VL-7B    | -     | -     | 83.00 |
>
> We will include this content in the paper and continue to keep it updated. Thank you for your suggestions, which have made our paper more comprehensive.
>
> ---
>
> **Reference:**
>
> [1] ChartX & ChartVLM: A Versatile Benchmark and Foundation Model for Complicated Chart Reasoning, Arxiv 2024.
>
> [2] MMC: Advancing Multimodal Chart Understanding with LLM Instruction Tuning, NAACL 2024

---

> ### Author Response · Authors · 2024-11-24
> **Awaiting Your Confirmation**
>
> Dear Reviewer QY2B,
>
> With the discussion phase ending in **3 days**, we would greatly appreciate any further guidance or confirmation from your side. We wanted to follow up on our previous response to your valuable suggestions. Your insights are very important to us, and we are eager to ensure that we have addressed all concerns appropriately.
>
> Warm Regards, :-)

---

### Official Review · Reviewer_FpUV · 2024-11-02

**Soundness:** 3
**Presentation:** 3
**Contribution:** 2
**Rating:** 5
**Confidence:** 5

**Summary:**

The ChartBench benchmark addresses limitations in existing MLLMs’ chart comprehension by introducing a dataset with 42 chart types across 66.6k charts and 600k question-answer pairs. This benchmark requires models to interpret chart elements (e.g., color, legends, coordinate systems) without relying on data point annotations. ChartBench also introduces the Acc++ metric, designed to prevent models from making correct answers based on chance, by introducing similar query structures with slight variances in values. Two baseline approaches are presented to improve model performance on these complex, unannotated charts.

**Strengths:**

1. The benchmark includes a wide array of chart types beyond conventional bar and line charts, enhancing its utility for complex visual reasoning evaluation.

2. Acc++ mitigates the limitations of earlier metrics by reducing model reliance on chance, providing a refined tool for measuring comprehension accuracy.

3. Baseline Models and Task Variety: The paper provides two baseline models based on chain-of-thought and supervised fine-tuning, which effectively highlight the limitations in current MLLMs’ performance when applied to unannotated chart reasoning tasks.

**Weaknesses:**

1. The value extraction metric based on Acc++ does not make sense. Since the metadata has the full CSV annotation of the chart data, an effective method to evaluate the ability of MLLMs should be the comprehensive recognition of all elements in the chart. A binary classification for the value extraction task cannot fully demonstrate the visual understanding capability of MLLMs.

2. The template-based generation of instruction data somehow lacks diversity. As illustrated in Figure 3(c), the distribution difference between ChartBench and other benchmarks is less significant than the other two dimensions.

3. The generation pipeline of ChartBench lacks sufficient novelty compared with other benchmarks, e.g., ChartX.

**Questions:**

1. In Table 1, ChartX also seems to be visually grounded. Could you further check it?

**Details Of Ethics Concerns:**

See Weaknesses.

---

> ### Author Response · Authors · 2024-11-20
>
> Thank you for your review and detailed comments! You can find our detailed response to your questions and concerns below. Please let us know if you have any further concerns or suggestions.
>
> ---
>
> **Q1: The Novelty of Value Extraction Metric Based on Acc++**
>
> > Since the metadata has the full CSV annotation of the chart data, an effective method to evaluate the ability of MLLMs should be the comprehensive recognition of all elements in the chart. A binary classification for the value extraction task cannot fully demonstrate the visual understanding capability of MLLMs.
>
> For charts with sufficient data point annotations (e.g. ChartQA), directly evaluating the prediction of all elements is feasible. However, the majority of charts in ChartBench lack data point annotations, meaning MLLMs cannot obtain all numbers through OCR. This results in extremely low accuracy when directly comparing predicted tables with CSV annotations, making model rankings meaningless due to the low accuracy.
>
> StructChart[1] proposed SCRM (Structuring Chart-oriented Representation Metric), which evaluates table predictions using a method similar to IoU (i.e., correctly predicted elements / total elements). However, SCRM only requires a portion of the elements to be correctly predicted within a specified error margin. This means that even with high SCRM scores, the extracted tables are not reliable for numerically sensitive applications, such as financial report analysis.
>
> Considering these factors, we propose using Acc++ to predict a single value from the chart. For charts without data point annotations, MLLMs must use visual logic to understand the chart and provide the final answer. This approach has three advantages:
>
> 1. **Faithful Evaluation**: Models with high VE (Value Extraction) scores on ChartBench always provide reliably accurate values.
> 2. **Deployment Flexibility**: Our evaluation does not require LLM as evaluators, making it cost-effective and flexible.
> 3. **Avoiding Random Guesses**: In cases where the model makes random guesses, traditional metrics like Acc tend to be around 50%, and Acc+ proposed by MME[2] tends to be around 25%, while our Acc++ tends to be around 0%.
>
> ---
>
> **Q2: The Diversity of Template-based Instruction**
>
> > The template-based generation of instruction data somehow lacks diversity. As illustrated in Figure 3(c), the distribution difference between ChartBench and other benchmarks is less significant than the other two dimensions.
>
> ChartBench differs from other datasets (e.g. ChartQA) primarily in **whether the charts have detailed data point annotations**. To ensure that the evaluation results on ChartBench are comparable to those on ChartQA, we mainly follow the instruction types used in ChartQA & ChartX and compare the performance of the same MLLMs on both benchmarks in Table 3. As shown, when models must infer results from charts lacking detailed data point annotations using visual clues, the model rankings change significantly. This indicates that some models rely on their OCR capabilities to generate candidate answers rather than possessing true visual chart understanding.
>
> Hence, ChartBench differs from previous datasets in both the charts and meta tables but **maintains consistency in instruction design**. In fact, ChartQA itself includes questions related to chart type, value extraction, and global conception, but lacks clear categorization labels, making statistical analysis difficult. In contrast, ChartBench organizes instructions by task categories during the design phase.
>
> ---
>
> **Q3: About the Novelty of Pipeline**
>
> > The generation pipeline of ChartBench lacks sufficient novelty compared with other benchmarks, e.g., ChartX.
>
> Our pipeline incorporates previous excellent works, such as ChartX[3], ChartLLaMA[4], and MMC[5]. Here, we would like to emphasize the novelty of ChartBench, which is designed to **evaluate whether MLLMs can handle charts lacking detailed data point annotations**. It is a common scenario in fields like financial report analysis. This capability has been overlooked by the community, and we hope ChartBench will draw attention to the importance of understanding such charts. This reflects whether MLLMs **truly possess visual logical reasoning capabilities, rather than relying solely on textual logic**.

---

> ### Author Response · Authors · 2024-11-20
>
> **Q4: About ChartX in Table 1**
>
> > In Table 1, ChartX also seems to be visually grounded. Could you further check it?
>
> Thank you for your kind reminder. ChartX is an excellent work that provides high-quality charts, ample metadata, and diverse instruction data. However:
>
> 1. Not all the charts in ChartX are visually grounded, and the classification does not focus on specialized analysis of charts lacking detailed data point annotations.
>
> 2. The instruction design in ChartX emphasizes overall evaluations, such as converting charts to tables, summarizing, redrawing, and other Q&A tasks. As a result, there are fewer questions focused on numerical values, making it difficult to reflect the accuracy of MLLMs in extracting reliable numerical data.
>
> Nonetheless, we will revise the description in Table 1 to highlight the features and contributions of ChartX.
>
> ---
>
> **Reference:**
>
> [1] StructChart: Perception, Structuring, Reasoning for Visual Chart Understanding, Arxiv 2023.
>
> [2] MME: A Comprehensive Evaluation Benchmark for Multimodal Large Language Models, Arxiv 2023.
>
> [3] ChartX & ChartVLM: A Versatile Benchmark and Foundation Model for Complicated Chart Reasoning, Arxiv 2024.
>
> [4] ChartLlama: A Multimodal LLM for Chart Understanding and Generation, Arxiv 2023.
>
> [5] MMC: Advancing Multimodal Chart Understanding with LLM Instruction Tuning, NAACL 2024

---

> ### Author Response · Authors · 2024-11-24
> **Awaiting Your Confirmation**
>
> Dear Reviewer FpUV,
>
> With the discussion phase ending in **3 days**, we would greatly appreciate any further guidance or confirmation from your side. We wanted to follow up on our previous response to your valuable suggestions. Your insights are very important to us, and we are eager to ensure that we have addressed all concerns appropriately.
>
> Warm Regards, :-)

---

> > ### Comment · Reviewer_FpUV · 2024-12-01
> >
> > Thank you for your detailed response. Based on your feedback, I feel that the effectiveness and comprehensiveness of Acc++ are not yet fully demonstrated, which remains my primary concern regarding the acceptance of this paper. Additionally, when a chart lacks data annotations, it is often challenging to precisely determine the data values due to axis precision. Therefore, considering value extraction as a regression task with specific error bounds—similar to metrics like mAP in detection tasks—would serve as a more reasonable and robust evaluation criterion.

---

> > > ### Author Response · Authors · 2024-12-01
> > >
> > > In MME, the Acc+ metric indeed faces the aforementioned issues on value extraction tasks. However, in ChartBench, our improved Acc++ aims to address this by focusing on the construction of negative samples.
> > >
> > > For **positive** samples, we expect a good MLLM to correctly identify the expressions, regardless of axis precision.
> > >
> > > For **negative** samples, we ensure their expressions are as consistent as possible with the positive ones, differing only in the **numerical values**. Our rules for numbers in negative samples are:
> > > 1) randomly sampled from the same chart's datapoints;
> > > 2) significantly different from the numbers in positive samples (with a difference greater than 10% of the positive sample's GT).
> > >
> > > This design prevents ambiguity between positive and negative samples, e.g.,  `The temperature of Tokyo in August is 26.1°` and `The temperature of Tokyo in August is 26.0°,` which is akin to applying a regression metric with an error bound.
> > >
> > > It also avoids situations where MLLMs with weaker instruction-following capabilities cannot produce accurate numbers. Typically, for such MLLMs, using a regression metric with an error bound requires *first extracting numbers from the model's response* using  LLMs, adding extra computational burden due to LLM API calls.
> > >
> > > We hope this explanation addresses your concern!

---

> > > > ### Comment · Reviewer_FpUV · 2024-12-01
> > > >
> > > > Thank you for the detailed explanation. However, the current definition of negative samples, based solely on a threshold for relative error, does not take into account the precision of the chart. For instance, in the example, “The temperature of Tokyo in August is 26.1° and The temperature of Tokyo in August is 26.0°”, whether these should be classified as positive or negative samples should depend on the precision with which the data can be represented in the chart, rather than just the relative error compared to the ground-truth value.
> > > >
> > > > Regarding the second advantage you mentioned, a comprehensive evaluation metric for value extraction is not necessarily aimed at accommodating all models but rather at effectively capturing and evaluating the full range of data distributions.

---

> > > > > ### Author Response · Authors · 2024-12-01
> > > > >
> > > > > Yes, there indeed can be issues with chart precision in datasets like ChartQA.
> > > > >
> > > > > However, the charts in the ChartBench test set are sourced from `matplotlib` and `online plotting websites`. In contrast, some charts in ChartQA are sourced from websites like `Pew Research`, where the charts often reflect general trends rather than precise values. Theoretically, these charts in ChartBench, which have undergone quality control (with manual removal of occlusions and other issues), **accurately** and **faithfully** reflect all the values in the meta table.
> > > > >
> > > > > As shown in the ChartBench examples provided in the appendix, some data points may be challenging for humans to discern intuitively. However, we believe that MLLMs are fully capable of accurately inferring the corresponding numbers, as all positions are plotted according to numerical proportions, which should result in a precise number rather than an ambiguous range for a good MLLM.
> > > > >
> > > > > Additionally, the regression with an error bound is a valuable metric. Our Acc++ is designed as an **alternative** metric that does not require LLM assistance. We discuss this point in the appendix, where the results show that Acc++ and the number with a 5% error margin (the metric used in ChartQA) exhibit similar ranking trends. Hence, we adopt Acc++ to avoid the heavy LLM API requirements. We will include regression results for the value extraction task on some SOTA models to provide more comprehensive evaluations. Thanks for your suggestion!
> > > > >
> > > > > Looking forward to further discussion with you!

---

> > > > > ### Author Response · Authors · 2024-12-03
> > > > >
> > > > > Dear Reviewer FpUV,
> > > > >
> > > > > To further address your concern, we provide the **NQA** (Number QA) task, which uses the same questions as the **VE** (Value Extraction) task but employs LLM to normalize model outputs. We use the regression task metric with 5% error bounds (similar to relaxed accuracy in ChartQA). Comparing the trends in **Tables 3&4** for the VE and NQA tasks, we believe the Acc++ metric is reasonable for ChartBench.
> > > > >
> > > > > Looking forward to your early reply.
> > > > >
> > > > > Warm Regards, :-)

---

### Author Response · Authors · 2024-11-20
**Incorporated feedback and new results**

Thanks to all reviewers for your detailed reviews and constructive suggestions. In our rebuttal, we have addressed the following issues:

1. **Emphasized the Motivation and Novelty of ChartBench**: ChartBench is the first benchmark to focus on the performance of MLLMs on **charts without data point annotations**. In real-world scenarios, the ability to infer accurate numerical values from charts lacking data point annotations, based on visual cues instead of OCR, is a crucial foundation for assessing the reliability of MLLM analyses.

2. **Addressed Concerns About Synthetic Data**: The test set (benchmark) charts in ChartBench are collected, cleaned, and validated from the real world. We also provide a synthetic data training set to enhance MLLM performance. To alleviate reviewers' concerns, we have included in the appendix the performance of models after SFT on the training set across multiple benchmarks (Tables 16-17). We have also added results on ChartX. The experiments show that the training set not only helps with chart-related tasks but also improves performance on general tasks. We will move this content to the main body of the paper from the appendix to facilitate readers' understanding of ChartBench.

3. **Discussed Diversity in Charts, Tables, and Instructions**: We detail our approach to controlling the diversity of charts and tables and the motivation behind our instruction design.

4. **Added More Experimental Results**: We have included evaluation results for Qwen2-VL, DeepSeek, and InternLM2. We have also categorized the models in the tables to make the paper clearer.

5. **Discussed the Importance of Unannotated Charts and Real-World Applications**: Charts without data point annotations are very common in real-world applications, such as stock K-line charts and information-rich line charts. Only a few positions have data point annotations, while other numerical values need to be inferred by MLLMs based on visual cues. Providing accurate and reliable numerical values is essential for MLLMs to offer reasonable and credible summaries and analyses.

---

### Author Response · Authors · 2024-12-01
**ICLR Reminder**

Dear Reviewers,

We have carefully addressed your concerns, and we are very eager to get your feedback. This would greatly help us in making any necessary adjustments or clarifications.

Thank you very much for your assistance and understanding.

Best regards, :-)

---

### Meta-Review · Area_Chair_9HaH · 2024-12-22

**Metareview:**

This paper presents ChartBench, a dataset centered on unannotated charts, plus a new metric. Reviewers liked its broad chart coverage and detailed evaluations. However, they found its technical contributions modest, with heavy reliance on synthetic data and unclear real-world applicability. Despite the dataset’s potential value, reviewers felt the improvements over existing benchmarks weren’t compelling enough. Due to the lack of strong support, the overall recommendation from the AC is reject.

**Additional Comments On Reviewer Discussion:**

During the rebuttal, reviewers raised concerns about the precision of Acc++ and its real-world applicability, leading to additional experiments. They also asked for results from newer models. The authors responded with new experiments and further clarifications, but despite these efforts, the reviewers still considered the improvements incremental and the real-world impact unclear. Notably, Reviewer FpUA highlighted concerns about the metric’s ambiguity and reliability. Although the authors introduced a new NQA analysis in response, the AC believes more extensive revisions and discussions are needed to ensure the robustness of the benchmark.

---

### Decision · Program_Chairs · 2025-01-22

Reject